# Identification of a deep-branching lineage of algae using environmental plastid genomes

Mahwash Jamy [1], Thomas Huber [2], Thibault Antoine [3,4], Hans-Joachim Ruscheweyh [5], Lucas Paoli [6], Eric Pelletier [3,7], Tom O. Delmont [3,7] ✉ & Fabien Burki [2] ✉

Marine algae underpin entire ocean ecosystems. Yet algae in culture poorly represent their large environmental diversity, and we have a limited understanding of their convoluted evolution by endosymbiosis. Here, we perform a phylogeny-guided plastid genome-resolved metagenomic survey of *Tara* Oceans expeditions. We present a curated resource of 660 new non-redundant plastid genomes of environmental marine algae, vastly expanding plastid genome diversity within major algal groups, including many without closely related reference genomes. Notably, we recover four plastid genomes, including one near-complete, forming a deep-branching plastid lineage of nano-size algae that we informally name leptophytes. This group is globally distributed and generally rare, although it can reach relatively high abundance in the Arctic. A near-complete mitochondrial genome showing strong co-occurrence with leptophyte plastids is also recovered and assigned to this group. Leptophytes encompass the enigmatic plastid group DPL2, one of the very few known plastid groups not clearly belonging to major algal groups and previously known only from 16S rDNA sequences. Comparative organellar genomics and phylogenomics indicate that leptophytes are sister to haptophytes, and raise the intriguing possibility that cryptophytes acquired their plastids from haptophytes. Collectively, our study demonstrates that metagenomics can reveal hidden organellar diversity, and improve models of plastid evolution.

Eukaryotic algae are essential to the success of life on earth, performing nearly 40% of primary production and forming the basis of many food webs[1]. Algae are incredibly diverse in forms and functions, ranging from some of the smallest known eukaryotes in the open ocean phytoplankton to undersea forests of giant coastal seaweed. The phylogenetic position of algae in the eukaryotic tree of life shows that

they have evolved in multiple supergroups, owing their origins to a complex and largely unresolved history of endosymbioses. The first eukaryotic algae originated from the endosymbiotic integration of cyanobacteria into predatory host cells. This primary endosymbiosis led to the diversification of eukaryotes with photosynthetic organelles called primary plastids in the Archaeplastida supergroup, containing

[1]Department of Aquatic Sciences and Assessment, Swedish University of Agricultural Sciences, Uppsala, Sweden. [2]Department of Organismal Biology, Program in Systematic Biology, Uppsala University, Uppsala, Sweden. [3]Génomique Métabolique, Genoscope, Institut François Jacob, CEA, CNRS, Univ Evry, Université Paris-Saclay, Evry, France. [4]Research Federation for the Study of Global Ocean Systems Ecology and Evolution, FR2022/Tara GOsee, Paris, France. [5]Department of Biology, Institute of Microbiology and Swiss Institute of Bioinformatics, ETH Zürich, Zürich, Switzerland. [6]Global Health Institute, School of Life Sciences, École Polytechnique Fédérale de Lausanne (EPFL), Lausanne, Switzerland. [7]Present address: Research Federation for the Study of Global Ocean Systems Ecology and Evolution, FR2022/Tara GOsee, Paris, France. ✉e-mail: tomodelmont@gmail.com; fabien.burki@ebc.uu.se

major algal groups such as red algae and green algae (including land plants)[2,3]. Subsequently, plastids were transferred from red and green algae to unrelated groups of eukaryotes by further endosymbioses, spreading photosynthetic capabilities throughout the tree of eukaryotes[4]. Major questions related to plastid evolution remain open, notably about the number of endosymbioses and partners involved to explain the observed phylogenetic distribution of plastids. The evolutionary history of complex plastids with red algal origin has been among the most difficult to resolve, in spite of the ecological importance of some of these algae in modern oceans, such as diatoms, dinoflagellates, and coccolithophores[5,6]. Multiple hypotheses have been proposed, many involving successive endosymbioses between eukaryotes over vast evolutionary distances, but there is no consensus and very little empirical evidence to support one or another hypothesis[2,7–9].

A more complete catalogue of plastid genomes across the algal tree is needed to better understand the evolution of plastid endosymbiosis. In recent years, comparatively more data has been produced for nuclear genomes of algae—including many transcriptomes—than for the smaller and in principle easier to sequence plastid genomes[10]. There are currently several major groups of algae that have poor representation of plastid data. For example, only 17 plastid genomes are available for haptophyte species, one of the most abundant and bloom-forming marine algal groups containing as many as 1000 described species and a much larger environmental diversity[5]. Beyond known algae, a few so-called deep-branching plastid groups, i.e., plastids that do not belong to any established algal groups, have been reported but currently lack any genomic or morphologic description. The discovery of deep-branching plastid groups is extremely rare, comparable to the discovery of new animal groups[11], owing to a good understanding of what the major algal groups are. The most notable cases of new deep-branching groups in the past 20 years include the rappemonads—but they have since been formally assigned to haptophytes into the new class Rappephyceae[12]—or the enigmatic environmental plastid lineages DPL1 and DPL2 for which only 16S rDNA amplicon sequences are available[13]. While DPL1 was weakly placed within haptophytes but outside of known classes, DPL2 branched deeper, possibly as sister to haptophytes but without statistical support[13]. Finally, the Picozoa (previously picobiliophytes), which were originally described as a new algal group with unknown affinities to eukaryotes, are now considered aplastidic members of Archaeplastida[14].

Here, we used 280 billion metagenomic reads covering a wide range of *Tara* Oceans planktonic size fractions[15,16] as well as additional resources[17,18] to assemble and manually curate 660 non-redundant environmental plastid genomes (ptMAGs). We show that genome-resolved metagenomics is a powerful approach for reconstructing organellar genomes and use this unprecedented plastid genomic resource to determine the phylogeny and geographical distribution of planktonic algae. We report a generally rare but widespread and deep-branching clade of plastid genomes encompassing the enigmatic DPL2 that we informally name leptophytes. From the few Arctic samples where leptophytes were the most abundant, we also recovered a mitochondrial contig that was assigned to this group based on its phylogenetic position and highly significant positive correlation in read coverage. Based on the phylogenetic position of both plastid and mitochondrial genomes, we discuss the implications of leptophytes for our understanding of plastid endosymbioses.

## Results

### A large resource of environmental plastid genomes from the sunlit ocean

To characterise the diversity of eukaryotic algae in the global sunlit ocean, we surveyed large *Tara* Oceans metagenomic co-assemblies (~12 million contigs >2.5 kb) covering plankton size fractions ranging from 0.8 μm to 2 mm (Supplementary Data 1)[15,16]. These co-assemblies, each deriving from samples from a particular oceanic basin, have been shown to improve the recovery of genomes that are too scarce to be assembled from individual samples[16,19]. Briefly, we used a phylogeny-guided metagenomic approach based on the RNApolB (*rpoB*) gene, and complementary genomic and environmental information to characterise and manually curate an initial set of 1,448 plastid metagenome-resolved genomes (ptMAGs) with anvi'o[20,21] (see Methods). Two additional single-contig ptMAGs related to the focal clade of our study (see subsequent sections) were subsequently added from mOTUs-db[18]. After filtering for redundancy (ANI >98%), our final plastid genome database derived from all oceanic regions consisted of 660 new ptMAGs, as well as 166 reference plastid genomes (Supplementary Fig. 1, Supplementary Data 1). The occurrence of 44 single copy plastid core genes indicated that the ptMAGs are 62.9% complete on average, with low redundancy (average: 2.5%) (Supplementary Fig. 2). 126 ptMAGs contained at least 40 of the 44 core plastid genes and were thus deemed near complete. However, no ptMAG mapped as a circular sequence due to the absence of the ribosomal RNA operon which is often missing from MAGs. The ptMAGs had a median size of 65.7 kbp, with GC content ranging between 24.9% and 46.3%, and encoded up to 168 genes (Supplementary Data 1; Supplementary Figs. 3-5). The ptMAG abundance, as estimated by sequencing depth, varied by more than four orders of magnitude, ranging from a chlorophyte ptMAG with ~21,000× coverage to a diatom ptMAG with only ~2× coverage across all *Tara* Oceans metagenomes (Supplementary Data 1). Collectively, the ptMAGs were far more abundant in the sunlit oceans compared to selected culture representative genomes (82.5% of the total signal), in line with a previous comparison of nuclear genomes among the same metagenomes[16] (Supplementary Fig. 6).

We reconstructed a broad multigene phylogeny of our plastid genome database to assess its taxonomic diversity. Given the large taxon sampling, we inferred a Maximum Likelihood phylogeny using the relatively simple site-homogeneous LG + F + I + G4 model based on 93 plastid-encoded genes. This phylogeny recovered all major groups of algae and is overall in good agreement with previously published plastid trees (Fig. 1, Supplementary Fig. 7)[9,22]. The ptMAGs occurred in most, but not all, major groups of algae. We did not recover ptMAGs for red algae and glaucophytes, which are known to be rare or absent in the open ocean[23], nor for the divergent peridinin plastids of dinoflagellates, which are highly fragmented and lack the *rpoB* marker gene[24]. Conversely, the ptMAGs greatly expanded the known plastid genome diversity of ochrophytes (n = 375, including 304 diatoms displaying a very broad cellular size range), and haptophytes (n = 196, including Phaeocystales found in large size fractions likely due to their distinct colony-forming lifestyle[5]). A smaller number of ptMAGs were also recovered for cryptophytes (n = 16, restricted to the small size fractions), and green algae (n = 45, predominantly found in the smallest size fraction). In multiple cases, the ptMAGs represented deep-branching novel genome diversity within established algal groups where cultured references are lacking. For instance, one ptMAG (REFM_CHLORO_00002) deeply branched as sister to Pavlovales (Fig. 1 and Supplementary Fig. 7). Eight chlorophyte ptMAGs recovered from different oceanic regions formed a monophyletic clade distantly related to a clade of Chloropicophyceae. Within ochrophytes, 15 ptMAGs with a broad geographic distribution were closely related but clearly different from the reference pelagophyte plastid genomes (Supplementary Fig. 8). Finally, among green algae, 13 ptMAGs with a large variation in genome sizes formed a distinct clade related to pedinophyte endosymbionts of dinoflagellates, which include the undescribed MGD and TGD[25] as well as *Lepidodinium chlorophorum*[26] (Supplementary Fig. 9). This group might represent a larger diversity of endosymbionts than previously recognised. Taken together, these examples demonstrate the usefulness of metagenomics to recover a broad diversity of plastid genomes among

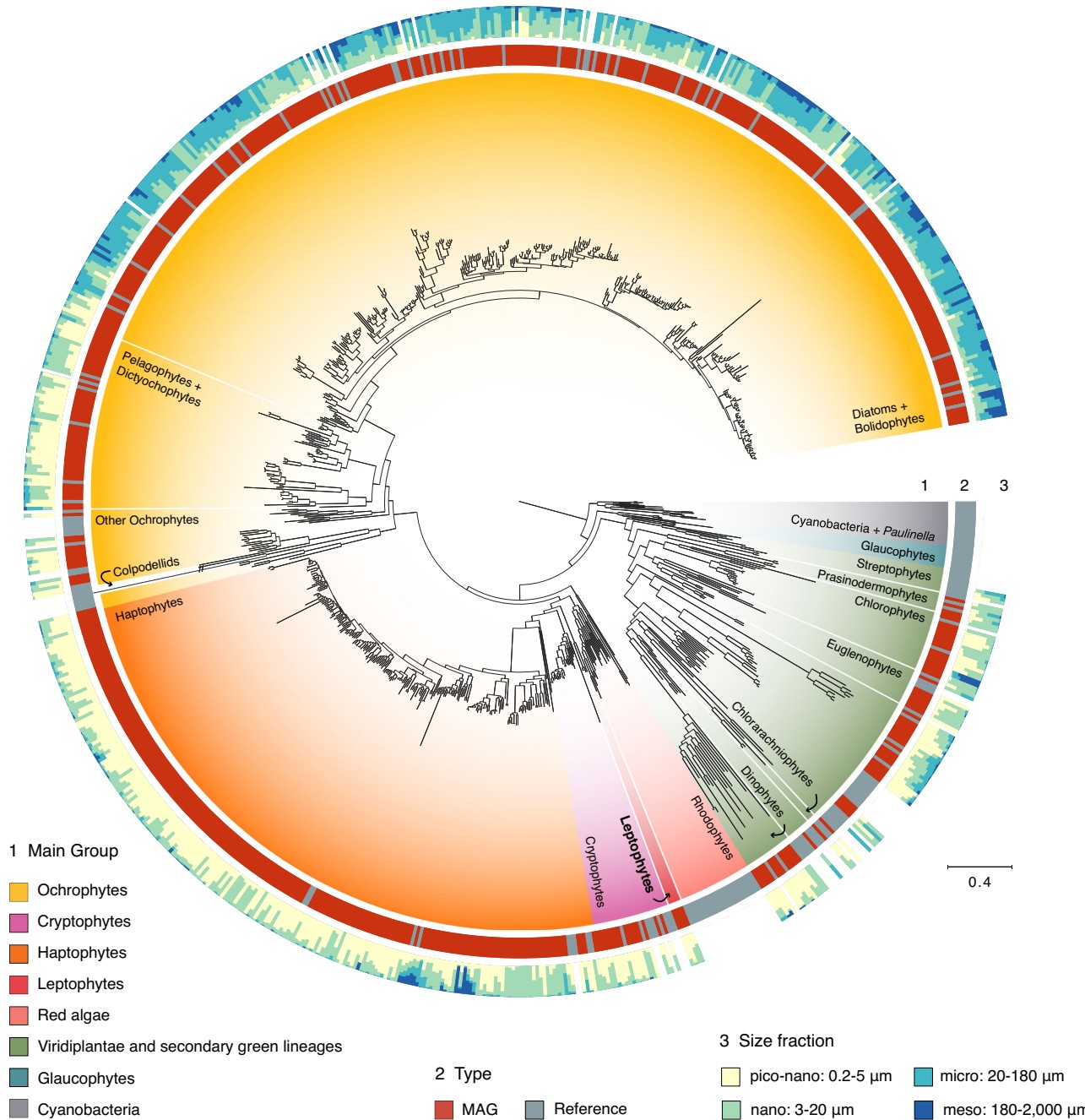

**Fig. 1 | Global phylogenetic analysis of plastid genomes from the sunlit ocean.** Maximum-Likelihood phylogeny inferred using 93 plastid-encoded genes and the LG + F + I + G4 model in IQ-TREE. The phylogeny contains 179 manually selected reference genomes (166 plastid and 13 cyanobacterial genomes) and 660 plastid MAGs (ptMAGs) from various algal groups; cyanobacteria (including the *Paulinella* plastid) were set as outgroup. Ring 1 around the phylogeny indicates algal groups; Ring 2 depicts whether the taxon is a reference sequence or a MAG; Ring 3 depicts the estimated size of each marine taxon as determined by metagenomic mapping of 937 *Tara* Ocean metagenomes (see Methods). Colpodellid branches have been shortened to 40% of their original length to improve visual clarity. The full phylogeny with taxon labels is shown in Supplementary Fig. 7.

multiple major groups, in particular filling critical evolutionary gaps with environmental genomic diversity for clades lacking reference genomes.

## Leptophytes: a new deep branching plastid group

Aside from the extended genomic diversity within clearly defined plastid groups, we also recovered a fully supported deep-branching clade formed by four red algal-derived ptMAGs that is related to, but distinct from all major groups with complex red plastids, i.e., haptophytes, cryptophytes, ochrophytes, and myzozoans (Fig. 1). To

facilitate discussion, we propose the informal name leptophytes for this new group of plastid genomes. The name refers to the putative small size (*lepto-* indicates something small) of the corresponding algal cells, which we infer to be below 5 μm based on genomic size fraction distributions across the *Tara* Oceans metagenomes (Supplementary Fig. 10). Leptophytes were collectively detected in the surface waters of most oceanic basins (Supplementary Fig. 11). At the level of individual genomes, Lepto-01 and Lepto-02 were only detected in the Arctic Ocean characterised by cooler, less saline, and more nutrient-rich waters, while Lepto-03 and Lepto-04 showed a broader and non-polar

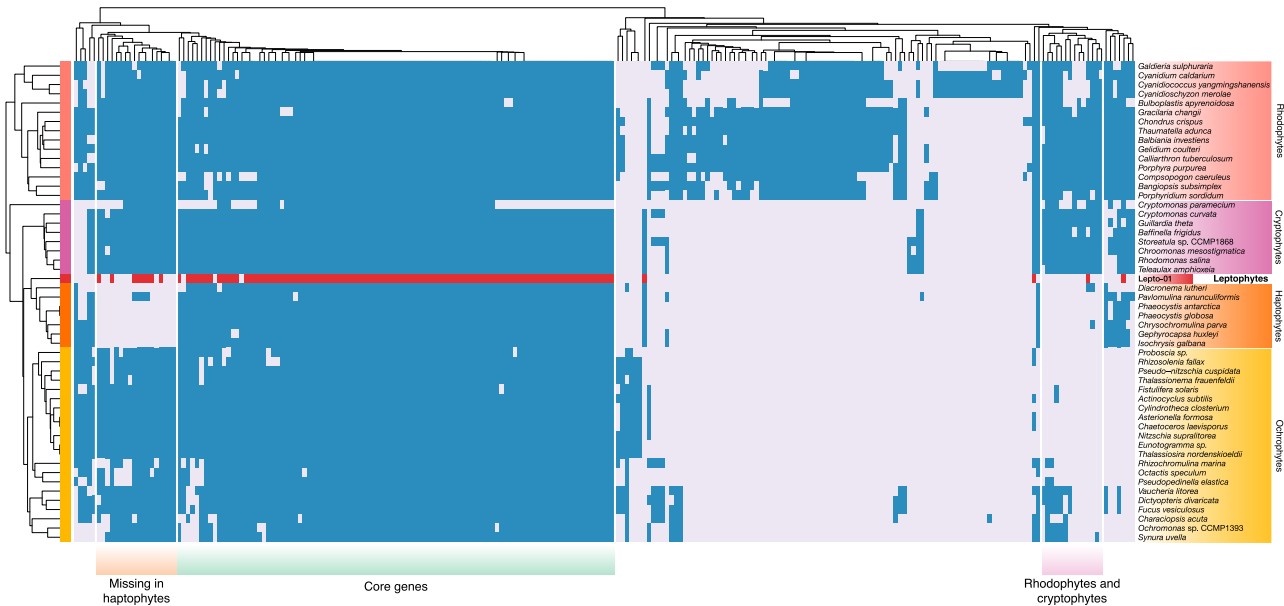

**Fig. 2 | Distribution patterns of plastid-encoded genes across algae with red plastids.** A binary heatmap and dendrograms of genes and taxa generated using the UPGMA algorithm, based on the presence or absence of 237 plastid protein-coding genes. Blue and grey boxes represent gene presence and absence, respectively. For leptophytes, present genes are highlighted in red for improved clarity. A larger version of this figure, including gene names, is presented in Supplementary Fig. 17.

distribution associated with warmer, more saline, and nutrient-poor waters (Supplementary Figs. 12-13). Although leptophyte genomes were usually found in relatively low abundance (representing 0.14% of the total plastid signal, and detected in 104 out of 147 *Tara* stations with a median cumulative coverage of 2.3×), Lepto-01 stood out as one of the most detected plastid genomes in two Arctic stations (up to 124× mean coverage in the 0.8-2000 μm size fraction), providing evidence that leptophytes can experience localised increases in abundance under certain conditions (Supplementary Table 1). Interestingly, two 16S rDNA gene fragments were recovered in Lepto-01 and Lepto-04, respectively, which revealed that leptophytes are most similar to a few amplicon sequences of the enigmatic DPL2 plastid group[13]. The 16S rDNA of Lepto-04 was 100% similar to DPL2, while Lepto-01 shared 91% similarity to DPL2. The relatedness of DPL2 to leptophytes was confirmed by 16S rDNA phylogeny, which also showed a weak affinity of the group to haptophytes (Supplementary Fig. 14). The reported subtropical distribution and pico-size of DPL2[13] largely overlap with Lepto-03 and Lepto-04 but not with the Arctic preference of Lepto-01 and Lepto-02, thus altogether indicating that leptophytes form a more diverse and widespread group than the narrower DPL2 lineage.

Lepto-01 is the best representative genome for the group. This assembly had the highest coverage (detected in 20 stations with an average mean coverage of 6.3×) but it was also near complete and contiguous with a single contig of 104,203 bp. This ptMAG contained 144 genes with 118 proteins (no detectable introns) and 24 tRNAs (Supplementary Fig. 15). Interestingly, leptophyte plastid genomes seem to lack inverted repeats commonly found in plastid genomes[27], although whether they instead contain direct repeats such as in some haptophytes[28], or lack repeat regions altogether remains unresolved (Supplementary Fig. 15). The functional repertoire of Lepto-01 shows that it is a photosynthetic plastid genome, retaining genes for the core components of both photosystem I and II (*psa*- and *psb*- genes), carbon fixation (*rbcL* and *rbcS*), cytochrome b6/f complex (*pet*- genes), ATP synthase (*atp*- genes), and chlorophyll biosynthesis (*chlI*) (Fig. 2, Supplementary Fig. 15). While less complete and more fragmented, the three other leptophyte ptMAGs are largely syntenic with Lepto-01 and support these functional trends (Supplementary Fig. 16). The near

complete assembly of Lepto-01 allowed us to compare its gene content with that of red algae and red algal-derived plastids. UPGMA-clustering of homologous gene occurrence showed that the gene content of leptophytes is most similar to that of haptophytes (Fig. 2). In particular, leptophytes are lacking ten genes (including *dnaB*, *ftsH*, *petF*, *psbW*, *rpl4*, *rpl18*, *rpl29*, *rpl35*, *sufC*, *ycf33*) that are among the 18 genes almost completely absent in haptophytes, but mostly present in all other algal groups with plastids of red-algal origin (Fig. 2, Supplementary Fig. 17).

## Leptophyte plastids have phylogenetic affinities with haptophytes and cryptophytes

To gain further insight into the position of leptophytes in the plastid tree, we reconstructed and evaluated phylogenies using several complex site-heterogeneous mixture models in Bayesian and ML implementations. These models, which consider site-specific amino-acid frequencies, have been shown to generally improve tree estimations but they come at heavier computational costs[29]. We thus produced a reduced dataset based on the same 93 genes as in the full phylogeny but containing 107 plastid genomes to speed up the analyses while maintaining the phylogenetic breadth for red algae and red algal plastid-containing lineages. Due to overall higher completeness, reference genomes were preferred over ptMAGs when possible, and the least complete leptophyte ptMAG (Lepto-02) was excluded. The battery of advanced phylogenetic analyses all unambiguously confirmed that leptophytes form a novel group of plastid genomes related to haptophytes and cryptophytes (Fig. 3a). This affinity is also supported by a shared rare bacterial gene replacement by horizontal transfer of the ribosomal protein gene *rpl36* into the plastid genomes of all three groups (Fig. 3b). We will refer to this group as the Cryptophytes-Haptophytes-Leptophytes (CHL) group.

Leptophytes were recovered as sister to both haptophytes and cryptophytes (HC-sister topology) with the Bayesian model CAT + GTR + G4 (PP = 0.99), as well as the ML models GTR + CAT-PMSF + G4 (BS = 88) and LG + MEOW80 + G4 (UFB = 61) in which site profiles are estimated directly from the data[30,31]. The same position for leptophytes was recovered with the empirical mixture model cpREV+C60 + G4 model and in an analysis that removed the 10 fastest-evolving taxa

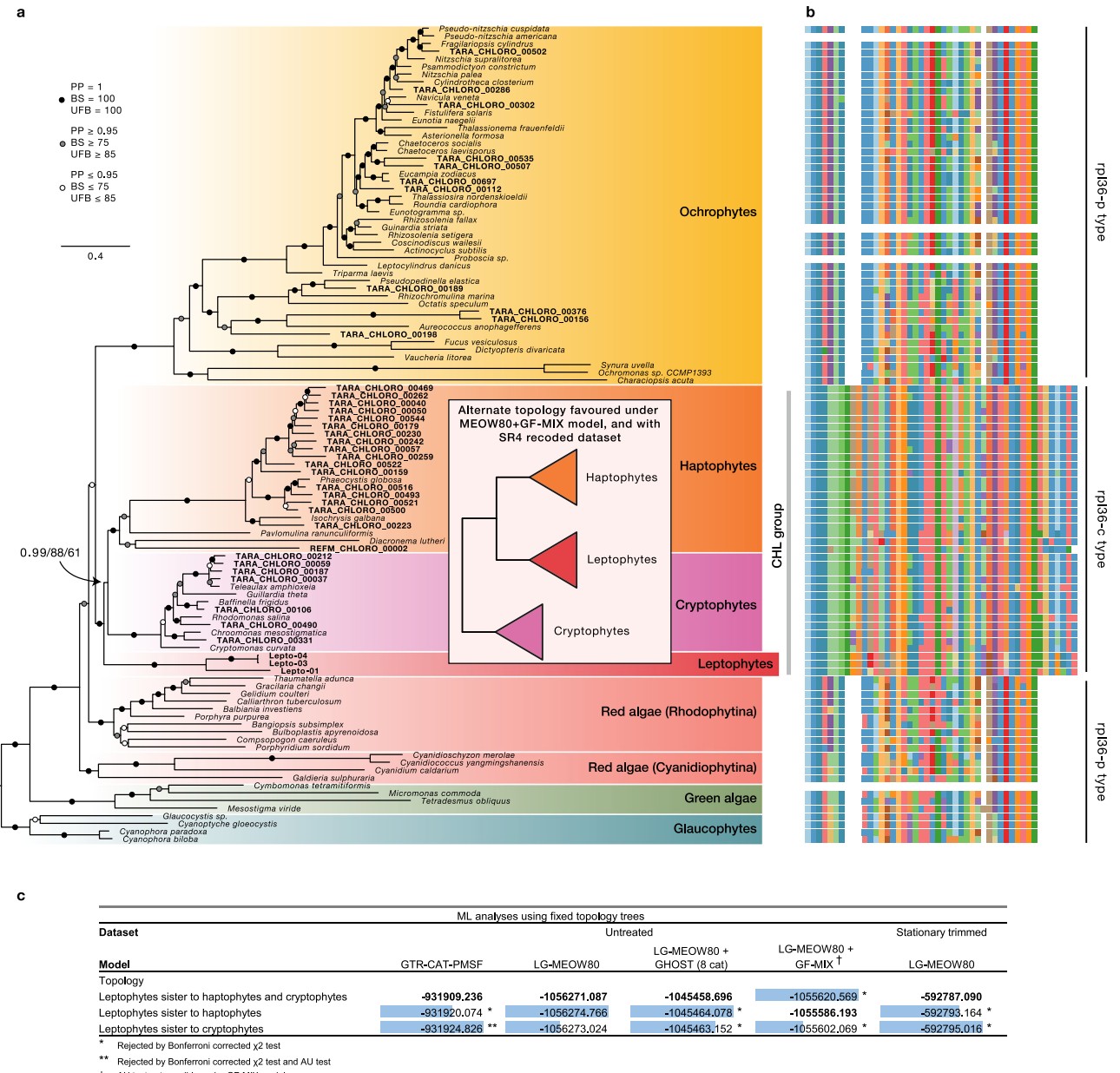

**Fig. 3 | Phylogenomic analyses based on a reduced taxon-sampling. a** Bayesian inferencece based on 20,292 amino acid sites across 107 taxa under the CAT + GTR + G4 model. Branch support values are given in the following order: Bayesian posterior probabilities (PPs), non-parametric bootstrap support (BS; 100 replicates) under the GTR + CAT-PMSF model, and ultrafast bootstrap support (UFB; 1000 replicates) under the LG + MEOW80 + G4 model. The inset shows the alternative topology of the CHL clade. **b** Alignment of the rpl36 gene across corresponding taxa in the tree. **c** Maximum likelihood estimates under different models for alternate positions of leptophytes within the CHL clade. Likelihoods in bold text represent the highest scoring topology. Difference in log-likelihood scores between each topology and the best-scoring topology is shown by data bars in blue. A single asterisk indicates topologies rejected by the Bonferroni-corrected chi-squared test, while two asterisks indicate topologies also rejected by an Approximately Unbiased (AU) test. See also Supplementary Table 2 and Supplementary Data 2 for corresponding p-values and degrees of freedom.

(Supplementary Figs. 18–19). Because of the relatively modest bootstrap support for the position of leptophytes in ML inferences, we compared the likelihood of the three possible topologies within the CHL group, enforced as topological constraints but letting other branches and parameters to be optimised by ML. Under the GTR + CAT-PMSF model, both alternative topologies were rejected by a likelihood chi-squared test with a Bonferroni correction for multiple comparisons[32], and the sister position to cryptophytes was additionally rejected by an AU test (Fig. 3c). The likelihood values under the LG + MEOW80 + G4 model had less than four points difference across the three topologies, and none were statistically rejected (Fig. 3c).

We then evaluated whether the position of leptophytes was influenced by two common types of systematic errors in phylogenetic reconstruction: heterotachy of evolutionary rates, and compositional heterogeneity across tree branches. To do this, we calculated the likelihoods of the three alternate phylogenies as fixed topologies under newly developed complex models as additional parameters to the LG + MEOW80 + G4 model. Heterotachy was modelled with GHOST[33], while compositional heterogeneity was accounted for with the GFmix model[31,32]. The GHOST model continued to favour the HC-sister topology, but the best topology under the GFmix model changed to support the monophyly of leptophytes and haptophytes

(H-sister topology) (Fig. 3a, c). With both models, the alternative tested topologies were rejected by the Bonferroni-corrected likelihood chi-squared test (Fig. 3c). Given the observation that compositional bias may have impacted our phylogenetic reconstructions, we attempted to minimise this artefact by recoding the 20 amino acids to a reduced 4-class alphabet (SR4 recoding)[34], and by removing the most heterogeneous sites (~25% of positions in the alignment) with a stationary-based trimmer[35]. In agreement with the GFmix model, the recoded dataset recovered the H-sister topology, albeit with low support (PP = 0.75; Supplementary Fig 20). However, the stationary-trimmed alignment recovered the HC-sister topology, also strong support under the CAT + GTR + G model (PP = 0.99, but low support with the LG + MEOW80 + G model (UFB = 72) (Supplementary Fig. 21). Overall, these results indicate that resolving the topology of the CHL group in general and placement of leptophytes in particular is not trivial, as both HC-sister and H-sister topologies remain credible (Supplementary Note 1).

### A mitochondrial genome places leptophytes as sister to haptophytes

Plastid genomes do not necessarily mirror the long-term evolutionary trajectory of their host cell because of the complex history of endosymbioses[2,4]. In an effort to identify potential nuclear-encoded sequences for leptophytes, we searched the Arctic stations data−where the leptophyte abundance was the highest−for *psbO* and 18S rDNA candidate genes (see Methods). The *psbO* gene is essential for photosynthesis, it is always host-encoded and never found in non-photosynthetic organisms[36]. Although it is derived from the endosymbiont and thus would not be a good phylogenetic marker for the host, our rationale was that it could pinpoint to host contigs that may contain additional genes. These analyses did not provide promising candidates for either gene.

We then turned to mitochondria, as mitochondrial genomes have the double advantage of typically being multi-copy and in theory mirroring the evolution of the host as this organelle traces back to the origin of eukaryotes[31,37]. From the six Arctic samples where the Lepto-01 ptMAG was the most abundant, we performed a targeted metagenomic co-assembly (total of 2.54 billion metagenomic reads providing a cumulative coverage of 299× for the Lepto-01 ptMAG) that produced 56,103 contigs larger than five kb. Functional annotations based on reference mitochondrial genomes resulted in the identification of 34 mitochondrial contigs (mtMAGs, Supplementary Data 3).

To find a possible correspondence with the Lepto-01 ptMAG, mtMAGs needed to fulfil two complementary requirements. Most importantly, their read coverage should show strong correlation with the Lepto-01 ptMAG across the *Tara* Oceans metagenomes. Secondly, their phylogenetic position among reference mitochondrial genomes should not be placed within known major algal groups. Among the 34 mtMAGs, one genome (Lepto-01_mtMAG_004) showed highly significant read coverage correlation with the Lepto-01 ptMAG ($R^2 = 0.96$, p-value < 0.01) (Fig. 4, Supplementary Fig. 22). Most interestingly, Lepto-01_mtMAG_004 also corresponded to a novel deep-branching lineage firmly placed as sister to haptophyte mitochondrial genomes (Fig. 4; UFB = 100%, PP = 1.0). This mtMAG is 37,793 bp long and contains 37 protein coding genes as well as 22 tRNA genes (Supplementary Fig. 23), with the presence of various marker genes indicating that it is nearly complete. Based on both criteria of abundance and phylogeny, we infer that Lepto-01_mtMAG_004 and the Lepto-01 ptMAG are from the same novel host lineage related to haptophytes.

## Discussion

The vast majority of organellar genomic knowledge has been gained from cultured organisms. We demonstrate that plastid and mitochondrial genomes can readily be recovered from large metagenomic

assemblies using phylogeny-guided genome-resolved metagenomics. This study represents the first global-scale survey of environmental plastid genomes (ptMAGs), providing a culture-independent view of the diversity, functioning and evolutionary history of plastid genomes occurring in marine surface water. ptMAGs better represent the diversity and abundance of planktonic algae than reference genomes, spanning most major marine phytoplanktonic groups and often filling in evolutionary gaps in areas of the tree poorly represented by references.

One very interesting new group of plastid genomes are the leptophytes, a globally distributed and generally rare deep-branching lineage formed by four ptMAGs in our data. Novel plastid diversity at this taxonomic depth is very rarely reported. Currently, only the environmental plastid lineages DPL1 and especially DPL2 are still considered possible deep-branching phytoplankton lineages outside of eukaryotic supergroups[13]. Based on 16S rDNA data recovered from the ptMAGs and biogeographical comparison, we show not only that one of the leptophytes (Lepto-04) corresponds to DPL2, thus representing the first genomic data available for this enigmatic group, but also that leptophytes form a more diverse, widespread, and abundant group than previously known. Part of our findings have now been independently recapitulated, with the notable recovery of the Lepto-01 ptMAG from a different metagenomic dataset[38].

The plastids of leptophytes are evolutionarily related to cryptophytes and haptophytes, indicating that they possess red-algal derived plastids. This is firmly supported by phylogenetic analyses as well as gene content comparison and the exclusive presence of the c-type paralog of the ribosomal protein gene *rpl36* in cryptophytes, haptophytes and leptophytes (the CHL group). The exact phylogenetic position of leptophyte plastids within the CHL group remains ambiguous in spite of the use of a dense taxon-sampling, 93 genes, and a range of sophisticated evolutionary models and data treatments. Taking into account this phylogenetic uncertainty, we propose two competing hypotheses for the evolutionary origin of leptophyte plastids: 1) the HC-sister topology (leptophytes are sister to both cryptophytes and haptophytes), which was favoured by all site-heterogeneous models, or 2) the H-sister topology (leptophytes are sister to haptophytes, to the exclusion of cryptophytes), which was favoured only after specifically taking into consideration amino acid compositional heterogeneity across taxa, and also by the 16S rDNA phylogeny.

In contrast to the robust support for most bipartitions in the plastid tree (Fig. 3a), the ambiguity in the placement of leptophytes likely resulted from weak stochastic phylogenetic signals due to the ancient origin of the plastids and closely spaced speciation events early in the CHL group. Although a precise timeframe for the origin of the CHL plastids is unavailable, it has been suggested that both cryptophytes and haptophytes acquired their plastids more than 1 billion years ago[39]. Based on the phylogenetic position of leptophytes, it is reasonable to assume that this timeframe represents a minimum age estimate for the divergence of this group. This ancient age, combined with the extremely short internal branch uniting haptophytes and cryptophytes (Fig. 3) and the limited phylogenetic signal in plastid genomes, means that it is probably not possible with our current phylogenetic models and data to better resolve this part of the plastid tree.

Unlike the unsettled phylogenetic position of the leptophyte plastids, we retrieved a near complete mitochondrial contig that we have assigned to leptophytes based on highly significant abundance correlation and well supported sister relationship to haptophytes. This is particularly interesting because unlike plastids, mitochondria have been vertically transmitted throughout the evolution of eukaryotes and thus show the history of the hosts[31]. This phylogenetic position bears important implications on our understanding of the origin and spread of complex red plastids across the eukaryotic tree. To explain

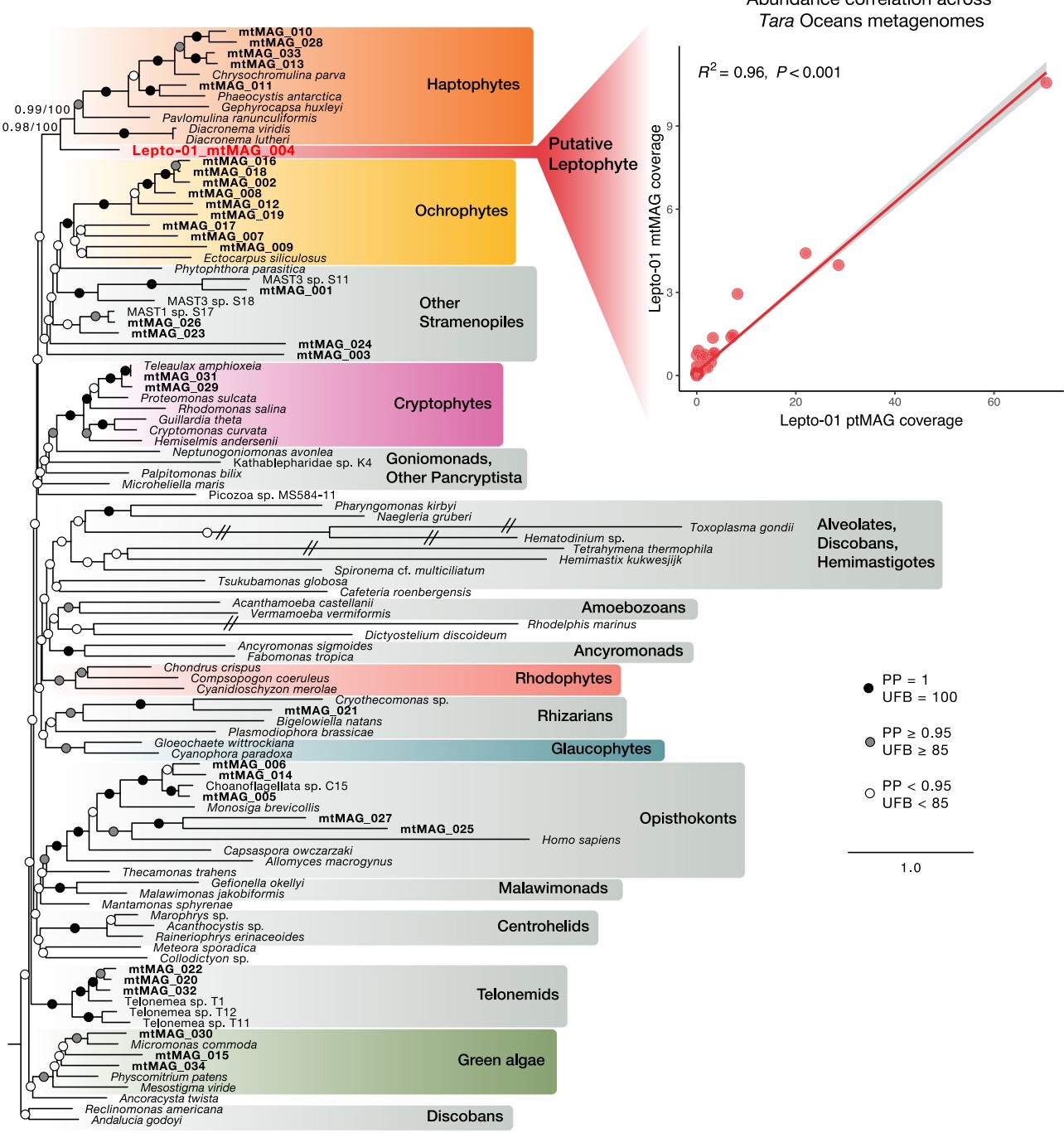

**Fig. 4 | Phylogeny and abundance patterns of a putative leptophyte mito-chondrial genome.** Maximum-Likelihood phylogeny inferred using 28 mitochondrial genes in IQ-TREE. Support values on branches indicate Bayesian posterior probabilities (PPs) under the CAT + GTR + G model and ultrafast bootstrap support (UFB; 1000 replicates) under the LG + C60 + G4 model. The phylogeny contains 34 mitochondrial MAGs (mtMAGs, highlighted in bold) recovered from a coassembled metagenomic dataset of the six *Tara* Oceans samples with the highest Lepto-01 abundance, and 68 manually selected reference genomes. The phylogeny was arbitrarily rooted with jakobids (discobans). The inset shows the abundance correlation between the putative Lepto-01 mtMAG and the Lepto-01 ptMAG across *Tara* Oceans metagenomes from the 0.22–3 μm size fraction. The p-value ($p = 2.1 \times 10^{-75}$) was calculated from a two-sided *t*-test of the regression slope (H₀: slope = 0). No correction for multiple comparisons was applied. Source data is provided in the associated code repository[82].

the patchy distribution of these plastids in the tree, the existing models of serial endosymbiosis all identify cryptophytes as the unique recipient of secondary red plastids (plastids derived directly from red algae), mostly because cryptophytes are the only known algae that have retained a red algal nucleomorph[40,41]. Most models also specifically propose a plastid transfer between cryptophytes and haptophytes to take into account the rare *rpl36* gene replacement into their plastid genomes[42] and affinities of both groups in several plastid

phylogenies[22,43]. Our new plastid and mitochondrial phylogenies are consistent with available data, but adding leptophytes allows us to make new predictions on the relative timing of events and possible direction of plastid transfer.

Reconciling our two hypotheses for the placement of leptophytes in the plastid phylogeny with the best current estimate of eukaryotic phylogeny[44], we propose two models for the evolution of complex red plastids, focusing here on the CHL group for clarity (Fig. 5). Both

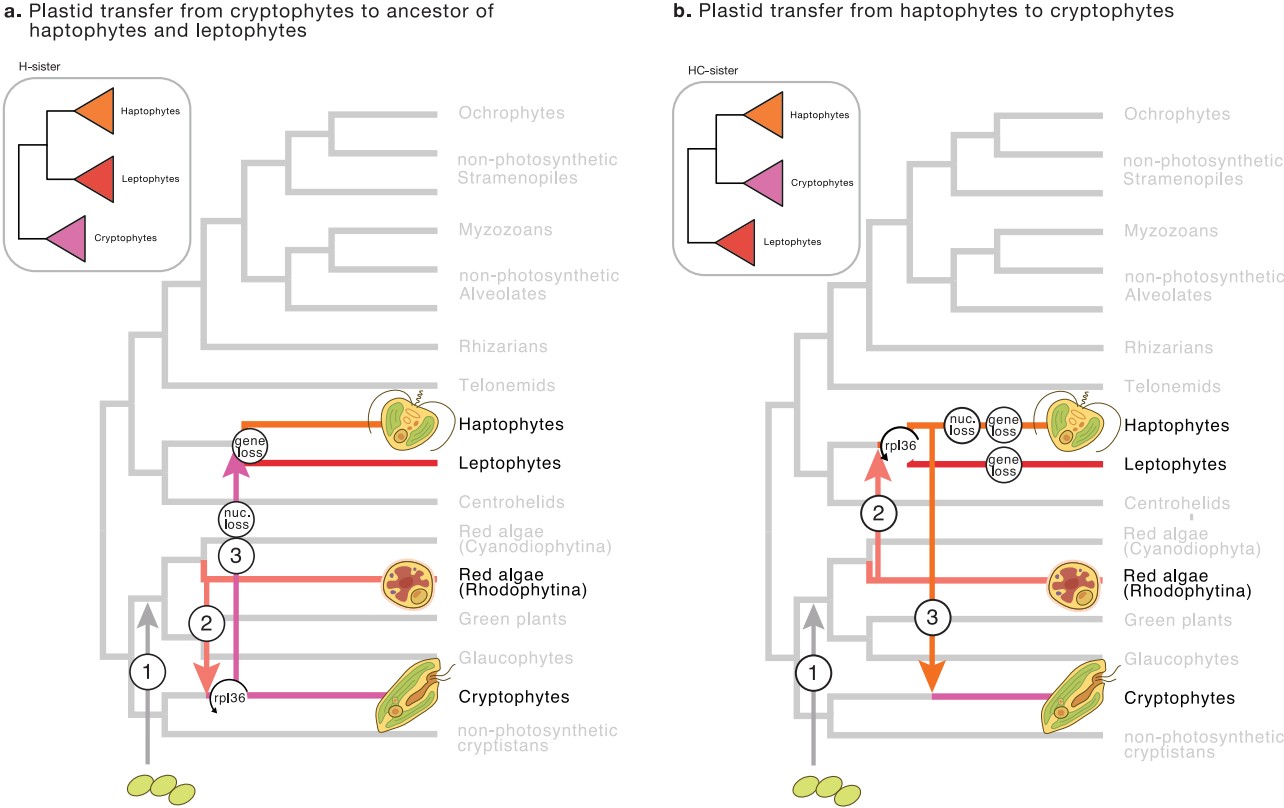

**a.** Plastid transfer from cryptophytes to ancestor of haptophytes and leptophytes

**b.** Plastid transfer from haptophytes to cryptophytes

rpl36 rpl36 replacement    nuc. loss nucleomorph loss    gene loss gene loss from plastid genome

**Fig. 5 | Main proposed models for evolution of complex red plastids. a** Scenario of endosymbiosis events if leptophyte plastids are sister to those of haptophytes, and **b** scenario if leptopyte plastids are sister to both haptophyte and cryptophytes plastids. The main trees depict hypothetical host relationships with red algae, cryptophytes, haptophytes, and leptophytes highlighted. Numbers on the tree represent the level of endosymbiosis events. Inset phylogenies show the corresponding plastid phylogenies. Plastid transfers to ochrophytes and myzozoans are not depicted for clarity. Illustrations of algae are from ref. 83.

models assume a sister relationship of leptophytes to haptophytes in the eukaryote tree based on our mitochondrial phylogeny. Whether this means that haptophytes should be expanded to include lepto-phytes as a new deeply diverging class, or that leptophytes is a distinct but related lineage, will require detailed morphological information (such as the presence of a haptonema) not available at the moment. The first model presumes that leptophytes are sister to haptophytes (H-sister) in the plastid phylogeny as well (Fig. 5a). Leptophytes would have inherited their plastids, including the *rpl36*-c paralog, in a com-mon ancestor with haptophytes by tertiary endosymbiosis from cryptophytes and would lack a nucleomorph. The second model reconciles the HC-sister plastid topology with the position of lepto-phytes as sister to haptophytes in the host phylogeny (Fig. 5b). In this scenario, the secondary plastid from red algae was established in the common ancestor of leptophytes and haptophytes, where the ances-tral *rpl36*-p gene was replaced, before passing that plastid further to cryptophytes from a stem haptophyte. To our knowledge, this model is the first to propose the haptophyte ancestor as the host of the secondary endosymbiosis with red algae, and is in line with the general older origin of haptophytes over cryptophytes inferred by molecular clock analysis[39]. The model places the loss of nucleomorph in hapto-phytes subsequent to the tertiary transfer to cryptophytes, but leaves open the possibility that leptophytes have also retained this organelle. To weigh in on both models, we searched for evidence of red algal nucleomorph in the same metagenomic data used to reconstruct the ptMAGs. While we found ample evidence of cryptophyte nucleo-morphs, we found no evidence of putative leptophyte nucleomorphs (see Methods). Without a more conclusive plastid phylogeny, or other

supporting evidence, it is currently not possible to decide which model, if any, is correct. However, we note that the H-sister model is more parsimonious as it requires fewer parallel gene losses based on our plastid gene content comparison (5 vs 33; Supplementary Fig. 24), and it is also more consistent with phylogenies of the SELMA proteins involved in protein translocation from the cytoplasm to the plastid[45]. More generally, our advanced phylogenetic analyses and broad taxon sampling continue to recover—albeit with only moderate support—the monophyly of all red algal-derived plastids, unlike in a recent proposal that argues for a separate secondary plastid acquisition in ochrophytes[46]. The origin of the ochrophyte plastid is a pressing issue to address in the future.

Ultimately, deciding between the origin of the leptophyte plastid and order of transfer to or from cryptophytes and haptophytes will rest on characterising the nuclear and putative nucleomorph genomes of leptophytes, ideally in the context of cultivation, in order to deter-mine their specific position in the eukaryotic tree and investigate their cell biology. This will be challenging, as our data suggest that lepto-phytes are generally rare and it remains unclear if they occur in easily accessible coastal regions. Despite the fundamental importance of algae with complex red plastid as marine primary producers, we still understand little of how this great diversity of algae came to be. Our data show that players still hidden in the rare biosphere hold important clues that could bring new critical evidence. We demonstrated that organellar genome-resolved metagenomics allows access to this hid-den diversity and should be considered in the future as a valuable approach to survey not only plastids but also mitochondria across biomes.

## Methods

### *Tara* Oceans metagenomes

The 937 metagenomes from *Tara* Oceans used in the study are publicly available at the EBI under project PRJEB402.

### Phylogeny-guided genome-resolved metagenomics for plastids

We used two complementary datasets (metagenomic assemblies and evolutionary-informative proteins) to characterise ptMAGs. On the one side, we used 11 large metagenomic co-assemblies from *Tara* Oceans (~12 million contigs from 798 eukaryote-enriched metagenomes) which were organised into 2550 metabins using constrained automatic binning and processed with anvi'o[20,21] v7 for manual binning purposes[16]. On the other side, we used proteins corresponding to the DNA-dependent RNA polymerase B subunit (RNApolB) and found in those *Tara* Oceans metabins, in the context of a phylogeny of representative sequences (amino acid level with sequence similarity <90%)[47]. Until now, the two datasets had only been used to characterise genomes corresponding to RNApolB clades of giant viruses and mirusviruses. Here, we focused on a large plastid RNApolB clade most closely related to that of Cyanobacteria and used this signal as guidance for genome-resolved metagenomics using the anvi'o interactive interface. Briefly, we mainly used sequence composition and differential coverage across metagenomes of the corresponding co-assembly to characterise and manually curate 1448 ptMAGs in the corresponding metabins. The strong correlation between initial signal (number of plastid RNApolB genes) and outcome (number of characterised ptMAGs) across metabins (Supplementary Fig. 1) demonstrated the effectiveness of this method to survey the environmental genomics of plastids in complex marine metagenomes. Preliminary phylogenies (see sections below) with 32 genes indicated a deep-branching lineage (the leptophytes) composed of three ptMAGs.

### Recovery of additional ptMAGs from the mOTU database

To broaden the scope of our survey, we exploited 85,123 metagenomic assemblies from the mOTU global metagenomic resource covering a wide range of environmental samples[18]. A total of 1,969,342 RNApolA genes were previously characterised from this resource[48]. Note that RNApolA displays a very similar evolutionary signal compared to RNApolB[49]. Here, we identified 6954 contigs >50 kbp that contained a RNApolA gene with relatively high sequence similarity to that of our characterised ptMAGs (DIAMOND blast, with percent identify >70% at the amino acid level). Based on preliminary phylogenies, we found one contig characterised from a *Tara* Oceans metagenome and corresponding to a leptophyte (contig ID Lepto-01_REFM_CHLORO_00001 with a length of 104,203 nt). In addition, we also found one contig (also characterised from a *Tara* Oceans metagenome) and corresponding to a deep-branching clade of haptophytes (contig ID REFM_CHLORO_00002 with a length of 84,869 nt). We integrated these two contigs into our database of ptMAGs.

### Creation of a non-redundant plastid genomic database

We collected all the manually curated *Tara* Oceans ptMAGs as well as the two ptMAGs from mOTUS-db and included 166 manually selected reference plastid genomes from cultured organisms. We determined the average nucleotide identity (ANI) of each pair of genomes using skani[50] v0.2.1. Genomes were considered redundant when their ANI was >98% (minimum alignment of >25% of the smaller genome in each comparison). For genomes found to be redundant, reference chloroplast genomes were selected over ptMAGs, and in most cases where only MAGs were redundant, the longest one was selected. We estimated the completeness and redundancy of each genome based on the occurrence of 44 core-plastid genes[51] (Supplementary Table 3), and removed ptMAGs with a redundancy level higher than 15%. This analysis provided a non-redundant database containing the 166 reference plastid genomes and 660 ptMAGs.

To assess potential mitochondrial contamination, all ptMAGs were screened for the presence of 25 canonical mitochondrial genes (Supplementary Table 4), as listed in ref. 52. Five non-redundant ptMAGs were found to contain genes annotated as typical mitochondrial genes, which were confirmed by BLAST searches. Further manual inspection indicated that these instances of contamination were not due to chimeric misassemblies. Instead, they resulted from inadvertent co-binning of mitochondrial and plastid contigs. The contaminated contigs were removed from their respective ptMAGs.

### Biogeography of the GOEV database

We performed a mapping of 937 metagenomes from *Tara* Oceans to calculate the mean coverage (vertical coverage) and detection (horizontal coverage) of each genome in the non-redundant plastid genomic database. Briefly, we used BWA v0.7.15 (minimum identity of 95%) and a FASTA file containing all contigs from the database to recruit short reads from each metagenome[53]. We considered a genome to be detected in a metagenome when >25% of its length was covered by reads. The number of recruited reads below this cut-off was set to 0 before determining the mean coverage of genomes, as an effort to minimise non-specific read recruitments.

### Genome annotation of ptMAGs

Plastid annotation was done with MFannot[54] v1.3.6 using the genetic code 11. The resulting ASN format files were converted to GenBank format using the *asn2gb* utility (https://ftp.ncbi.nlm.nih.gov/asn1-converters/by_program/asn2gb/) from the NCBI toolkit. We then extracted all annotated proteins from the GenBank files using the script *gb_to_prot.py* (available at https://github.com/burki-lab/ptMAGs/blob/main/src/gb_to_prot.py). For the leptophyte ptMAGs, intron prediction was carried out using RNAweasel (https://megasun.bch.umontreal.ca/apps/rnaweasel/), the 5S rRNA gene was detected with Infernal Cmscan (https://www.ebi.ac.uk/jdispatcher/rna/infernal_cmscan), and maps were generated using OGDraw version 1.3.1 (https://chlorobox.mpimp-golm.mpg.de/OGDraw.html)[55]. Downstream analyses with OrthoFinder (see **Comparative analysis of plastid gene content in leptophytes and related lineages**) revealed that MFannot did not detect the *rpl22* gene, labelling it as a "hypothetical protein" instead. The annotations of the leptophyte ptMAGs were manually updated to include this gene.

### Taxonomic annotation of ptMAGs

We manually determined the taxonomy of ptMAGs based on the global phylogenetic analysis (see section **Phylogenomic analyses of plastid dataset**), using guidance from the reference plastid genomes.

### Phylogenomic plastid dataset assembly

We began by using the two plastid gene sets originally defined by Janouškovec et al 2010[56]: a larger set of 68 plastid genes, and a more conserved subset of 34 genes. These served as the foundation for constructing preliminary phylogenies. During dataset assembly (see below), two genes were excluded: *acsF* due to low taxon occupancy, and *psbH* due to anomalously long branches in its single-gene tree. This reduced the gene sets to 32 and 66 genes, respectively. The resulting preliminary phylogenies identified leptophytes as a novel clade, but could not confidently resolve their position using maximum-likelihood analyses (Supplementary Figs. 25–26). Therefore, we attempted to increase the phylogenetic signal, by including additional genes used in Ponce-Toledo et al 2017[57] and Pietluch et al 2024[46], provided they were also present in leptophytes, resulting in a final set of 93 genes (Supplementary Data 4).

We first built a dataset of 93 protein-coding genes from the 179 reference taxa (166 plastid genomes and 13 cyanobacterial genomes). This was done as follows: (1) Homologous sequences from all reference taxa were retrieved with BLASTP[58] v2.15.0 searches using sequences

from *Fucus vesiculosus* (which has a well-annotated plastid genome; NC_016735) as queries (e-value: 1e-02), (2) For each gene, sequences were aligned with MAFFT[59] v.7.407 using the -auto option, and then trimmed with trimAL[60] v1.4.1 using a gap threshold of 0.8, (3) Single-gene trees were inferred with raxml-ng[61] v.1.2.0 using the LG4X model and 100 rapid bootstrap searches, (4) Gene trees were manually parsed to identify duplicates and extremely long branches corresponding to sequences from another gene, which were then removed from the dataset.

We then proceeded to construct a dataset initially including the 1448 ptMAGs identified (including redundant ptMAGs). Using reference sequences from the previous step as queries, we retrieved homologous sequences from the ptMAGs using BLASTP (e-value: 1e-02) and the gene annotation from MFannot. The 93 single-gene dataset was aligned individually using MAFFT-G-INS-I (using the --unalign 0.6 option to avoid over-alignment), and gently trimmed with trimAL (gap threshold: 0.9). Single-gene trees were inferred with IQ-TREE[62] v2.2.2.6 using the best fitting model as determined by ModelFinder Plus[63]. We manually parsed the single gene trees to check for putative chimeras and generated our final dataset comprising 93 genes and 839 taxa (660 ptMAGs and 179 references) with > 4% data.

The concatenated alignment was generated as follows: for each curated gene, we first used Prequal[64] v1.02 to filter sequence stretches with no clear homology using a posterior probability threshold of 0.95, then aligned with MAFFT-G-INS-I (--unalign 0.6), and subsequently trimmed with BMGE v1.12[35] (BLOSUM35 matrix, gap threshold: 0.8). After concatenating, we obtained a supermatrix (839-taxon dataset) with 19,242 aligned amino acid sites, which was used for broad-scale phylogenetic analyses. Finally, we also generated a smaller, 107-taxon dataset to enable analyses under more complex and computationally demanding models. For this dataset, raw sequences for the 93 genes were again filtered, aligned, trimmed and concatenated as previously described, yielding a supermatrix with 20,292 amino acid sites.

## Phylogenomic analyses of plastid datasets

**Site-homogeneous models**. To assess the phylogenetic diversity of the ptMAGs, the full 839-taxon supermatrix was used to compute a maximum-likelihood tree using the best-fitting site-homogeneous model, LG + F + I + G4 in IQ-TREE v.2.2.2.6. Branch support was assessed with 1000 ultrafast bootstrap replicates.

**Site-heterogeneous models**. We used the smaller, 107-taxon dataset to perform analyses with the more complex, site-heterogeneous models in order to determine the position of leptophytes in the plastid tree. An initial tree was inferred using the best-fitting cpREV+C60 + G4 model in IQ-TREE with 1000 ultrafast bootstrap replicates. This phylogeny was used as a guide tree to compute custom site-profiles directly from the input alignment using the programme MEOW[31] (-ri -p R -C 5 -f H -l; custom version of MEOW used is provided at https://github.com/burki-lab/ptMAGs/blob/main/src/meow_custom.R), an approach which often outperforms generic mixture models (such as the C-series)[65]. MEOW (MAMMaL Extension On Whole alignments) extends the MAMMaL approach by using all variable sites in the alignment to estimate custom site-profile classes. Following Williamson et al.[31], we estimated three sets of profile classes using varying proportions of high- and low rate classes respectively: (1) MEOW(80,0) indicates 80 classes from high rate sites and none from low rate sites, (2) MEOW(60,20) represents 60 classes from high rate sites and 20 classes from low rate sites, and (3) MEOW(40,40) corresponds to 40 classes each from the two partitions. We inferred a maximum likelihood phylogeny under the best fitting model based on the BIC criterion, LG + MEOW(60,20) + G4, with 1000 UFBOOT replicates, and for brevity, we refer to this model as LG + MEOW80 + G throughout the manuscript.

Another Maximum-likelihood phylogeny was inferred with the 107-taxon dataset with the CAT-PMSF approach[30]. Briefly, a phylogeny inferred under the site-homogeneous LG + G model in IQ-TREE was used as a fixed tree for analysis with the CAT + GTR + G model in PhyloBayes-MPI[66] v1.8. Two Markov Chain Monte Carlo (MCMC) chains were run for more than 5000 cycles until the effective sample sizes of nearly all parameters (seven out of eight) were above 100 (Supplementary Table 5). The posterior mean exchangeabilities and the custom site-frequency profiles were extracted for analysis in IQ-TREE and support was assessed with 100 non-parametric bootstrap replicates.

Finally, we also analysed the 107-taxon dataset in PhyloBayes under the CAT + GTR + G model via three independent MCMC chains. The chains were run for almost 9000 cycles each. After discarding the first 2000 cycles as burn-in, convergence was assessed (with maxdiff reaching 0.047) and consensus trees were generated using the bpcomp command.

**Derivatives of 107-taxon dataset**. First, we tested the impact of fast-evolving taxa in our dataset by removing 10 taxa (all ochrophytes) with the largest root to tip distances, generating a 97-taxon dataset. Second, we dealt with possible compositional heterogeneity across taxa by using BMGE's stationary-based trimmer to remove sites based on Stuart's test of marginal homogeneity (BMGE -s FAST -h 0:1 -g 1; 4752 out of 20,292 sites = 23.4% sites removed). Finally, we dealt with possible compositional heterogeneity and sequence saturation by recoding the dataset into SR4 categories[34]. Both the non-recoded datasets were analysed in IQ-TREE with the LG + MEOW80 + G model with 1000 ultrafast bootstraps. Additionally, the stationary-trimmed alignment was analysed in PhyloBayes using the CAT + GTR + G model, with three MCMC chains runs for 7000 cycles after a burn-in of 5000 cycles, achieving a maxdiff of 0.104. The SR4 recoded dataset was also analysed under the CAT + GTR + G model in PhyloBayes, using three MCMC chains run for 22,000 cycles, with a burn-in of 4000, reaching a maxdiff of 0.101.

## Assessing alternative topologies

**Site-heterogeneous models**. To assess alternate positions of leptophytes, we performed constrained tree searches in IQ-TREE. We considered three possible positions in the algal tree: (1) sister to haptophytes and cryptophytes (HC-sister), (2) sister to haptophytes (H-sister), and (3) sister to cryptophytes (C-sister). For each of these three positions, we considered two topologies; one in which the complex plastids are monophyletic, and the other where the complex plastids are non-monophyletic (as recently recovered by ref. 46), bringing the total number of tested topologies to six. Constrained topologies were inferred with IQ-TREE under the GTR + CAT-PMSF + G and LG + MEOW80 + G models for the 107-taxon dataset, and the xmC60SR4 model for the SR4 recoded dataset, with all parameters optimised freely. In the case of the LG + MEOW80 + G model, ten independent runs in IQ-TREE were conducted for each constraint to avoid likelihood scores from searches stuck in local optima[67]. In all cases, the topologies were evaluated with the Approximately Unbiased (AU)[68] test implemented in IQ-TREE. Additionally, the best scoring topology under each model was compared with alternative topologies using the Bonferroni-corrected chi-squared test[31,32]. Briefly, the likelihood ratio test statistic (LRS; two times the difference between the maximised log-likelihood of the two topologies under consideration) is first calculated. The p-value for the chi-squared test (p) is calculated as the probability that a chi-squared random variable would be larger than the LRS with the degrees of freedom defined as the number of branches collapsed in the strict consensus tree of the two trees. The Bonferroni-corrected p-value is calculated as $(1- \{1 - p\}^A)$ where $A$ is the set of trees compatible with the strict consensus tree.

**GHOST and GF-mix models.** We additionally tested the six alternative topologies under GHOST[33] and GFmix[31,32] as additional parameters to the LG + MEOW80 base model. For GHOST, the log-likelihood scores of the six topologies (inferred under the LG + MEOW80 + G model) were calculated under the LG + MEOW80 + H8 model in IQ-TREE, using the topologies as fixed trees. The LG + MEOW80 + H8 indicates a model with eight linked heterotachy classes and was the best fit model using the BIC criterion when compared to models with four or six heterotachy classes. Topologies were tested using the AU test and the Bonferroni-corrected chi-squared test as described in the previous section.

The GFmix model models compositional heterogeneity by modifying the vector of amino acid frequencies in a branch-specific manner to account for shifts in the relative frequencies of amino acids in different branches of the tree. While prior studies using this model have divided their alignment into multiple partitions to account for varying compositional heterogeneity in mitochondrial and nuclear encoded genes (e.g., refs. 31,32), we opted to use a single partition. This is because we do not expect substantial discrepancy in compositional heterogeneity between the only plastid-encoded genes considered in this study. With no a priori knowledge of how taxa might be compositionally biased, we first estimated the groups of amino acids that are depleted and enriched in different lineages from our data as done in ref. 31. Briefly, we used a custom R script (C. McCarthy, Dalhousie University, Canada, available at https://github.com/burki-lab/ptMAGs/blob/main/src/quickBinomial.R) to carry out the following steps: (1) tabulate the counts of all 20 amino acids for each taxon, (2) perform a chi-squared test on these counts and use the Pearson's residuals to construct a UPGMA tree of taxa, (3) separate taxa into two groups based on the UPGMA tree, (4) conduct a binomial test for each amino acid and calculate a score reflecting the differences in counts between the two groups, and (5) classify amino acids into "enriched," "depleted," or "other" categories based on their scores (Supplementary Note 1). The enriched and depleted classes of amino acids were used by GFmix v1.2 to compute the log-likelihoods of the six alternative topologies, which were inferred using the LG + MEOW80 + G model and provided as fixed input trees.

### 16S rDNA phylogeny
Barrnap v0.9 (https://github.com/tseemann/barrnap) was used to detect 859 bp and 808 bp fragments of the 16S rDNA gene in the Lepto-01 and Lepto-04 ptMAGs respectively. A BLAST search of the Lepto-01 16S rDNA sequence revealed that it was most similar to the DPL2[13] group with ~91% sequence similarity (accession numbers EF574856 and KX935025). However, the Lepto-04 16S rDNA sequence best BLAST hits included DPL2 as well as cyanobacteria. Manually BLASTing fragments of the Lepto-04 16S rDNA sequence against nt indicated that it was likely chimeric, with the first ~440 bp corresponding to DPL2 (100% similarity), and the latter half originating from cyanobacteria. We therefore trimmed the Lepto-04 16S rDNA sequence to only the first 440 bp. To confirm the association of leptophytes with DPL2, we constructed a 16S rDNA dataset including Lepto-01, Lepto-04, DPL2, and representatives of each eukaryotic algal clade. The 65-taxon dataset was aligned with MAFFT-L-INS-I, and trimmed with trimAL (-gt 0.1) to get an alignment with 1496 sites. A Maximum-Likelihood phylogeny was inferred with raxml-ng with 20 ML searches and non-parametric bootstrapping until convergence using the GTR + G model.

### Rpl36 alignment and phylogeny
To investigate whether leptophytes possessed the *rpl36*-p type or *rpl36*-c type gene[42], we extracted the L36 amino acid sequences from the 107 taxa in the subset phylogeny when available. The sequences were aligned with MAFFT-L-INS-I, trailing ends trimmed manually, and the alignment was visualised alongside the 93-gene PhyloBayes phylogeny using the R package ggtree[69] v3.6.2.

### Synteny analysis of leptophyte ptMAGs
Synteny analysis of the leptophyte ptMAGs was performed using the GenBank files generated during genome annotation using PyGenomeViz v1.0.0 via the PyGenomeViz Streamlit Web Application (https://pygenomeviz.streamlit.app/). Sequence similarity of protein coding genes was estimated using MMseqs2[70] v15-6f452 for reciprocal best-hit CDS search.

### Comparative analysis of plastid gene content in leptophytes and related lineages
To gain insights into the evolutionary dynamics and functional capabilities of the leptophyte plastid, we compared its gene content with that of red algae and other red algal-derived plastids. As ptMAGs can have missing genes due to incompleteness, we opted to use only reference plastid genomes where possible. For this analysis, we selected 15 rhodophyte, seven haptophyte, eight cryptophyte and 21 ochrophyte reference plastomes from our plastid genomic database. Similarly, we used the Lepto-01 ptMAG, the only near-complete leptophyte ptMAG, as the sole representative of leptophytes. We extracted amino-acid sequences from all selected plastid genomes and used OrthoFinder[71] v2.5.5 with default settings to detect homologous proteins. This analysis yielded 281 phylogenetic hierarchical orthogroups (HOGs) containing 7,648 protein-coding genes (98.5% of the total). However, as no orthology-predictor is perfect, we manually refined the HOGs by inspecting gene annotations, single-gene trees, and performing BLAST and InterProScan searches (Supplementary Data 5). This step yielded 237 HOGs containing sequences from at least three taxa. Heatmaps depicting gene presence/absence were plotted using the R package pheatmap v1.0.12 (https://github.com/raivokolde/pheatmap), and species and orthologs clustered using UPGMA clustering.

### Leptophyte associations with environmental parameters
We assessed correlations between leptophyte abundance and eight physiochemical parameters: sea surface temperature, salinity, dissolved silica, nitrate, phosphate, iron, and seasonality indices of nitrate and sea surface temperature. These parameters were obtained from Delmont et al 2022[16], representing data pulled from climatology and biogeochemical modelling data (World Ocean Atlas 2013 and PISCES v2)[72,73] to account for missing physio-chemical samples in the *Tara* Oceans in-situ dataset. Seasonality indices were defined as the range of the nitrate and temperature in one grid cell divided by the total range of that variable across all *Tara* sampling stations.

We calculated Pearson's correlation coefficients between the abundance of each leptophyte ptMAG and the environmental parameters using the R package, corrgram v1.14.

### Survey for putative nucleomorphs of leptophytes
We searched for genomic signal corresponding to red-algal nucleomorphs associated with leptophytes across the 11 Tara Oceans metagenomic co-assemblies. Specifically, we surveyed the assembled RNApolB genes, following the protocol used to characterise ptMAGs but this time using publicly available cryptophyte nucleomorph RNApolB genes as a reference. With this approach, we did not identify any nucleomorph-affiliated RNApolB genes co-occurring with the ptMAGs of leptophytes.

### Survey for host nuclear sequences of leptophytes
**Survey for leptophyte psbO sequences.** We considered the *psbO* gene as a good candidate to attempt finding contigs corresponding to the nuclear genome of leptophytes, as it is encoded by all photosynthetic eukaryotes, usually, in a single copy, in their nuclear genome, and is absent from non-photosynthetic eukaryotes[36]. We identified *psbO* sequences in the Arctic metagenomic co-assembly (available at: https://www.genoscope.cns.fr/tara/; where Lepto-01 is abundant)

through an HMM search based on the PFAM accession PF01716, carried out using HMMER v3.4 as implemented in anvi'o v8. A total of 51 hits were detected which were added to 346 reference *psbO* sequences from different algal groups. Sequences were aligned (MAFFT-L-INS-I), trimmed (trimAL, -gt 0.1), and then used to infer a tree with raxml-ng (20 searches, non-parametric bootstrapping until convergence) to determine the phylogenetic affiliation of the 51 *psbO* sequences. However, all *psbO* candidates fell within established algal groups, and no putative *psbO* sequence corresponding to leptophytes could be identified. We hypothesised that this result could be due to low coverage of the leptophyte nuclear genome. We investigated this by focusing on the two filters (both belonging to station 194) where Lepto-01 was among the most abundant eukaryotic plastids (124× and 70× coverage; Supplementary Data 1). We extracted the top 50 most abundant *psbO* sequences in these two filters from the recently assembled *psbO* database[36]. We generated an alignment and inferred a phylogeny as before, but again, did not identify any strong candidates for leptophytes. Most *psbO* sequences had only a small number of metagenomic reads mapping to them per sample (median number of mapped reads = 3), except for those from highly abundant chlorophytes and ochrophytes. This suggests that nuclear genomes generally have substantially lower coverage than the plastid genomes, making it challenging to retrieve nuclear genomic sequences from eukaryotes other than the most abundant ones.

**Survey for leptophyte 18S rDNA sequences**. We reasoned that a putative leptophyte 18S rDNA sequence would branch outside known algal lineages, and would correlate in abundance with the ptMAG abundance. We started with a targeted approach by generating a co-assembly of the six samples where Lepto-01 was most abundant using Megahit[74] v1.2.9. We then extracted 18S sequences from the co-assembly using Barrnap (length cutoff: 0.2) with the aim of getting a small, manageable pool of candidate sequences for further investigation. These 18S sequences were BLASTed (e-value: 1e-10) against PR2[75] v5.0.0 and the nt database to identify them, however, all sequences were linked to well established clades and no candidate leptophyte 18S sequence could be identified.

We then turned to metabarcoding data from the top 15 filters where Lepto-01 was most abundant[76]. We attempted to match the 18S-V9 ASVs (*n* = 5336) to Lepto-01 based on correlation of abundances following the workflow established by Zavadska et al 2024[77]. Briefly, we estimated a correlation coefficient (Spearman's correlation) between ASV relative abundance and the relative, and absolute abundance of the Lepto-01 ptMAG. We identified 13 and seven candidate ASVs, respectively, that showed a Spearman's correlation coefficient greater than 0.7. These ASVs were further analysed by performing BLAST searches against the PR2 database and nt database. However, none of the candidates matched leptophytes and the top hits were to cercozoan sequences in each case. This may be due to variation in plastid genome copy number under different conditions, leading to an inconsistent ratio between plastid and nuclear genome abundance.

**Survey for leptophyte mitochondrial sequences**
**Extracting mitochondrial contigs from targeted samples.** We first identified core mitochondrial marker genes by functionally annotating a set of seven publicly available haptophyte and cryptist mitochondrial genomes in anvi'o (*anvi-run-ncbi-cogs*). The following mitogenomes were used: NC_005332.1 (*Gephyrocapsa huxleyi*), OL703630.1 (*Gephyrocapsa oceanica*), OL703631.1 (*Gephyrocapsa muellerae*), AB930144.1 (*Chrysochromulina* sp. NIES-1333), AF288090.1 (*Rhodomonas salina*), NC_010637.1 (*Hemiselmis andersenii*), and NC_031832.1 (*Palpitomonas bilix*). This step revealed 13 core genes present in these mitochondrial genomes that were functionally annotated as: Cytochrome *c* oxidase subunit III, Cytochrome *c* and quinol oxidase polypeptide I, Cytochrome *c* oxidase subunit II (periplasmic domain),

Cytochrome $b/b_6$, Proton-conducting membrane transporter, NADH-ubiquinone/plastoquinone oxidoreductase chain 4 L, NADH-ubiquinone/plastoquinone oxidoreductase chain 6, NADH-ubiquinone/plastoquinone oxidoreductase chain 3, ATP synthase subunit α, ATP synthase subunit c, Ribosomal protein L16p/L10e, Ribosomal protein S12/S23, ATP synthase $F_O$ subunit.

We then turned to the co-assembly of the six filters where Lepto-01 was most abundant and extracted all mitochondrial contigs that were at least 5 kbp long. To do so, we first searched for contigs containing the *cox3* gene which encodes the cytochrome c oxidase subunit III enzyme, using an HMM search (based on the PFAM accession PF00510) in anvi'o v8, and functionally annotated the contigs as before (*anvi-run-ncbi-cogs*). We then selected contigs containing at least six of the 13 core genes present in the reference genomes, which yielded a total of 34 contigs representing partial mitochondrial genomes or mtMAGs (Supplementary Data 3).

**Abundance correlation analyses.** We expected the putative mtMAG of Lepto-01 to show a correlated abundance pattern with the Lepto-01 ptMAG. To test this, we mapped all 34 mtMAGs against 937 metagenomes from *Tara* Oceans, using the same approach applied to the plastid genomes (see Biogeography of the GOEV database. Abundance between each mtMAG and the Lepto-01 ptMAG was compared and plotted in R using the ggplot2[78] v3.5.1 and ggpmisc[79] 0.6.0 packages. Abundance correlations were tested using two ways (1) applying a detection threshold where at least 25% of a genome's length had to be covered by reads for it to be considered present in a metagenome, and (2) without applying this threshold. Both approaches produced highly consistent results, thus, only results from the analyses without the coverage threshold are presented here.

**Phylogenetic inference.** We reasoned that a putative leptophyte mitochondrial genome would be expected to form a distinct branch from established clades in a mitochondrial phylogeny. To assemble a mitochondrial phylogenetic dataset, we used the publicly available dataset of Williamson et al 2025[31] with 93 protein-coding genes and 100 taxa as a starting point. We subset this dataset to retain 50 eukaryotic taxa and the genes present in their corresponding mitochondrial genomes (40 protein-coding genes). To this dataset, we added our 34 mtMAGs and 18 additional reference genomes from various sources to increase the taxon sampling of cryptophytes and haptophytes as well as under-represented lineages such as Picozoa (Supplementary Data 6). This was done as follows: (1) the tool Codetta[80] v2.0 was used to determine the genetic code of all mtMAGs and additional reference genomes, (2) mtMAGs and additional reference genomes were annotated MFannot using the standard or mould mitochondrial genetic code as appropriate, (3) The resulting ASN files were converted to GenBank format using the *asn2gb* tool, and all amino acid sequences were extracted using the custom script *gb_to_prot.py* (available at https://github.com/burki-lab/ptMAGs/blob/main/src/gb_to_prot.py), (4) Homologous sequences from mtMAGs and additional references were retrieved by BLASTP searches using the 40 genes as queries, (5) Sequences for each gene were aligned using MAFFT-L-INS-I, and trimmed with trimAL using a gap threshold of 0.8, (6) Single-gene-trees were inferred with IQ-TREE using the best-fitting model, (7) Gene trees manually parsed to check for problematic sequences. 12 genes were excluded from the dataset at this point due to low taxon occupancy, and the final dataset comprised 28 protein-coding genes (Supplementary Table 6).

To generate the concatenated alignment, we aligned each gene with MAFFT-G-INS-I (--unalign 0.6), and gently trimmed alignments with BMGE (BLOSUM30 matrix, gap threshold: 0.8). We concatenated the genes to obtain an alignment with 102 taxa, 28 genes, and 6,302 sites. The dataset was used for ML analysis in IQ-TREE using the site-heterogeneous LG + C60 + G model with 1000 ultrafast

bootstraps. Additionally, we analysed the dataset in a Bayesian framework with PhyloBayes under the CAT + GTR + G model. Three independent MCMC chains were run for 15,000 cycles each, with a burn-in of 4500 cycles, and achieving a maxdiff value of 0.23.

## Reporting summary

Further information on research design is available in the Nature Portfolio Reporting Summary linked to this article.

## Data availability

The 937 metagenomes from *Tara* Oceans used in the study are publicly available at the EBI under project PRJEB402. Data our study generated has been deposited in an online repository: https://doi.org/10.17044/scilifelab.28212173[81]. This link provides access to the individual FASTA files from each plastid and mitochondrial genome used in our study (including the 660 non-redundant ptMAGs and 34 mtMAGs), the co-assembly of the top six samples where Lepto-01 was most abundant, individual gene alignments, concatenated and trimmed alignments, and maximum-likelihood and Bayesian tree files for the phylogenomic dataset. Source Data for Fig. 4, Supplementary Figs. 10-12, and Supplementary Fig. 22 can be found on the linked GitHub repository, while source data for Supplementary Figs. 2-6 is provided as Supplementary Data 1.

## Code availability

All scripts used for genome annotation and phylogenetic analyses are available on GitHub: https://github.com/burki-lab/ptMAGs with the identifier: 10.5281/zenodo.17635604[82].

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

## Acknowledgements

We thank Shinichi Sunagawa for having facilitated the recovery of relevant data from the mOTU metagenomic database maintained by his research group at the Department of Biology at ETH Zürich. We thank A. Roger, H. Baños, and C. McCarthey for discussions, and for kindly providing custom scripts for running the phylogenetic models MEOW and GF-MIX. We thank J.E. Dharamshi for discussions about phylogenetic analyses. Our survey was made possible by the sampling and sequencing efforts of the *Tara* Oceans Project. *Tara* Oceans (which includes the *Tara* Oceans and *Tara* Oceans Polar Circle expeditions) would not exist without the leadership of the *Tara* Oceans Foundation and the continuous support of 23 institutes (https://oceans.taraexpeditions.org/). This article is contribution number 164 of Tara Oceans. Phylogenetic analyses were enabled by resources provided by the National Academic Infrastructure for Supercomputing in Sweden (NAISS 2024/5-197), partially funded by the Swedish Research Council through grant agreement no. 2022-06725. M.J. was supported by the Swedish Research Council (International Postdoc grant 2022-00351). FB's research is supported by grants from the European Research Council (ERC consolidator grant 101044505), the Swedish Research Council VR (2021-04055), and Science for Life Laboratory. TD's research is supported by a grant from the l'Agence Nationale de la Recherche (ANR-23-CE02-0022). We also thank the commitment of the CNRS and Genoscope/CEA. Some of the computations were performed using the platine, titane and curie HPC machine provided through GENCI grants (t2011076389, t2012076389, t2013036389, t2014036389, t2015036389 and t2016036389).

## Author contributions

F.B. and T.O.D. conceived the project. T.O.D. characterised the ptMAGs. M.J., F.B and T.H., performed phylogenetic analyses. T.A., E.P. and T.O.D. created the plastid genomic database and performed surveys for nucleomorphs. H.J.R. and L.P. retrieved relevant data from mOTUs. M.J. and T.H. annotated the ptMAGs, and E.P. performed mapping analyses. F.B., M.J., and T.O.D. wrote the manuscript with input from all the authors.

## Funding

## Competing interests

The authors declare no competing interests.
