## [Transparent Peer Review file · Nature Communications]

Identification of a deep-branching lineage of algae using environmental plastid genomes

Corresponding Author: Dr Fabien Burki

Version 1:

Reviewer comments:

Reviewer #1

(Remarks to the Author)

Utilizing the vast data generated by Tara Oceans expeditions, the authors amassed ~700 non-redundant plastid genomes that are manually curated. They have found plastid genomes from the "Leptophytes", which is a deep-branching algal lineage. Given their presence in the read data from nano-size fraction, the authors thereby named it as Leptophyte. Thorough phylogenetic analyses revealed that leptophytes forms clade with either haptophytes or cryptophytes, depending on the tree model.

Even though there are concerns about the novelty and credibility of this study which will be discussed in the following paragraphs, I would like to commend the authors for their efforts in handling and curating such a vast dataset. The scope and aim are clear and straightforward. The findings in this study must provide an important base for exploring the diversity of plastids. However, I'd like to raise two major concerns regarding this study: its originality and the credibility of its taxonomic and evolutionary interpretation.

First, the originality of this study raises concerns regarding its suitability for publication in Nature Communications. This lineage was previously described as DPL2 by Choi et al. (Current Biology, 2017). That study, which was also based on Tara Oceans data, documented the presence of this lineage in the photic zone and within the picoplankton size range. It also proposed a close phylogenetic relationship between DPL2 and haptophytes, with moderate support (posterior probability of 0.92). The current study does not provide significant new biological insights such as morphological or ecological information. Although the plastid genomes of the "Leptophytes" were recovered, their gene content appears largely similar with that of haptophytes (as shown in Figure 2), and no notably unique features are reported.

With respect to the taxonomy of this lineage, the authors should conduct a more thorough investigation into its relationship with haptophytes, using not only just DNA sequences in the metagenome data, but also data derived from the actual organisms. Given the phylogenetic placement, relatively short branch lengths, and the high similarity in plastid gene content with haptophytes (Figure 2), it is challenging to rule out the possibility that this group represents just another early-diverging lineage of haptophytes rather than a novel clade. Haptophytes are defined by unique features such as the presence of a haptonema and specific pigment compositions. Without additional morphological data (e.g., FISH to trace the cell), it is premature to propose "Leptophytes" as it implies the recognition of a novel (sub-)phylum level eukaryotic lineage different than haptophytes. Such a taxonomic designation would require far more robust and diverse evidence and I think it is currently impossible.

Although this study presents some novel plastid genome assemblies derived from publicly available sequence data, I think the manuscript falls short in terms of originality and lacks sufficient data for the thorough testing of evolutionary and phylogenetic hypotheses expected for publication in Nature Communications.

Line 205: Please provide a detailed description for the reason why the authors choosed 93 genes. What criteria were applied in choosing these specific genes? I am also curious about whether using an alternative or modified gene set might result in a different topology, or if the current topology is robust and consistent across various gene selections.

Reviewer #2

(Remarks to the Author)

The manuscript by Jamy et al. delivers very much awaited results of continued exploration of the Tara Oceans metagenomic data, now focusing on the representation of the diversity of plastid genomes in the data. The scientific merit of the study is obvious. Previous investigations utilizing as phylogenetic markers individual genes retrieved from environmental DNA samples have pointed to the existence of unknown deeply diverged plastid lineages or have documented the existence of novel uncultivated groups forming separate deep lineages within known major algal groups (green algae, ochrophytes, haptophytes), raising the question whether they also have a plastid and if yes, whether it is photosynthetic. The work by Jamy et al. progresses towards addressing these questions, although the concept of the study is admittedly not such as to aspire on provide comprehensive answers. The scope of the study is from the start restricted by focusing on metagenomic data from the sunlit ocean, so it cannot cover organisms that live in other types of marine habitats, let alone those thriving in freshwater ecosystems. Nevertheless, even focusing on one particular category of habitats have yielded important new findings that make the study by Jamy et al. highly valuable. In this regard it is interesting to compare this study with a recently released preprint by Shrestha et al. (bioRxiv, <https://doi.org/10.1101/2025.03.28.644651>), which reports on an exploration of plastid genomes in a much broader set of metagenomic samples (including the freshwater ones). The general conclusions of both independent studies are largely overlapping, including the possibly most exciting result of both studies, that is obtaining the first (nearly complete) plastome sequences from an uncultivated marine lineage denoted by Jamy et al. as “leptophytes” and embracing the previously known DPL2 lineage of plastidial 16S rRNA genes hypothesized to represent a new algal group. Thanks to the effort by both teams it is now clear there indeed exists a deeply diverged evolutionary lineage of plastids affiliated to the plastids from haptophytes and cryptophytes. Nevertheless, the identity of the actual organismal (i.e., host) lineage harbouring these plastids remains unknown, leaving a major gap in our knowledge of algal diversity open.

Thus, while I am generally very positive about the study by Jamy et al. and I believe it is in principle worth publishing in a top-tier journal like Nature Communications, I believe some extra work and improvements in the presentation of the data is necessary to more fully realize the potential of the material in hand. Below I provide a list of points that the authors should address. Despite their number they do not require an excessive amount of time or effort, and I hope they are all constructive, helping to make the paper even better than it is now.

Critical points regarding the methodology:

- line 94: two references are cited #16 and #19. The latter is a paper describing the development of a particular software (MarkerMAG) and it is not clear to me how this paper relates to what the authors mention in the respective sentence citing the paper. Is it possible that you wanted to cite something else here? Regardless, if the tool MarkerMAG was indeed used in the study, it should be clear how (it is not mentioned in the Methods section).
- The description of methods and procedures additionally misses a description of how the mOTUs-db was used to retrieve the two ptMAGs analysed by the authors. I have checked the mOTUs-db website it is not at all obvious how to proceed to retrieve any ptMAG from the database, so please add a few lines on this.
- line 101: “filtering for redundancy (average nucleotide identity <98%)” – the sentence does not make sense to me unless assuming that instead of “<98%” there should be “>98%” used in the text. Please check this.
- There seems to be some chaos regarding the number of reference plastid genomes used in the analyses by the authors. According to the information provided in line 102 and then later in the Methods section (lines 381 and 389), the authors selected 164 reference plastid genomes. These are supposed to be listed in Supplementary Table 1 (see line 103) and I assume these correspond to plastomes with identifiers starting with “REFG_”. However, there are 167 rather than 164 REFG_ genomes listed in the table. Furthermore, while there are 167 genomes listed, the identifiers go from REFG_CHLORO_00001 to REFG_CHLORO_00166. This is because the table includes two different plastomes labelled as “REFG_CHLORO_00165”, one corresponding to KY860574 (a cryptophyte plastome) and included in the list between REFG_CHLORO_00020 and REFG_CHLORO_00021, and the other corresponding to NC_020371 and listed at the expected position (after REFG_CHLORO_00164). I am not sure if such a labelling discrepancy might have affected the analyses reported by the authors, but they certainly have to check this carefully and also ensure each plastome has a unique identifier in the table. To make the confusion even worse, at other places in the manuscript (legend to Fig. 1, line 117; Methods – line 419 and 435) the authors mention the use of 180 reference plastid genomes. If two different sets of reference plastid genomes were indeed used in different analysis (but I guess this is possibly not true), then some clarification for the rationale is needed and a list of all the 180 references needs to be provided. Finally, still another number of reference plastid genomes – 166 – is mentioned in the legends to Supplementary Figs 2-5.
- I think a major omission in the description of the methodology employed by the authors concerns the procedure used to create Fig. 2 (and its more detailed version displayed as Supplementary Fig. 15). The authors mention the analysis is based on 355 “plastid-encoded genes” (is a gene really “encoded”?), but it is not specified how these genes were selected and, most crucially, how the presence/absence of particular genes has been examined in individual genomes. Some plastid genes are rapidly evolving and it may be challenging to identify orthologs correctly. Furthermore, as is apparent from Supplementary Fig. 15 many of the 355 genes are labelled orfXXX (where XXX stands for a particular number), i.e. they do not correspond to broadly conserved and well-annotated plastid genes. Please, provide a description how these genes were delimited and how orthology across the plastid genomes analysed was assessed. We need to be certain the presence/absence pattern of these genes as displayed in the scheme does not reflect simply the occurrence of the same “orfXXX” annotated as such by MFannot in different plastid genomes – note that the same label frequently means only the same length of the orfs, i.e. the same number of codons, not orthology as such. I think that for the sake of reproducibility, the authors should provide a table listing all the genes for all the plastomes so that an interested reader could unambiguously identify each gene (at the sequence level); these data are not provided even at the Figshare site linked to the manuscript.

- The authors assessed the completeness of their ptMAGs by checking the occurrence of 44 core-plastid genes listed in Supplementary Table 5. A large proportion of these genes encode components of the photosynthetic apparatus, so these are expected to be missing from plastomes coming from non-photosynthetic plastids. It is not clear to me whether the authors have used the completeness as a criterion for filtering out any of the potential ptMAGs and if yes, whether this might systematically remove from their final ptMAG set the plastomes of non-photosynthetic plastids. Regardless, are there any ptMAGs in the final dataset that are likely representing non-photosynthetic taxa? I believe this is relevant question and an interesting aspect of plastid diversity ignored in the present version of the manuscript.

Analyses specifically concerning leptophytes:

- Plastid genomes typically include a duplicated region known as the inverted repeat, but surprisingly this term does not appear at all in the text despite the fact it is an important feature with direct bearings to the process of assembly of plastome sequences. It is particularly notable that the leptophyte ptMAG, although incomplete, implies a plastome organization lacking an inverted repeat (at least judging from Supplementary Fig. 13). I believe the authors should explicitly mention whether they see any evidence for inverted repeats being part of leptophyte plastomes (careful manual analysis of the assembly data including raw reads may be necessary to clarify this), and if not, how unusual this would be compared to plastomes of haptophytes and cryptophytes. In addition, I think the authors should explicitly comment on what the Lepto-01 ptMAG is most likely missing. Judging from the map it is possible that only the remaining parts of the 16S and 23S rRNA genes (rns and rnl) plus two tRNA genes (trnI and trnA) that are usually placed in between these two genes are missing. In fact, the annotation of the Lepto-01 ptMAG as presented by the authors in Supplementary Figs 13 and 14 (and the respective annotation file at the linked Figshare site) is incomplete and misses a 5S rRNA gene (rrn5) that is located exactly when one would expect in analogy with other plastomes – in the unannotated (“intergenic”) region just downstream the rnl gene; the presence of the gene is readily seen when the respective region is analysed using Infernal Cmscan (https://www.ebi.ac.uk/jdispatcher/na/infernal_cmscan). Hence, the authors should update the annotation of the ptMAG by adding this extra gene.

- The authors have attempted to illuminate the nature of the “host” lineage harbouring the leptophyte plastid. Specifically, in Results they write (lines 165-166): “However, efforts to identify potential hosts for Lepto-01 at stations where its signal was strongest were unsuccessful (see Methods)”. In Discussion they then add this (lines 335-340): “While we found ample evidence of cryptophyte nucleomorphs elsewhere, we found no evidence of putative leptophyte nucleomorphs (see Methods)”. The methods section then indeed includes a section describing specific analyses the authors carried out in this regard. While the outcome of their effort has been negative, I believe these analyses should be featured more prominently in the manuscript and require some clarifications as well as some extra effort. Regarding the attempts to identify candidate leptophyte nuclear psbO genes and potential nucleomorph “RNApolB” genes, one should bear in mind that these genes are expected to harbour intronic regions that disrupt the coding sequence. However, it is not clear from description of the search procedure employed by the authors whether and how they accounted for the possibility of the coding sequences of the target genes being interrupted by introns, which obviously complicates retrieval of the sufficiently complete amino acid sequences from the metagenomic data. Next, it should be explicitly mentioned that the authors were looking for “RNApolB” genes specifically related to cryptophyte nucleomorphs (as sequences affiliated to homologs from chlorarachniophyte nucleomorphs would not be good candidates for leptophyte nucleomorph genes). In fact, I am slightly confused by the authors using the term “RNApolB” genes. Eukaryotes generally have three different homologs of the single prokaryotic RNApolB protein, corresponding to subunits of RNA polymerase I, II and III. So which protein of those do the authors mean here?

- Regarding the search for candidate leptophyte 18S rRNA gene: my own experience with metagenome data analyses is such that particular 18S rRNA sequences are frequently missing from sequences assemblies despite the fact they are represented in the sequencing reads. Was the analysis of the V9 metabarcoding ASV sequences mentioned in lines 602-605 based directly on reads? Likely so, but this is not completely clear from the very brief description of the procedure. Regardless, I think it would make sense to dig a bit deeper, and above all to test directly the most likely possibilities or the hypotheses raised by the authors themselves. Specifically, the authors should directly check the reads from the metagenomic samples that contain leptophyte sequences for the presence of 18S rRNA signatures corresponding to the known uncultivated haptophyte-related lineages (presented in refs #42 and #43) or to the CRY3 lineage. Next, there is a recent preprint by Romero et al. (bioRxiv, <https://doi.org/10.1101/2025.03.26.645542>) reporting on a systematic analysis of 18S rRNA sequences in metagenomic data and reporting in numerous novel eukaryote lineages. One is wondering whether this study might help identify candidates for leptophyte 18S rRNA sequences. Finally, I believe it is likely that at least partial mitochondrial genome sequences from leptophytes should be present in samples containing their plastid genes, so I strongly encourage the authors to look for mitochondrial genome sequences co-occurring with the leptophyte plastome sequences and representing phylogenetically separate lineages placed in the global tree such that they at least potentially may correspond to leptophyte mitochondria. The leptophyte story is clearly central to the whole paper and this extra effort may make is stronger.

Additional comments to the analyses and their interpretation:

- I think the informativeness of the study would be further increased by elaborating a bit more on the identified ptMAGs other than those from leptophytes. For example, according to Supplementary Fig. 3 the authors have identified a few MAGs that are exceptionally large, in one case >600 kbp and according to the data presented in part b of the figure coming from a dinophyte. One wonders why the genome is so large (much larger than any reference dinophyte plastome) – is it because of an inflation of intragenic regions, and if yes, what kind of sequences makes this extra stuff? How certain the authors are the assembly is correct and does not include sequences that do not belong to the plastome?

- Another particular point I believe the authors should exert more effort to illuminate the nature of the clade of eight

chlorophyte ptMAGs that for a separate clade perhaps related to Chloropicophyceae (shown in detail in Supplementary Fig. 8a). Indeed, there are at least two deeply diverged “prasinophyte” clades known from environmental amplicons of the 18S rRNA gene but not yet represented by cultivated members, called Clade VIII and Clade IX by Viprey et al. (<https://enviromicro-journals.onlinelibrary.wiley.com/doi/10.1111/j.1462-2920.2008.01602.x>). One would speculate that the aforementioned ptMAG clade most likely comes from the same organisms as the 18S rRNA amplicons assigned to Clade VIII or Clade IX, and it would be interesting to test this hypothesis directly by looking for the respective 18S rRNA gene sequences in the metagenomic samples enriched for plastome sequences of the clade under question. Matching the ptMAG clade to a previously known 18S rRNA clade would be an important step forward in charting the green algal phylogenetic diversity. An analogous analysis could be done for some of the ochrophyte ptMAGs, such as those related to pelagophyte plastomes (Supplementary Fig. 8b) that occupy a position equivalent to some of the uncultivated MOCH (marine ochrophyte) lineages in the 18S rRNA tree (see Fig. 6 in Terpis et al. 2025, [https://www.cell.com/current-biology/abstract/S0960-9822\(24\)01632-4](https://www.cell.com/current-biology/abstract/S0960-9822(24)01632-4)).

- One of the general incentives behind exploring hitherto missing branches of the algal tree of life is to eventually resolve the puzzling evolutionary history of higher-order plastids, particularly those originating from red algae. One of the points that is controversial – or one would say somewhat unexpectedly became controversial only recently with the study by Pietluch et al. (2024; cited by Jamy et al. as ref. #55) – is whether the different “chromalveolate” plastids are monophyletic to the exclusion of extant red algal lineages or, as proposed by Pietluch et al., evolved diphyletically from different red algal ancestors. The improved sampling and especially the addition of leptophyte plastomes delivered by Jamy et al. creates a dataset that should be more informative on this contentious issue than any dataset analysed before. Another salient aspect of the analyses presented by Jamy et al. is the use of the highly sophisticated methodology of phylogenetic inference, including the most recently developed complex substitution models. Hence, one would think that the present study will say something significant regarding the origins of the complex red plastids. Thus, I encourage the authors to elaborate a bit more on the general implications of the new data for the reconstruction of the plastid history. Indeed, they have already considered the alternative scenario suggested by Peitluch et al. in their analyses, but what I am missing is a more explicit discussion on the results. Do their analyses illuminate the question of monophyly/non-monophyly of complex red plastids and if yes, which of the two possibilities is more credible? A reader would appreciate some discussion.

- The authors should double-check the ptMAGs they report in their study. As it became clear only today (thanks to a student of mine looking at the data), at least in one case the reconstructed ptMAG includes sequences that clearly do not belong to a plastid genome. Specifically, TARA_CHLORO_00069 contains a contig (CHL_AON_Bin_218_19_c_000000000001) that based on the annotation provided by the authors themselves corresponds to a part of a mitochondrial genome (not the presence of genes *atp1* and *cob* that have never been seen in plastomes but are typical for mitochondrial genomes). A systematic check of the whole set is needed to identify possible similar cases and the readily detectable contaminating sequences should be removed from the ptMAGs.

Various additional issues:

- line 56: should there be a comma before “such as diatoms”?

- line 187: instead of “retaining the core components” please write “retaining genes for the core components”, as what is retained in the plastid genomes are genes, not the actual components encoded by them.

- lines 188-189: in relation to the previous note, use italics when mentioning the various plastid genes. The same for “*rpl36*” in lines 212 and 310.

- line 226: “were less than four points difference” – I think the grammar is perhaps not correct here, please check

- line 232: “show in that order” – the phrase sounds strange to me, I would expect something like “are shown in the following order”, but I am not a native English speaker.

- line 235: delete the hyphen from “*rpl-36*” for the sake of consistency

- line 284: “c-type paralog of the ribosomal gene *rpl36*” – I have two issues here. First, while being aware that the term “paralog” was used already in the original report on the *rpl36* gene replacement by Rice and Palmer (2026), I don’t think he term is used properly in this case. By definition, paralogs are genes related via a gene duplication event, so to claim the c-type and p-type *rpl36* versions being paralogous one would have to provide evidence they are descendants of two different copies of a single duplicated gene, with the duplication event having happened somewhere at a deep branch of the eubacterial phylogeny. However, no such evidence has been provided (by Rice and Palmer or the authors of the present manuscript) and presently it is completely unknown how the c-type and p-type *rpl36* versions evolved. Hence, I would avoid using the term “paralogs” and use something not implying a specific evolutionary event, e.g. “forms” (check also line 326). Second, in my opinion *rpl36*, rather than being a “ribosomal gene”, is a “ribosomal protein gene”.

- line 301: “limited information of plastid genomes” – would it be more appropriate to write “limited information content of plastid genomes”?

- line 433: “IQTree” – I think this is not how usually the name of the program is spelled; compare also with a different (correct?) spelling, i.e. “IQ-TREE”, at other places in the manuscript.

- lines 510, 512, 524: there is a problem with correct printing of the Greek letter “chi” here. Note also that the manuscript is

inconsistent in that at other places the alternative writing “chi-squared” (e.g., line 239) is used. I would unify this across the whole text.

- line 545: “16S phylogeny” – I think it is inappropriate to omit “rRNA” or “rDNA” from the name of the locus used as the phylogenetic marker, so please, use “16S rRNA” or “16S rDNA” (the latter may be the best option, as it is the DNA sequence that is de facto used) in the whole text (including Supplementary Fig. 12). Analogously, write “18S rRNA” or “18S rDNA” where appropriate (line 597 and the whole following paragraph).

- line 562: “the *rpl36* amino acid sequences” – technically this is not correct. While there is a gene name “*rpl36*” (in italics), the respective protein product is called L36.

- line 583: “We identified HMM *psbO* sequences” – I don’t understand what is a “HMM *psbO* sequence”, perhaps delete “HMM” from the sentence

- update ref. 30 (Williamson et al.), the preprint has now been published in a journal (Nature)

- delete “CB” from the journal title in ref. #54 (Ponce-Toledo et al. 2017)

- fix the journal abbreviation in ref. #67 (Yu 2020), the last word correctly is “Bioinform.”

- Fig. 1: while I do see using names for various taxa is always to some extent arbitrary, I would still consider to improve the consistency of the nomenclature by changing some of the names. Thus, “Euglenids” is a name referring to a broad euglenozoan lineage that includes a paraphyletic assemblage of aplastid taxa plus a single plastid-bearing (i.e. algal) subclade, usually called Euglenophyceae. Hence, to stay with names of algal taxa when labelling the different branches of the plastid genome tree, I would write “euglenophytes” instead of “Euglenids”. It would also be more consistent with the overall nomenclatural style to write “Rhodophytes” instead of “Red algae” (in fact, “Rhodophyta” is used to label the respective taxa in Supplementary Fig. 7 showing the full version of the tree). Next, the labeling of a particular sector of the tree with “Chrysophytes + Brown algae” is somewhat imprecise, as this part of the tree includes plastid genomes from additional ochrophyte taxa, including raphidophytes or pinguiphytes (note that TARA_CHLORO_00423 in fact comes from a pinguiphyte, which is not obvious from the tree presented by the authors for their omission of pinguiphyte references). One way how to solve this would be to simply write “other ochrophytes” instead of listing all the constituent taxa here. Finally, the sector of the tree labelled “Diatoms” in fact includes also bolidophytes, and this should be reflected in the tree. I think it would be useful to add a note in the legend to the figure that a full version of the tree is provided in Supplementary Fig. 7 (in analogy with a remark at the end of the legend to Fig. 2).

- taxon names for some of the reference plastomes: I understand it is very difficult to keep track with the taxonomic updates regarding sequence data deposited in databases under outdated or inconsistently applied names, but still I think it’s important to try to be as accurate as possible. Thus, some of the taxonomic names of the reference plastomes included by the authors should be updated throughout the manuscript (including all tables and figures). Here are some examples I have spotted upon a quick inspection of the list:

REFG_CHLORO_00059 – what is referred as “*Chromerida* sp. RM11” has been described in 2012 as “*Vitrella brassicaformis*” (<https://www.sciencedirect.com/science/article/pii/S1434461011000939>)

REFG_CHLORO_00030 – what is referred as “*Cryptophyta* sp. CCMP2293” has been described as “*Bafinella frigidus*” (<https://onlinelibrary.wiley.com/doi/full/10.1111/jpy.12766>)

REFG_CHLORO_00136 – what is referred as “*Dictyocha speculum*” has been reclassified as “*Octactis speculum*” (<https://onlinelibrary.wiley.com/doi/full/10.1111/pre.12181>)

REFG_CHLORO_00009 – what is referred as “*Nannochloropsis gaditana*” has been reclassified as “*Microchloropsis gaditana*” (<https://www.tandfonline.com/doi/full/10.2216/15-60.1>)

REFG_CHLORO_00053 – what is referred as “*Pyconococcus provasolii*” should be named “*Pseudoscourfieldia marina*”, as the former is a junior synonym of the latter (<https://onlinelibrary.wiley.com/doi/full/10.1111/jpy.13482>)

REFG_CHLORO_00012 – is there a reason why the species name “*Prasinococcus capsulatus*” (<https://www.tandfonline.com/doi/full/10.1080/23802359.2019.1698370>) should not be applied to what the authors presently call “*Prasinococcus* sp. CCMP1194”?

REFG_CHLORO_00143 – the organism is referred to as “*Pavlova* sp.” (Supplementary Table 1, Supplementary Fig. 7) or as “*Pavlova* sp. NIVA-4/92” (Fig. 2 and several supplementary figure), but in fact it has been identified as the species *Diacronema lutheri* (<https://academic.oup.com/gbe/article/13/8/evab178/6337978>) as is thus taxonomically redundant with the second plastome from this species included in the dataset by Jamy et al. (NC_020371). Unsurprisingly, these two plastomes branch together in the trees presented in the paper, separate by a zero branch length.

REFG_CHLORO_00021 – while the correct name “*Pavlomulina ranunculiformis*” is used by the authors in most trees, the same organism is labeled as “*Haptophyceae* sp. NIES-3900” in Supplementary Fig. 7 and Supplementary Table 1, so this discrepancy should be fixed, too.

- Fig. 3: Frankly, I do not know how to understand the meaning of the blue bars in part c of the figure, which are explained in the legend as follows: "Difference in log-likelihood scores between each topology and the best-scoring topology is shown by data bars in blue". At face value the bar sizes do not really intuitively follow the numbers indicated in the table. Most likely I am missing something, but the authors could try to provide an explanation of the data that is easier to understand.

- Fig. 4: I am not sure why the cryptophyte branch in the left part of the figure and the haptophyte branch in the right part of the figure are drawn in colour along their whole length. I would think it makes sense to start colouring the branch from the point of plastid gain to the tip, otherwise the meaning of the branches in colour is elusive to me. The same concern holds to Supplementary Fig. 20.

Reviewer #3

(Remarks to the Author)

New deep-branching environmental plastid genomes on the algal tree of life

Mahwash Jamy¹, Thomas Huber², Thibault Antoine³, Hans-Joachim Ruscheweyh⁴, Lucas Paoli⁵, Eric Pelletier³, Tom O. Delmont^{3*}, Fabien Burki^{2*}

The authors have mined Tara Oceans data to assemble and curate MAGs corresponding to plastid genomes, with a focus on those derived by higher-order endosymbioses involving red algal donors. It is a remarkable paper. The authors do not over-state its conclusions. To my knowledge it is indeed the most extensive survey of plastid genomes in this way, and they uncover an incredible wealth of information, including discovery of a new algal lineage, which they reasonably assign the following informal name: leptophytes.

The Introduction accurately covers the history of this complex field and is well cited in this regard.

One could argue that, in the complete absence of any knowledge of a host for the leptophyte plastid, it is too soon to be proposing evolutionary scenarios about how this organelle came to have a genome that looks like it does. My first reading was exactly this. I was wanting more description of the host lineage search in the main text (not passed on to the Methods, which was done on two occasions). But having read the discussion to the end, I can see why the authors are comfortable floating two hypotheses, both with backbone trees that are clearly labeled as hypothetical. Any way one looks at these data, there are still a lot of gaps to fill in our knowledge; all we know for certain now is that this leptophyte plastid lineage clearly puts it in the haptophyte / cryptophyte part of the plastid tree (based on multiple lines of evidence), but it is nevertheless highly distinct. And so we can reasonably say that whatever the host is, it too is rather distinct from the nuclear component of these algae. And if not, well, that would be really interesting too.

To conclude my general comments, this paper has a level of novelty that deserves publication in a high-impact journal – it points to the existence of an entirely new branch of photosynthetic life. That is a rare and exciting find. I have the following specific suggestions for the authors to consider.

Specific comments:

-I suggest the authors add some 'landmark' taxa on Fig 1 tree. It is not easy to align the gray 'reference' OTUs on Ring 2 with the actual branches (which in most cases are quite far away). So why not provide a few well-known taxon names on the actual branches, e.g. *G. huxleyi*, *Phaeocystis*, *Thalassiosira*, *Guillardia*, *Aureococcus*, etc., where there is space. This would help experts pinpoint their favourite groups and see how much new diversity the new paper adds to that particular part of the tree. If done strategically, these additions would not detract from the larger take-home messages.

-Results – first paragraph. The authors discuss ANI and completeness, which makes sense, but it might be good to also include depth of MAG coverage. This is brought up later when discussing abundance of the leptophytes, but non-specialists (used to sequencing genomes 'normally' and not assembling them from metagenomic data) would appreciate getting a sense of the range of depth of sequence coverage in the diversity of ptMAGs.

-Figure 2. What is the tree topology on the left based on? The legend says a larger version is shown in Supp. Fig 15, but that figure looks to be the exact same as Figure 2 (except for presence of gene names) and doesn't say what the tree at left is based on either. The legend refers to UPGMA clustering, but which tree, the one on the left or the one on the top? These points need clarification.

Lines 135-137 – "In multiple cases, the ptMAGs represented novel genome diversity..." As written, this statement undersells the significance of the data and is not in line with the novelty conveyed in the abstract. Assuming that the gray reference sequences were chosen to represent the full breadth of sequence diversity from cultured representatives, then I would say that novel genome diversity is massively represented in the tree shown in Figure 1. I guess it depends on what is meant by 'novel diversity', but looking within haptophytes, diatoms and pelagophytes, the authors have recovered a tremendous number of novel genomes. I suggest they highlight this even more in the main text, to be more consistent with the (accurate) description in the abstract.

-Lines 165-66: "However, efforts to identify potential hosts for Lepto-01 at stations where its signal was strongest were unsuccessful (see Methods)." As noted above, I think the authors would do well to briefly integrate some of this information into the main text. The MS is very short (like an old Letter to Nature) and thus there should be plenty of room. And

furthermore, the authors could touch on why it matters what the host is, as a prelude to the Discussion.

-Line 196: present in all other algal groups shown, or all other algal groups (inc. green and glaucophyte algae)?

-Line 284: Paralog is not the right word to refer to the c-type rpl36 in the CHL plastid because it doesn't (obviously) involve gene duplication. Probably xenolog is better (which I don't see much and am not a fan of). I think the evidence suggests that rpl36 in CH (and now L) is most likely an example of orthologous replacement (i.e., a bacterial ortholog replacing the canonical plastid copy). Whether it is, strictly speaking, an ortholog (i.e., evolved by sequence divergence post speciation) is unclear.

-Line 337-340. "While we found ample evidence of cryptophyte nucleomorphs elsewhere, we found no evidence of putative leptophyte nucleomorphs". What does 'elsewhere' mean here, exactly? As noted above, I think there would be space to integrate the methods information on how they searched for the leptophyte 18S rRNA gene sequences. This would be good for completeness and would help nonspecialist readers better understand the nuances of the alternative models presented in Fig. k4.

Signed: John Archibald

Version 2:

Reviewer comments:

Reviewer #1

(Remarks to the Author)

The revised manuscript provides significantly improved results and discussion, strengthening its overall impact. Based on the authors' claims in the response to the comments, I no longer doubt its suitability. However, I would still like to add comments on a few points.

I previously noted that a major weakness of this manuscript is the lack of validation for the proposed phylum-level informal grouping of the so-called "leptophyceae", even though the authors' tremendous effort to compile a resource database of ~700 plastid genomes from marine pelagic ocean metagenome. In their response letter, the authors emphasized the considerable difficulty in locating putative leptophyte species with the available genome sequence through FISH etc. and underscored the practical challenges of setting up long-term experiments. I fully understand this limitation. But there still seems to be a reluctance to embrace the scientific proposition of leptophytes, which might stem from the absence of "biological" characterization for this new lineage. I think we have to meet halfway to make this study more meaningful with the current metagenomic dataset.

To that end, I suggest the authors make fuller use of metagenome data to provide environmental and ecological context for leptophyceae by:

- 1) Estimating their relative abundance and co-occurrence with other species groups using metagenome reads or assemblies.
- 2) Assessing correlations between meaningful environmental variables (e.g., temperature, light intensity, nutrient availability) and the occurrence of leptophyceae.

I recommend making this environmental and ecological metagenomic analysis a main figure in the manuscript. This would offer a broader "biological" perspective on leptophyceae and could stimulate further interest and targeted searches for this lineage in the ocean. For your reference, please see Fig. 4 of Alexander et al. (2023), Eukaryotic genomes from a global metagenomic data set illuminate trophic modes and biogeography of ocean plankton. *MBio* 14(6): e01676-23.

I especially appreciate the newly added mitochondria section. Considering the coefficient of the regression model for the mt & pt MAG coverage (~6/40), the relative copy number between pt and mt DNA seems to be about 100:15 for leptophytes (though this could vary among species). This led me to think that, given the summed coverage of Lepto 3 and 4 in Supplementary Table 1 (as well as Supplementary Figure 12, where I can see low mean coverage but many dots in panels C and D), which are 116 and 105, respectively, it might be possible to recover some mt contigs from the co-assemblies of all the sites where Lepto 3 and 4 are present—similar to what the authors did for Lepto-01's mt MAG. The presence of the Lepto-01 mt MAG would additionally increase sensitivity in finding contigs, if any exist.

Line 176 "providing evidence that leptophytes can be abundant under certain conditions": This appears to contradict the authors' results, as most psbO coverages were ~3 (median), and no putative leptophyte psbO was found. As the authors stated that mitochondrial genomes are typically multi-copy (which also holds for plastid genomes), using plastid genome coverage as a proxy for species abundance could be misleading. Any explanation on this?

Minor points:

- Fig. 1: The taxonomic labels are hidden (Leptophytes etc).
- Both "co-assembly" and "coassembly" are used. Was this intentional?

Reviewer #2

(Remarks to the Author)

The full review is provided in the text document attached.

Reviewer #3

(Remarks to the Author)

I appreciate the extremely thorough revision (and response to reviewer comments) undertaken by the authors. I am satisfied with the changes made, and am excited to see the discovery of a putative 'leptophyte' mitochondrial MAG -- a very important addition to an already important manuscript.

Version 3:

Reviewer comments:

Reviewer #1

(Remarks to the Author)

The authors have addressed adequately the comments that I raised in the revised version of manuscript. Therefore, I have no further comments. It seems it is ready for publication.

Reviewer #2

(Remarks to the Author)

I have provided a very detail account on the previous two versions of the manuscript, so this time I took the liberty of immersing myself more shallowly into the updated version, also because of being extremely occupied by other duties for an extended period of time from now on. Hence, I have restricted myself to checking the responses by the authors to the critical points I raised in my previous review and glanced over the version of the manuscript with tracked changes, and it seems that all the issues have been dealt with satisfactorily. Crucially, it seems Jamy et al. have completely rebuilt the mitochondrial phylogenomic dataset, which was my major concern in the previous review. Importantly, as I expected the new phylogenetic tree does not change the key conclusion of the previous analysis that the putative leptophyte mitogenome constitutes as sister lineage of haptophyte mitogenomes. However, having focused on the new version of Fig. 4 I do see a few minor issues that need to be fixed:

1. Note that compared to the previous version you have changed the name of the haptophyte correctly called "Diacronema lutheri" to the outdated name "Pavlova lutheri". I think this is most likely not intentional, as you have kept the name "Diacronema lutheri" in the figures displaying results of plastid phylogenomic analyses. Please fix the name for the sake of accuracy and internal consistency.
2. I would object the delimitation of "Cryptophytes" in the mitochondrial tree, which is expanded to embrace *Neptuniomonas avonlea*. The point is that "cryptophytes" are generally presented in the paper as a group stemming from an ancestor that has acquired a higher-order plastid. Indeed, the very etymology of cryptophytes/Cryptophyta/Cryptophyceae is indicative is an algal group comprised of members that are predominantly, or at least ancestrally, photosynthetic. Hence, despite being aware of the fact that the concept of "cryptophytes" varies in the literature, I strongly endorse its use as an equivalent of the cryptist clade "seeded" by an acquisition of a plastid, i.e. as the algal group in Cryptista (analogously to Euglenophyceae being the algal subgroup of Euglenida). As *Neptuniomonas avonlea* and most likely presumably goniomonads in general do not have a plastid, and there is also considerable genomic evidence that this is not a secondary state, I find including goniomonads among cryptophytes illogical. Imagine a naïve reader of the paper comparing Fig. 4 and Fig. 5. A conclusion such a reader must reach is that the plastid acquisition in the stem cryptophyte lineage depicted in Fig. 5 must have occurred before the divergence of *Neptuniomonas avonlea* and the (other) cryptophytes, which is however nearly certainly not true. Hence, consider narrowing the delimitation of cryptophytes in Fig. 4 to those taxa that have a plastid to avoid conveying a potentially misleading message. A note, the clade now annotated as "Cryptophytes" by Jamy et al. may alternatively (more aptly) be called Cryptomonada.
3. Related to the latter point is an issue with the paraphyletic grade delimited in Fig 4 as "Other Cryptista". If the previous point (narrowing cryptophytes to the plastid-bearing lineage) is accepted, then in principle "Other Cryptista" would have to be extended to embrace also *N. avonlea*. However, the real issue here is that as delimited at the moment the grouping comprises *Microheliella maris*, whose status as a "cryptist" is contentious. More specifically, the recent literature discriminates between a narrower clade Cryptista, which does not include *Microheliella*, and a more inclusive clade that has *Microheliella* sister to Cryptista and is called Pancryptista; see <https://royalsocietypublishing.org/doi/10.1098/rsob.210376>. I do admit that an alternative concept of Cryptista exists, advocated by the late Tom Cavalier-Smith, including in his posthumously published 2022 paper (<https://link.springer.com/article/10.1007/s00709-021-01665-7>), in which *Microheliella* is classified as a member of a newly established subphylum Endohelia in the phylum Cryptista. However, I personally find the distinction of Cryptista and Pancryptista well founded and functional, and I encourage the authors to embrace it in their paper.

And this is it, please accept the paper for publication after these few remaining points have been addressed by Jamy et al. I am not interested in seeing the manuscript once again before I find this important study published in Nature

Communications.

Dear Editor,

Please find below our responses to the comments from the three reviewers on the original manuscript. Below, we have reproduced the decision letter verbatim, with our detailed responses to *all comments* interleaved in blue for clarity. As you will see, our full answer is rather extensive, but we think that even if all our arguments did not (could not) make it to the manuscript, the reviewers would appreciate a highly detailed letter. Based on the comments, we have extensively modified our manuscript, which we hope will now satisfy the requirements for publication in *Nature Communications*. We would like to thank you and the reviewers for all inputs—we believe that this work has greatly improved in the revision process. Please note that line numbers whenever provided correspond to the manuscript file *without* track changes.

The most significant change to the manuscript is the retrieval of a mitochondrial MAG inferred to be from leptophytes, which led to a new main Figure (Figure 4). Thus, we now not only present a near-full plastid genome for a new deep-branching algal group, but also a near-complete mitochondrial genome. This discovery greatly augments our story with new insights into models of plastid evolution. The manuscript title has therefore been updated to: “A new deep-branching environmental lineage of algae”, which better represents the updated manuscript.

REVIEWER COMMENTS

Reviewer #1 (Remarks to the Author):

Utilizing the vast data generated by Tara Oceans expeditions, the authors amassed ~700 non-redundant plastid genomes that are manually curated. They have found plastid genomes from the “Leptophytes”, which is a deep-branching algal lineage. Given their presence in the read data from nano-size fraction, the authors thereby named it as Leptophyte. Thorough phylogenetic analyses revealed that leptophytes forms clade with either haptophytes or cryptophytes, depending on the tree model.

Even though there are concerns about the novelty and credibility of this study which will be discussed in the following paragraphs, I would like to commend the authors for their efforts in handling and curating such a vast dataset. The scope and aim are clear and straightforward. The findings in this study must provide an important base for exploring the diversity of plastids.

–Response: We thank the reviewer for their encouraging initial remarks.

However, I’d like to raise two major concerns regarding this study: its originality and the credibility of its taxonomic and evolutionary interpretation. First, the originality of this study raises concerns regarding its suitability for publication in *Nature Communications*. This lineage was previously described as DPL2 by Choi et al. (*Current Biology*, 2017). That study, which was also based on Tara Oceans data, documented the presence of this lineage in the photic zone and within the picoplankton size range. It also proposed a close phylogenetic relationship between DPL2 and haptophytes, with moderate support (posterior probability of 0.92). The current study does not provide significant new biological insights such as morphological or ecological information. Although the plastid genomes of

the “Leptophytes” were recovered, their gene content appears largely similar with that of haptophytes (as shown in Figure 2), and no notably unique features are reported.

–Response: We regret that the reviewer does not agree with our opinion—backed up by the other two very positive reviews—that this study presents several major novelties that we believe fully meet the criteria for publication in a top-tier journal like *Nat. Comm.* Firstly, we would like to insist that this study goes well beyond the really interesting case of leptophytes. As noted, we present a newly and highly original resource of about 700 manually curated plastid environmental genomes of marine pelagic algae, including groups such as diatoms or haptophytes that are crucial for ocean ecosystems. This represents the largest database available to date for the broad diversity of plastid genomes, including many groups that totally lack reference genomes. All of these genomes were made available to the scientific community. This dataset provides a better picture than ever before of plastid genomic diversity, and will be inevitable in all future studies aimed at understanding the diversity and evolution of marine algae. Moreover, our study is highly novel in that it corresponds to the first global plastid metagenome-assembled genomes dataset—we call them ptMAGs—for environmental algae. Since the deposition of our preprint (on Jan 18 2025), another preprint became available that confirms the high potential of this approach for organellar metagenomics (Shrestha et al. 2025. Global metagenomics reveals plastid diversity and unexplored algal lineages. <https://doi.org/10.1101/2025.03.28.644651>.) While metagenomics has transformed the study of bacteria and archaea, it has been much less applied to microbial eukaryotes—owing to their much higher complexity—and only a few small-scale attempts have been made to resolve organellar genomes. Thus, we believe that our global study of marine plastid genomes will serve as a reference for similar studies of environmental organellar genomics in the future.

On the perceived lack of novelty compared to the Choi et al. publication (Current Biology, 2017, <https://doi.org/10.1016/j.cub.2016.11.032>) initially reporting the DPL2 lineage, we again respectfully but firmly disagree. This initial study was exclusively based on short 16S metabarcoding data, and in our view it is not reasonable to downplay the significance of recovering complete genomes—including the full plastid genome of a novel group—as offering limited value compared to a few short 16S amplicon sequences. As discussed below, we now also present a putative mitochondrial MAG for leptophytes, reinforcing the novelty aspect. As the broader field has recognised for a long time for prokaryotes, but now also increasingly for eukaryotes (e.g. <https://doi.org/10.1016/j.xgen.2022.100123>), genome-resolved metagenomics enables advanced phylogenomic analyses with the use of complex models of evolution (which we employ here), gene content comparisons, and functional inferences that 16S data alone does not provide. Importantly, we not only recover genomes that correspond to the previously identified DPL2 16S lineages, but also reveal an unsuspected broader genetic diversity for leptophytes (~10% divergence), which prompted us to propose a new group *encompassing*—i.e. not simply corresponding to—DPL2. We also provide a new important ecological perspective since we show that the group is not always rare, providing a new and critical perspective on this lineage.

With respect to the taxonomy of this lineage, the authors should conduct a more thorough investigation into its relationship with haptophytes, using not only just DNA sequences in the metagenome data, but also data derived from the actual organisms. Given the phylogenetic placement, relatively short branch lengths, and the high similarity in plastid gene content with haptophytes (Figure 2), it is challenging to rule out the possibility that this group represents just another early-diverging lineage of haptophytes rather than a novel clade. Haptophytes are defined by unique features such as the presence of a

haptonema and specific pigment compositions. Without additional morphological data (e.g., FISH to trace the cell), it is premature to propose "Leptophytes" as it implies the recognition of a novel (sub-)phylum level eukaryotic lineage different than haptophytes. Such a taxonomic designation would require far more robust and diverse evidence and I think it is currently impossible.

–Response: We agree with the reviewer that isolating the host cell of leptophytes is of utmost importance, and we have already stated this in the discussion (lines: 391-393). Of course, were we able at this point to obtain morphological information on the host cells, we would include that in the manuscript. Unfortunately, this is an unrealistic request for a manuscript revision. As we show (Supplementary Figure 12), leptophytes are often extremely rare, and the few places where they are relatively less rare are very hard to reach (for example in the Canadian arctic). Even if we had suitable samples available with confirmed presence of leptophytes at high enough abundance—which we don't, establishing a FISH assay as suggested takes several months of optimisation in the best case scenario. To put this in perspective, the Burki lab has recently hired a postdoc to carry out just that—finding the leptophyte cells—but this is planned over a 3 years period.

Regarding naming the new lineage leptophytes, we argue that it is now widely accepted across the domains of life—Bacteria, Archaea, viruses, but also eukaryotes—that even single-gene sequence data alone can justify the informal naming of new lineages. Of course genomic data is preferable; the famous case of Asgard Archaea is one recent example where metagenomics led to the informal description of an entire superphylum (<https://doi.org/10.1038/nature21031>, <https://doi.org/10.1371/journal.pgen.1007080>). For eukaryotes, one can think of MALV or MAST as important lineages that were initially exclusively named based on single-gene sequence data, but there are many other examples. With leptophytes, we emphasize that we are not proposing a formal taxonomic classification here, but simply acknowledging—based on plastid genome data, including a near-complete genome—a new distinct environmental clade, and giving it an informal name to facilitate discussion. While our plastid data remain ambiguous regarding the exact position of leptophytes, our new mitochondrial data is clearer: leptophytes are sister to haptophytes. Whether this means that in the future haptophytes will be expanded to include leptophytes, or that leptophytes will remain a distinct lineage, one currently cannot say as it will require detailed morphological data not available at the moment (for example to check for the presence of a haptonema, as suggested by the reviewer). We have now made this clearer in the discussion (lines: 391-393).

But, even if future work shows that these organisms fall within haptophytes, we believe that the name “leptophytes” would still appropriately apply to this specific lineage. Among the many examples that we could cite to support our view, one relatively recent discovery (or re-discovery) is particularly relevant to the discussion here as it also involves haptophytes. Rappemonads were initially named to describe a group of plastid 16S rRNA sequences of uncertain evolutionary origin, but weakly associated with haptophytes (<http://doi.org/10.1128/AEM.64.1.294-303.1998>, <https://doi.org/10.1073/pnas.101333710>). As a putative new algal lineage, this group of amplicon sequences was informally named rappemonads (some low resolution CARD-FISH images confirmed the presence of plastids, but otherwise did not provide any morphological clues as to what this group was). Years later, it took the lucky establishment of a culture (*Pavlomulina ranunculiformis*) that turned out to be closely related to the environmental sequences of rappemonads to enable more detailed morphological work and sequencing data to unambiguously show that *Pavlomulina* and more generally the rappemonads are in fact *bona fide* members of haptophytes, forming a new class within the group

which was called Rappephyceae (<http://doi.org/10.1016/j.cub.2021.03.012>). Thus, as this example shows, the informal naming of rappemonads based on sequence data did not preclude the later establishment of a high-ranked taxonomic group of haptophytes based on a culture, on the contrary it inspired its name. We believe that the very same situation applies to leptophytes, with the important differences that we already have plastid genomes not just 16S rDNA, now likely also a near-complete mitochondrial contig, and we know the group is “at best” sister to haptophytes, not within it.

Although this study presents some novel plastid genome assemblies derived from publicly available sequence data, I think the manuscript falls short in terms of originality and lacks sufficient data for the thorough testing of evolutionary and phylogenetic hypotheses expected for publication in *Nature Communications*.

–Response:

At the risk of repeating ourselves, we respectfully disagree in full with the notion that we just present “some novel plastid genomes”. As argued above, we strongly believe that our study is much, much more than that and continue to think (as reviewers 2 and 3 do), even more than before based on the added analyses, that its scope is well within the range of *Nat. Comm.*

Line 205: Please provide a detailed description for the reason why the authors choosed 93 genes. What criteria were applied in choosing these specific genes? I am also curious about whether using an alternative or modified gene set might result in a different topology, or if the current topology is robust and consistent across various gene selections.

–Response: Thank you for pointing this out. This was lacking and we have now included a detailed description of the genes selected in the Methods section (lines 512-522):

“We began by using the two plastid gene sets originally defined by Janouškovec et al 2010⁵⁶: a larger set of 68 plastid genes, and a more conserved subset of 34 genes. These served as the foundation for constructing preliminary phylogenies. During dataset assembly (see below), two genes were excluded: *acsF* due to low taxon occupancy, and *psbH* due to anomalously long branches in its single-gene tree. This reduced the gene sets to 32 and 66 genes, respectively. The resulting preliminary phylogenies identified leptophytes as a novel clade, but could not confidently resolve their position using maximum-likelihood analyses (Supplementary Figs. 24-25). Therefore, we attempted to increase the phylogenetic signal by including additional genes used in Ponce-Toledo et al 2017⁵⁷ and Pietluch et al 2024⁴⁶, provided they were also present in leptophytes, resulting in a final set of 93 genes (Supplementary Table 6).”

Supplementary Table 6 lists the genes included in our analyses as well as the corresponding source studies. We emphasize that our study represents the most comprehensive plastid phylogenomic analyses to date with a substantially expanded dataset. Notably, we employed highly sophisticated models including the recently developed site- and branch-heterogenous substitution models presented in Williamson et al 2025 (<https://doi.org/10.1038/s41586-025-08709-5>).

The phylogenies inferred with the 32 and 64 gene-set concatenated alignments are consistent with the phylogeny inferred with the final 93 gene dataset, with branch support largely increasing the more genes

are included. Leptophytes branch sister to both cryptophytes and haptophytes in all phylogenies (CAT+GTR+G model), albeit with low support. Overall, these results indicate that the topology obtained is robust across various gene selections, but varies with the model used (e.g. the site-heterogeneous CAT-GTR+G model vs. the site-and-branch-heterogeneous GF-MIX model) as described in the main text.

Reviewer #2 (Remarks to the Author):

The manuscript by Jamy et al. delivers very much awaited results of continued exploration of the Tara Oceans metagenomic data, now focusing on the representation of the diversity of plastid genomes in the data. The scientific merit of the study is obvious. Previous investigations utilizing as phylogenetic markers individual genes retrieved from environmental DNA samples have pointed to the existence of unknown deeply diverged plastid lineages or have documented the existence of novel uncultivated groups forming separate deep lineages within known major algal groups (green algae, ochrophytes, haptophytes), raising the question whether they also have a plastid and if yes, whether it is photosynthetic. The work by Jamy et al. progresses towards addressing these questions, although the concept of the study is admittedly not such as to aspire on provide comprehensive answers. The scope of the study is from the start restricted by focusing on metagenomic data from the sunlit ocean, so it cannot cover organisms that live in other types of marine habitats, let alone those thriving in freshwater ecosystems. Nevertheless, even focusing on one particular category of habitats have yielded important new findings that make the study by Jamy et al. highly valuable. In this regard it is interesting to compare this study with a recently released preprint by Shrestha et al. (bioRxiv, <https://doi.org/10.1101/2025.03.28.644651>), which reports on an exploration of plastid genomes in a much broader set of metagenomic samples (including the freshwater ones). The general conclusions of both independent studies are largely overlapping, including the possibly most exciting result of both studies, that is obtaining the first (nearly complete) plastome sequences from an uncultivated marine lineage denoted by Jamy et al. as “leptophytes” and embracing the previously known DPL2 lineage of plastidial 16S rRNA genes hypothesized to represent a new algal group. Thanks to the effort by both teams it is now clear there indeed exists a deeply diverged evolutionary lineage of plastids affiliated to the plastids from haptophytes and cryptophytes. Nevertheless, the identity of the actual organismal (i.e., host) lineage harbouring these plastids remains unknown, leaving a major gap in our knowledge of algal diversity open.

Thus, while I am generally very positive about the study by Jamy et al. and I believe it is in principle worth publishing in a top-tier journal like Nature Communications, I believe some extra work and improvements in the presentation of the data is necessary to more fully realize the potential of the material in hand. Below I provide a list of points that the authors should address. Despite their number they do not require an excessive amount of time or effort, and I hope they are all constructive, helping to make the paper even better than it is now.

–Response: We sincerely thank the reviewer for their encouraging remarks, and thoughtful, constructive, extensive, and detailed review. We have addressed their suggestions below, some of which have greatly improved the manuscript to make it, we believe, more impactful.

Critical points regarding the methodology:

- line 94: two references are cited #16 and #19. The latter is a paper describing the development of a particular software (MarkerMAG) and it is not clear to me how this paper relates to what the authors mention in the respective sentence citing the paper. Is it possible that you wanted to cite something else here? Regardless, if the tool MarkerMAG was indeed used in the study, it should be clear how (it is not mentioned in the Methods section).

–Response: We thank the reviewer for identifying this error, as the ref #19 was inadvertently inserted in the sentence without any relevance. We have now replaced the reference by the relevant one.

- The description of methods and procedures additionally misses a description of how the mOTUs-db was used to retrieve the two ptMAGs analysed by the authors. I have checked the mOTUs-db website it is not at all obvious how to proceed to retrieve any ptMAG from the database, so please add a few lines on this.

–Response: The reviewer is correct about the methods section not extensively describing the recovery of two relevant ptMAGs from mOTUs. To address this point, we have now added this paragraph in the method section (lines 454-465):

“Recovery of additional ptMAGs from the mOTU database.

To broaden the scope of our survey, we exploited 85,123 metagenomic assemblies from the mOTU global metagenomic resource covering a wide range of environmental samples¹⁸. A total of 1,969,342 RNAPolA genes were previously characterized from this resource⁴⁸. Note that RNAPolA displays a very similar evolutionary signal compared to RNAPolB⁴⁹. Here, we identified 6,954 contigs >50 kbp that contained a RNAPolA gene with relatively high sequence similarity to that of our characterized ptMAGs (DIAMOND blast, with percent identity >70% at the amino acid level). Based on preliminary phylogenies, we found one contig characterized from a Tara Oceans metagenome and corresponding to a leptophyte (contig ID Lepto-01_REFM_CHLORO_00001 with a length of 104,203 nt). In addition, we also found one contig (also characterized from a Tara Oceans metagenome) and corresponding to a deep-branching clade of haptophytes (contig ID REFM_CHLORO_00002 with a length of 84,869 nt). We integrated these two contigs into our database of ptMAGs.”

- line 101: “filtering for redundancy (average nucleotide identity <98%)” – the sentence does not make sense to me unless assuming that instead of “<98%” there should be “>98%” used in the text. Please check this.

–Response: Thank you, fixed!

- There seems to be some chaos regarding the number of reference plastid genomes used in the analyses by the authors. According to the information provided in line 102 and then later in the Methods section (lines 381 and 389), the authors selected 164 reference plastid genomes. These are supposed to be listed in Supplementary Table 1 (see line 103) and I assume these correspond to plastomes with identifiers starting with “REFG_”. However, there are 167 rather than 164 REFG_” genomes listed in the table. Furthermore, while there are 167 genomes listed, the identifiers go from REFG_CHLORO_00001 to REFG_CHLORO_00166. This is because the table includes two different plastomes labelled as “REFG_CHLORO_00165”, one corresponding to KY860574 (a cryptophyte plastome) and included

in the list between REFG_CHLORO_00020 and REFG_CHLORO_00021, and the other corresponding to NC_020371 and listed at the expected position (after REFG_CHLORO_00164). I am not sure if such a labelling discrepancy might have affected the analyses reported by the authors, but they certainly have to check this carefully and also ensure each plastome has a unique identified in the table. To make the confusion even worse, at other places in the manuscript (legend to Fig. 1, line 117; Methods – line 419 and 435) the authors mention the use of 180 reference plastid genomes. If two different sets of reference plastid genomes were indeed used in different analysis (but I guess this is possibly not true), then some clarification for the rationale is needed and a list of all the 180 references needs to be provided. Finally, still another number of reference plastid genomes – 166 – is mentioned in the legends to Supplementary Figs 2-5.

–Response: Thank you very much for pointing this out. We greatly apologise for the confusion and have now fixed the inconsistencies. We carefully checked our collection of ptMAGs and reference genomes and report the correct numbers below as well as in the manuscript:

- **660 ptMAGs.** This is down from 667 ptMAGs in the previous version, following the removal of seven ptMAGs with high redundancy levels (>15%). Additionally, these 660 ptMAGs are, to the best of our knowledge, now free from mitochondrial contamination (see response to the corresponding comment below). The seven ptMAGs removed are: TARA_CHLORO_00225, TARA_CHLORO_00009, TARA_CHLORO_00548, TARA_CHLORO_00377, TARA_CHLORO_00247, TARA_CHLORO_00285, and TARA_CHLORO_00493.
- **166 reference plastid genomes.** This is down from 167 references in the previous version, following the removal of a taxonomically redundant reference (“Pavlova sp NIVA-4/92” as noted by the reviewer in another comment). These reference plastid genomes were used for both the phylogenetic analyses (Fig. 1, with a subset of taxa in Fig. 3), as well as for comparisons with the ptMAGs (Supplementary Figs. 2-6, with a subset of taxa in Fig. 2).
- In addition, we used **13 reference cyanobacteria** as outgroups in the phylogeny in Fig. 1 and Supplementary Fig. 7. The total number of references in Fig. 1 is therefore **179**.

Accordingly, we have:

- Updated the numbers in the manuscript. Deviations from these numbers are more clearly explained where applicable. For example, *Paulinella* was not included in the comparisons in Supplementary Figs. 2-6, thus the number of reference plastid genomes mentioned in the corresponding legends is 165.
- Re-run the alignments and phylogeny for Fig 1.
- Re-run the analyses for Supplementary Figs 2-6.
- Updated Supplementary Table 1.
- Updated the data provided in the Figshare repository.

- I think a major omission in the description of the methodology employed by the authors concerns the procedure used to create Fig. 2 (and its more detailed version displayed as Supplementary Fig. 15). The authors mention the analysis is based on 355 “plastid-encoded genes” (is a gene really “encoded”?), but it is not specified how these genes were selected and, most crucially, how the presence/absence of particular genes has been examined in individual genomes. Some plastid genes are rapidly evolving and it may be challenging to identify orthologs correctly. Furthermore, as is apparent from

Supplementary Fig. 15 many of the 355 genes are labelled orfXXX (where XXX stands for a particular number), i.e. they do not correspond to broadly conserved and well-annotated plastid genes. Please, provide a description how these genes were delimited and how orthology across the plastid genomes analysed was assessed. We need to be certain the presence/absence pattern of these genes as displayed in the scheme does not reflect simply the occurrence of the same “orfXXX” annotated as such by MFannot in different plastid genomes – note that the same label frequently means only the same length of the orfs, i.e. the same number of codons, not orthology as such. I think that for the sake of reproducibility, the authors should provide a table listing all the genes for all the plastomes so that an interested reader could unambiguously identify each gene (at the sequence level); these data are not provided even at the Figshare site linked to the manuscript.

–Response: We thank the reviewer for this comment as we were unaware that orfXXX indicates only the number of codons. Our previous analysis assumed orthology based only on gene annotation by MFannot.

We have now redone the analysis and described the methods in the manuscript (lines 681-696):

“Comparative analysis of plastid gene content in leptophytes and related lineages

To gain insights into the evolutionary dynamics and functional capabilities of the leptophyte plastid, we compared its gene content with that of red algae and other red algal-derived plastids. As ptMAGs can have missing genes due to incompleteness, we opted to use only reference plastid genomes where possible. For this analysis, we selected 15 rhodophyte, seven haptophyte, eight cryptophyte and 21 ochrophyte reference plastomes from our plastid genomic database. Similarly, we used the Lepto-01 ptMAG, the only near-complete leptophyte ptMAG, as the sole representative of leptophytes. We extracted amino-acid sequences from all selected plastid genomes and used OrthoFinder⁷¹ v2.5.5 with default settings to detect homologous proteins. This analysis yielded 281 phylogenetic hierarchical orthogroups (HOGs) containing 7,648 protein-coding genes (98.5% of the total). However, as no orthology-predictor is perfect, we manually refined the HOGs by inspecting gene annotations, single-gene trees, and performing BLAST and InterProScan searches (Supplementary Table 10). This step yielded 237 HOGs containing sequences from at least three taxa. Heatmaps depicting gene presence/absence were plotted using the R package pheatmap v1.0.12 (<https://github.com/raivokolde/pheatmap>), and species and orthologs clustered using UPGMA clustering.”

The results of this new analysis are consistent with previous results, and show that leptophyte plastids most closely resemble haptophyte plastids. The new Supplementary Table 10 now also allows readers to identify the sequences in each orthologous group from every plastid genome used in our analysis.

- The authors assessed the completeness of their ptMAGs by checking the occurrence of 44 core-plastid genes listed in Supplementary Table 5. A large proportion of these genes encode components of the photosynthetic apparatus, so these are expected to be missing from plastomes coming from non-photosynthetic plastids. It is not clear to me whether the authors have used the completeness as a criterion for filtering out any of the potential ptMAGs and if yes, whether this might systematically remove from their final ptMAG set the plastomes of non-photosynthetic plastids. Regardless, are there any ptMAGs in the final dataset that are likely representing non-photosynthetic taxa? I believe this is

relevant question and an interesting aspect of plastid diversity ignored in the present version of the manuscript.

–Response: While we estimated completeness of the ptMAGs during our investigations, we did not exclude any based on this criterion alone, as most of them had a relatively high completeness (average completion of 63%). Our criteria to remove ptMAGs were their length (minimum 20 kb), and redundancy estimate (max of 15%). As a result, our workflow is compatible with the recovery of non-photosynthetic ptMAGs, especially given that the marker gene used to extract ptMAGs (RNAPolB) is not related to photosynthesis. Based on functional annotations in the context of the main phylogeny (Figure 1), we did not identify any clade of non-photosynthetic ptMAGs in our database. As stated in the manuscript, we are aware that our workflow could not retrieve the peridinin plastid genomes of dinoflagellates (both photosynthetic and non-photosynthetic), given their highly unusual structure (highly fragmented and lacking *rpoB*).

Analyses specifically concerning leptophytes:

- Plastid genomes typically include a duplicated region known as the inverted repeat, but surprisingly this term does not appear at all in the text despite the fact it is an important feature with direct bearings to the process of assembly of plastome sequences. It is particularly notable that the leptophyte ptMAG, although incomplete, implies a plastome organization lacking an inverted repeat (at least judging from Supplementary Fig. 13). I believe the authors should explicitly mention whether they see any evidence for inverted repeats being part of leptophyte plastomes (careful manual analysis of the assembly data including raw reads may be necessary to clarify this), and if not, how unusual this would be compared to plastomes of haptophytes and cryptophytes. In addition, I think the authors should explicitly comment on what the Lepto-01 ptMAG is most likely missing. Judging from the map it is possible that only the remaining parts of the 16S and 23S rRNA genes (*rns* and *rnl*) plus two tRNA genes (*trnI* and *trnA*) that are usually placed in between these two genes are missing. In fact, the annotation of the Lepto-01 ptMAG as presented by the authors in Supplementary Figs 13 and 14 (and the respective annotation file at the linked Figshare site) is incomplete and misses a 5S rRNA gene (*rrn5*) that is located exactly when one would expect in analogy with other plastomes – in the unannotated (“intergenic”) region just downstream the *rnl* gene; the presence of the gene is readily seen when the respective region is analysed using Infernal Cmscan (https://www.ebi.ac.uk/jdispatcher/rna/infernal_cmscan). Hence, the authors should update the annotation of the ptMAG by adding this extra gene.

–Response:

Regarding inverted repeats

Since many plastid genomes often have inverted repeat (IR) regions, we did consider IR regions during our original analysis of the leptophyte plastid genomes. We did not observe an IR region, and originally thought that this could either reflect a true absence of IR regions, or simply be an artefact of sequence assembly (resolving repeats in metagenome assemblies of short-read sequence data is notoriously challenging).

Upon further consideration of the reviewer’s comment, we realised that the Lepto-01 contig—flanked by 16S rDNA on one end and 5S and 23S rDNA on the other—indeed implies the lack of IR regions. The question of whether the leptophyte plastid genome lacks repeat regions altogether (e.g. as in Pavloales and certain rhodophytes), or whether it has direct repeats instead (e.g. as in Isochrysidales

and certain land plants) is difficult to resolve. In principle, one could infer the presence/absence of direct repeats (DR) by examining read depth across the plastid genome; if present, the repeat regions would be expected to have approximately twice the coverage compared to the rest of the genome. In practice, however, this approach is difficult to apply in this specific case. The Lepto-01 ptMAG contains only a small fragment of the putative DR region (if it exists), i.e. the 16S and 23S rRNA gene fragments. These genes contain highly conserved sequence regions and it is difficult to distinguish between related taxa using the corresponding raw reads. This is one reason why rDNA genes are frequently missing from MAGs, or are chimeric. In fact, this may explain the likely 16S rRNA chimera we observe in the Lepto-04 ptMAG. Overall, we do not believe it is currently possible to resolve the presence or absence of DR regions in the leptophyte plastid genomes given the available data. Addressing this question will require either isolating leptophyte cells, or long-read metagenomics.

We have now added the following sentence to the main text (line 197-199): “Interestingly, leptophyte plastids genomes seem to lack inverted repeats commonly found in plastid genomes²⁷, although whether they instead contain direct repeats such as in some haptophytes²⁸, or lack repeat regions altogether remains unresolved (Supplementary Fig. 14).”.

We have also added a panel to Supplementary Fig. 14 (see below), with cartoons of possible leptophyte plastome organisation.

Regarding missing genes

First, we thank the reviewer for pointing out the missing 5S rRNA gene. The annotation for the Lepto-01 ptMAG has now been updated to include this gene in the genome map (Supplementary Fig. 14) and all relevant files shared on Figshare. On a related note, during the gene content analysis with OrthoFinder, we noticed that MFannot fails to correctly identify the *rpl22* gene in any input genome.

In the Lepto-01 ptMAG, the gene was identified as orf120. The annotation of this gene has now also been updated.

Furthermore, we now explicitly discuss other leptophyte missing genes in Supplementary Fig. 14. In the case that there are no repeat regions, it is possible that the only missing genes are trnI and trnA. If the plastid genomes contain direct repeats, leptophytes could be missing additional protein-coding genes present in the small single-copy (SSC) region.

- The authors have attempted to illuminate the nature of the “host” lineage harbouring the leptophyte plastid. Specifically, in Results they write (lines 165-166): “However, efforts to identify potential hosts for Lepto-01 at stations where its signal was strongest were unsuccessful (see Methods)”. In Discussion they then add this (lines 335-340): “While we found ample evidence of cryptophyte nucleomorphs elsewhere, we found no evidence of putative leptophyte nucleomorphs (see Methods)”. The methods section then indeed includes a section describing specific analyses the authors carried out in this regard. While the outcome of their effort has been negative, I believe these analyses should be featured more prominently in the manuscript and require some clarifications as well as some extra effort. Regarding the attempts to identify candidate leptophyte nuclear psbO genes and potential nucleomorph “RNAPolB” genes, one should bear in mind that these genes are expected to harbour intronic regions that disrupt the coding sequence. However, it is not clear from description of the search procedure employed by the authors whether and how they accounted for the possibility of the coding sequences of the target genes being interrupted by introns, which obviously complicates retrieval of the sufficiently complete amino acid sequences from the metagenomic data. Next, it should be explicitly mentioned that the authors were looking for “RNAPolB” genes specifically related to cryptophyte nucleomorphs (as sequences affiliated to homologs from chlorarachniophyte nucleomorphs would not be good candidates for leptophyte nucleomorph genes). In fact, I am slightly confused by the authors using the term “RNAPolB” genes. Eukaryotes generally have three different homologs of the single prokaryotic RNAPolB protein, corresponding to subunits of RNA polymerase I, II and III. So which protein of those do the authors mean here?

–Response: The reviewer raised several important points here, which we address one by one.

1. “The methods section then indeed includes a section describing specific analyses the authors carried out in this regard. While the outcome of their effort has been negative, I believe these analyses should be featured more prominently in the manuscript and require some clarifications as well as some extra effort.”

We agree, and the main text now includes a more detailed description of the search for sequences from the host lineage (lines 277-317). Excitingly, following the reviewer’s next suggestion, we have now characterized the mitochondrial genome of one leptophyte with high confidence, by combining phylogenetic signal and distribution patterns (see response to next comment for more details). The description for the search for the host nuclear sequences in the main text is as follows (lines 279-286):

“Plastid genomes do not necessarily mirror the long-term evolutionary trajectory of their host cell because of the complex history of endosymbioses^{2,4}. In an effort to identify potential nuclear-encoded sequences for leptophytes, we searched the Arctic stations data—where the leptophyte abundance was

the highest—for *psbO* and 18S rDNA candidate genes (see Methods). The *psbO* gene is essential for photosynthesis, it is always host-encoded and never found in non-photosynthetic organisms³⁶. Although it is derived from the endosymbiont and thus would not be a good phylogenetic marker for the host, our rationale was that it could pinpoint to host contigs that may contain additional genes. These analyses did not provide promising candidates for either gene”

2. “Regarding the attempts to identify candidate leptophyte nuclear *psbO* genes and potential nucleomorph “RNAPolB” genes, one should bear in mind that these genes are expected to harbour intronic regions that disrupt the coding sequence. However, it is not clear from description of the search procedure employed by the authors whether and how they accounted for the possibility of the coding sequences of the target genes being interrupted by introns, which obviously complicates retrieval of the sufficiently complete amino acid sequences from the metagenomic data.”

We agree with the reviewer that introns are a central component of the genomic landscape of unicellular eukaryotes.

Regarding *psbO*, we first searched the co-assembly of Arctic samples. While introns may have interfered with gene detection, a more significant challenge is likely the substantially lower coverage of eukaryotic nuclear genomes overall. To investigate this, we used the curated *psbO* database generated by Pierella Karlusich et al. 2022 (<https://doi.org/10.1111/1755-0998.13592>), and retrieved the top 50 *psbO* sequences from two metagenomes of *Tara* Station 194, where *Lepto-01* ptMAG coverage ranged from 124× to 70×. This database is based on metatranscriptomic assemblies and thus avoids the issue of introns. Most *psbO* sequences had only a small number of metagenomic reads mapping to them per sample (median for the top 50 most abundant sequences = 3), except for those derived from highly abundant chlorophytes and ochrophytes. This indicates that nuclear genomes generally have much lower coverage than plastid genomes, likely contributing to our failure of retrieving the leptophytes *psbO* sequences.

Regarding RNAPolB, the gene database comes from metagenomic assemblies, thus may contain introns, but this approach proved successful in targeting and recovering nearly one thousand eukaryotic MAGs from the sunlit oceans (<https://doi.org/10.1016/j.xgen.2022.100123>). In addition, while this is not yet in the form of a preprint, we share here with the reviewer that we have also successfully characterized a major new clade of red-algal nucleomorphs from the sunlit oceans using the same RNAPolB gene database. These two surveys worked despite the occurrence of introns because as long as a fragment is long enough, it can be adequately placed in a global phylogeny to identify putative clades of interest. At least, this is our experience with the *Tara* Oceans data over a 5+ years period. As a result, we believe that either (1) the leptophytes have a nucleomorph that did not have enough signal to be assembled in our *Tara* Oceans metagenomic survey (nucleomorphs have usually a similar coverage as the nuclear genome, much less compared to the plastid and mitochondrial genomes), or (2) the leptophytes lack a nucleomorph. Unfortunately, we could not answer this question in the present study. As a big follow-up effort, we ambition to reach the sequencing level needed to assemble a leptophyte nuclear genome using targeted filters. By doing so, we hope to be able to answer the pressing question of the nucleomorph.

3. “Next, it should be explicitly mentioned that the authors were looking for “RNAPolB” genes specifically related to cryptophyte nucleomorphs (as sequences affiliated to homologs from chlorarachniophyte nucleomorphs would not be good candidates for leptophyte nucleomorph genes).”

We have now clarified that we specifically looked for a nucleomorph related to cryptophytes (line 700-702):

“Specifically, we surveyed the assembled RNAPolB genes, following the protocol used to characterize ptMAGs but this time using publicly available cryptophyte nucleomorph RNAPolB genes as a reference.”

4. “In fact, I am slightly confused by the authors using the term “RNAPolB” genes. Eukaryotes generally have three different homologs of the single prokaryotic RNAPolB protein, corresponding to subunits of RNA polymerase I, II and III. So which protein of those do the authors mean here?”

We are also aware that eukaryotes have 3 types of RNAPolB (also 3 types of RNAPolA), and in our previous surveys (the published eukaryotic MAGs and the unpublished nucleomorph MAGs), each RNAPolB homolog is an opportunity to identify and subsequently characterize a genome of eukaryotic origin. In previous work on eukaryotic MAGs, we concatenated the 3 types of RNAPolB as well as the 3 types of RNAPolA to infer a phylogeny (see figure 2 of <https://doi.org/10.1016/j.xgen.2022.100123>). The key point is that having multiple types of RNAPolB (all being detectable with our dedicated HMM) is not an issue when screening for eukaryotic genomes across metagenomic assemblies. As a final note, compared to psbO, a major advantage of RNAPolB is its occurrence in all eukaryotes, not just those performing photosynthesis.

- Regarding the search for candidate leptophyte 18S rRNA gene: my own experience with metagenome data analyses is such that particular 18S rRNA sequences are frequently missing from sequences assemblies despite the fact they are represented in the sequencing reads. Was the analysis of the V9 metabarcoding ASV sequences mentioned in lines 602-605 based directly on reads? Likely so, but this is not completely clear from the very brief description of the procedure. Regardless, I think it would make sense to dig a bit deeper, and above all to test directly the most likely possibilities or the hypotheses raised by the authors themselves. Specifically, the authors should directly check the reads from the metagenomic samples that contain leptophyte sequences for the presence of 18S rRNA signatures corresponding to the known uncultivated haptophyte-related lineages (presented in refs #42 and #43) or to the CRY3 lineage. Next, there is a recent preprint by Romero et al. (bioRxiv, <https://doi.org/10.1101/2025.03.26.645542>) reporting on a systematic analysis of 18S rRNA sequences in metagenomic data and reporting in numerous novel eukaryote lineages. One is wondering whether this study might help identify candidates for leptophyte 18S rRNA sequences. Finally, I believe it is likely that at least partial mitochondrial genome sequences from leptophytes should be present in samples containing their plastid genes, so I strongly encourage the authors to look for mitochondrial genome sequences co-occurring with the leptophyte plastome sequences and representing phylogenetically separate lineages placed in the global tree such that they at least potentially may correspond to leptophyte mitochondria. The leptophyte story is clearly central to the whole paper and this extra effort may make is stronger.

–Response: The reviewer again made several useful suggestions. We address them one by one.

1. Search for 18S rDNA gene

The reviewer is correct that our description of the 18S rDNA gene is rather brief, a point also raised by reviewer 3, and we apologise for the confusion. We performed a thorough survey to search for a putative leptophyte 18S rDNA gene which we describe below, along with a biological interpretation of the results.

First, we note that linking 18S rDNA sequences to a MAG is inherently challenging. In the case of Lepto-01, we would expect the associated 18S rDNA sequence to correspond to a deep-branching lineage related to haptophytes, potentially within the HAP-4 clade (<http://doi.org/10.1127/pip/2016/0052>). Note that we initially considered CRY-3 a plausible candidate, but subsequent results from the mitochondrial survey and phylogenetic analyses have now made this unlikely. Importantly, identifying such novel 18S sequences in targeted samples alone is insufficient to establish a robust link to a MAG. Correlated abundance patterns across samples are also critical for inferring a link to the same biological entity.

Our initial strategy was to retrieve 18S rDNA sequences from a co-assembly of samples where Lepto-01 was highly abundant. While we acknowledge that this approach can overlook many eukaryotic taxa, it is a promising method for detecting abundant taxa in the metagenomes, as demonstrated by the recent Romero et al. preprint (<https://doi.org/10.1101/2025.03.26.645542>), and by previous work (<https://doi.org/10.1111/1755-0998.13147>). Our rationale was that this targeted approach might yield a small, manageable set of near-full length candidate sequences for further investigation. However, this attempt did not result in any candidates for leptophytes, as all the 95 retrieved sequences were linked to well established clades. This outcome could be due to the difference between abundance of the Lepto-01 plastid DNA and the host nuclear DNA in the same samples. That is, while the plastid DNA appears to be highly abundant in these samples, the host cells, and thus the host nuclear DNA, may be relatively rare. This interpretation is also supported by our failure to retrieve the candidate leptophyte *psbO* gene from the same coassembly. Furthermore, we show below (**2. Search for leptophyte mitochondria**) that plastid DNA in Lepto-01 seems to be at least seven times more abundant than its mitochondrial DNA. It is therefore reasonable to infer that the nuclear DNA is even less represented in the metagenomes.

We then turned to metabarcoding data. We followed a workflow established by a recent study (Zavadska et al 2024: <https://doi.org/10.1371/journal.pone.0303697>) to link protist MAGs with protist metabarcodes. Note that this study demonstrated that while it is possible to infer a one-to-one link between a MAG and an OTU/ASV, it is not always the case. The authors also observed other patterns, such as multiple OTUs corresponding to one MAG, or vice versa, highlighting the challenges of linking metabarcodes with MAGs. For our analyses, we retrieved 5,366 V9 ASVs from 15 samples with the highest Lepto-01 abundance. These ASVs included a substantial amount of novel diversity (1,868 ASVs < 90% similar to reference sequences), as well as sequences similar to deep-branching haptophyte clades (such as HAP-4 and HAP-5). We attempted to match each ASV to leptophytes, by using their relative abundance across the 15 samples to estimate a correlation coefficient (Spearman's correlation) with the relative and absolute abundance of the Lepto-01 ptMAG in the same samples. This resulted in 13 and seven candidate ASVs respectively with a Spearman's correlation value > 0.7. These ASVs were

investigated further by BLASTing against the PR² database and nt. However, none of the candidates were suitable as matches with leptophytes (the top matches in each case were with cercozoan sequences).

We hypothesize that this outcome is due to an extra challenge in linking metabarcodes with plastid MAGs. The abundance ratio between the 18S rDNA gene and the rest of the nuclear DNA of a population of cells can be expected to remain constant across samples. However, the same cannot be said for the abundance ratio between the 18S rDNA gene and plastid DNA; the amount of plastid DNA in a cell can be expected to vary as a function of the number of plastids per cell, and the number of genome copies per plastid. Several studies have shown that, at least for the green alga, *Chlamydomonas reinhardtii*, light and nutrient conditions impact the ratio of ptDNA:nucDNA (<http://doi.org/10.1104/pp.113.216291>, <http://doi.org/10.1105/tpc.106.045427>). Copy number of organellar genomes also seem to increase in the diatom, *Thalassiosira pseudonana*, during optimal growth conditions (personal communication with Thomas Mock). Plastid genome copy number may be similarly plastic in leptophytes, preventing a one-to-one match between an ASV and the Lepto-01 ptMAG using abundance correlation.

We have now expanded the relevant text in the methods section to better explain our rationale and workflow (lines 729-748).

2. Search for leptophyte mitochondria

This was really an *excellent* suggestion. We searched a co-assembly of the top six filters where Lepto-01 was most abundant for mitochondrial sequences and identified 34 mitochondrial contigs (details in the methods section on lines 750-811). Co-occurrence analyses revealed one contig (Lepto-01_mtMAG_004), a near-complete mitochondrial genome, which nearly perfectly correlated in abundance with the Lepto-01 ptMAG ($R^2 = 0.96$, $p < 0.01$) (a note on abundance correlation at the end of this comment). This contig therefore most likely corresponds to the Lepto-01 mitochondria (although this can only be confirmed once leptophyte cells are isolated). A phylogenetic analysis of 28 mitochondrial genes revealed the Lepto-01 mtMAG to branch sister to all other haptophytes. This is an important result, and accordingly, we have now added a new figure in the main text (Fig. 4).

The finding also has large implications for possible scenarios of plastid transfer (for instance by ruling out a scenario in which leptophytes are sister to cryptophytes in the eukaryotic tree), and so we have now updated Fig. 5 accordingly. Importantly, we have also added a new scenario of plastid transfer where the ancestor of leptophytes and haptophytes first received the plastid from rhodophytes, and then transferred the plastid from stem haptophytes to cryptophytes.

A note on abundance correlation between the Lepto-01 ptMAG and mtMAGs

While doing preliminary analyses, we noticed that there was a surprising pattern in abundance correlation with size fraction, i.e. the ratio of mtMAG coverage vs. the Lepto-01 ptMAG coverage was different in the larger and smaller size fractions.

To illustrate what we mean, we first show the new Supplementary Fig. 21 in the ms which shows abundance correlations across metagenomes with the 0.22-3 micron size fraction. The putative Lepto-01 mtMAG is plotted in the top left corner. The ratio between the Lepto-01 ptMAG to mtMAG abundance is roughly seven, and the abundance correlation has an R^2 value of 0.96.

Abundance correlations between Lepto_01 ptMAG and mtMAGs across Tara Oceans metagenomes (size fraction 0.22–3 μ m)

Next, the plot below shows abundance correlations across metagenomes with the 0.8-2000 micron size fraction. Here, the abundance correlation between the Lepto-01 ptMAG and mtMAG has an R^2 value of 0.95, but the ratio between their abundance is roughly 24 (about three times higher).

Abundance correlations between Lepto_01 ptMAG and mtMAGs across Tara Oceans metagenomes (size fraction 0.8–2000 μ m)

Finally, the plot below shows abundance correlations across all metagenomes regardless of size fraction. Here, the R^2 value of the abundance correlation between the Lepto-01 ptMAG and mtMAG drops to 0.65, which seems to be an effect of combining the various size fractions.

To summarise, we observe an effect of size fraction on the putative Lepto-01 mtMAG abundance in a particular sample. While the Lepto-01 mtMAG generally has a lower coverage than the ptMAG (possibly due to higher genome copy number in the plastid), we observe higher abundance values in the smallest size fractions (0.22-3 μm). Why we see this pattern is unclear. One possible explanation is that perhaps some leptophyte cells burst during size filtration and mitochondria pass through the small filter pores while the larger chloroplasts are retained. What is clear, however, is that abundance correlation should be tested while taking size fraction into account.

To conclude, we are very confident that mtMAG_004 corresponds to the mitochondria of Lepto-01. When taking size fraction into account, we observe a strong correlation between the Lepto-01 ptMAG and mtMAG_004, and mtMAG_004 branches sister to haptophytes in the mitochondrial phylogeny. Together, these results strongly suggest that we have recovered the mitogenome of Lepto-01.

However, we report only the results from the 0.22-3 μm size fractions, as they correspond to the largest number of metagenomes, and to not make the story overly complex.

Additional comments to the analyses and their interpretation:

- I think the informativeness of the study would be further increased by elaborating a bit more on the identified ptMAGs other than those from leptophytes. For example, according to Supplementary Fig. 3 the authors have identified a few MAGs that are exceptionally large, in one case >600 kbp and according to the data presented in part b of the figure coming from a dinophyte. One wonders why the

genome is so large (much larger than any reference dinophyte plastome) – is it because of an inflation of intragenic regions, and if yes, what kind of sequences makes this extra stuff? How certain the authors are the assembly is correct and does not include sequences that do not belong to the plastome?

–Response: This is a fair point. We manually checked the three largest ptMAGs, ranging in size from 627,879 to 181,762 bp, and all belonging to the same clade of dinoflagellates with plastids of green algal origin. In particular, we focused on the largest ptMAG, TARA_CHLORO_00662, composed of 11 contigs. While the possibility of assembly artefacts can never completely be ruled out, there are multiple indications that TARA_CHLORO_00662 is a legitimate ptMAG.

First, we present the figure below which shows TARA_CHLORO_00662 (original name: CHL_PSW_Bin_87_1_c) in the context of its detection across *Tara* Oceans metagenomes. Tips of the dendrogram represent contigs of the ptMAG while the rings represent different metagenomes with thick black bars indicating high abundance. The figure shows that all contigs of the ptMAG have perfect abundance correlation across *Tara* Oceans metagenomes, making us highly confident that the contigs have been correctly binned together. We observed similar patterns for the other large ptMAGs in our dataset.

Second, plotting the genome map of the ptMAGs with OGDRAW allowed us to visualise the large proportion of intergenic space in nearly *all* contigs. Please note that the scale is different for every contig in the figure below.

Third, while constructing the phylogenomic dataset with 93 genes, we inferred single gene trees (SGTs) including TARA_CHLORO_00662. The sequences stemmed from eight out of the 11 contigs and consistently clustered with green-coloured dinoflagellates in all trees, further increasing our confidence in this ptMAG.

Finally, we zoomed out to the entire clade of green-coloured dinoflagellates and mapped size, number of genes, and proportion of genic to intergenic regions on the corresponding phylogeny (figure below). We observe a large variation in plastid genome size that is also accompanied by an inflation of intergenic regions and to some extent, by an increase in number of genes (mostly unidentified orf genes). That there are several large plastid genomes within the same clade, and not scattered randomly in the algal tree, indicates that these ptMAGs have captured genuine biological signal.

As to the question of what sequences make up this intergenic region, we checked for the presence of repetitive regions in the TARA_CHLORO_00662 ptMAG by generating a dot plot with ModDotPlot (<https://doi.org/10.1093/bioinformatics/btae493>). The figure below shows that repetitive elements are not the major contributor to this genome size expansion. A more thorough exploration would be better suited to a future manuscript focused specifically on this lineage.

Self-Identity Plot: CHL_PSW_Bin_87_1_c_000000000001

We have now added a new figure (Supplementary Fig. 9) to elaborate on this unusual clade of ptMAGs.

- Another particular point I believe the authors should exert more effort to illuminate the nature of the clade of eight chlorophyte ptMAGs that form a separate clade perhaps related to Chloropicophyceae (shown in detail in Supplementary Fig. 8a). Indeed, there are at least two deeply diverged “prasinophyte” clades known from environmental amplicons of the 18S rRNA gene but not yet represented by cultivated members, called Clade VIII and Clade IX by Viprey et al. (<https://enviromicro-journals.onlinelibrary.wiley.com/doi/10.1111/j.1462-2920.2008.01602.x>). One would speculate that the aforementioned ptMAG clade most likely comes from the same organisms as the 18S rRNA amplicons assigned to Clade VIII or Clade IX, and it would be interesting to test this hypothesis directly by looking for the respective 18S rRNA gene sequences in the metagenomic samples enriched for plastome sequences of the clade under question. Matching the ptMAG clade to a previously known 18S rRNA clade would be an important step forward in charting the green algal phylogenetic diversity. An analogous analysis could be done for some of the ochrophyte ptMAGs, such as those related to pelagophyte plastomes (Supplementary Fig. 8b) that occupy a position equivalent to some of the uncultivated MOCH (marine ochrophyte) lineages in the 18S rRNA tree (see Fig. 6 in Terpis et al. 2025, [https://www.cell.com/current-biology/abstract/S0960-9822\(24\)01632-4](https://www.cell.com/current-biology/abstract/S0960-9822(24)01632-4)).

–Response: We agree that linking novel ptMAG groups to 18S rDNA sequences known only from environmental amplicons is important. However, as explained in the response to the previous comment, simply looking for 18S rDNA sequences in targeted metagenomic samples would be insufficient to link ptMAGs with 18S rDNA sequences. A strong abundance correlation between a ptMAG and the corresponding putative 18S rDNA sequence is also required.

We did an exploratory analysis to try and match an 18S rDNA sequence to the most abundant ptMAG (TARA_CHLORO_00434) from the clade of chlorophyte ptMAGs in question. The analysis was carried out as described in the previous comment, this time using relative abundances of ASVs and the ptMAG across 599 *Tara* Ocean samples. However, the ASV with the highest correlation coefficient (0.566) belonged to a haptophyte.

This analysis along with our previous attempt to find a putative leptophyte 18S rDNA sequence demonstrate that linking organellar genomes, which can vary in copy number with varying conditions, with nuclear sequences is not trivial. We therefore believe that more thorough attempts to link ptMAGs with nuclear sequences are outside the scope of this manuscript.

- One of the general incentives behind exploring hitherto missing branches of the algal tree of life is to eventually resolve the puzzling evolutionary history of higher-order plastids, particularly those originating from red algae. One of the points that is controversial – or one would say somewhat unexpectedly became controversial only recently with the study by Pietluch et al. (2024; cited by Jamy et al. as ref. #55) – is whether the different “chromalveolate” plastids are monophyletic to the exclusion of extant red algal lineages or, as proposed by Pietluch et al., evolved diphyletically from different red algal ancestors. The improved sampling and especially the addition of leptophyte plastomes delivered by Jamy et al. creates a dataset that should be more informative on this contentious issue than any dataset analysed before. Another salient aspect of the analyses presented by Jamy et al. is the use of the highly sophisticated methodology of phylogenetic inference, including the most recently developed

complex substitution models. Hence, one would think that the present study will say something significant regarding the origins of the complex red plastids. Thus, I encourage the authors to elaborate a bit more on the general implications of the new data for the reconstruction of the plastid history. Indeed, they have already considered the alternative scenario suggested by Peitluch et al. in their analyses, but what I am missing is a more explicit discussion on the results. Do their analyses illuminate the question of monophyly/non-monophyly of complex red plastids and if yes, which of the two possibilities is more credible? A reader would appreciate some discussion.

–Response: The reviewer raises a valid point. Whether complex red plastids are monophyletic or non-monophyletic is indeed a fundamental unanswered question.

One reason we initially refrained from a discussion of this issue was due to concerns about the methodology presented in the study by Pietluch et al. (2024). Specifically, their analyses did not employ site-heterogeneous mixture models which are essential to mitigate long-branch attraction artefacts. In addition, their inclusion of the *rpl36* gene, and the extremely fast-evolving colpodellid lineages likely introduced further sources of bias. Given these limitations, we felt it was premature to draw strong conclusions based on their results, and therefore did not delve into this question.

That said, our own analyses generally recovered the monophyly of complex red plastids with varying levels of support (summarised below):

- CAT-GTR phylogeny: monophyly of complex red plastids (PP=0.99)
- cpREV+C60+G phylogeny: Rhodophytina sister to CHL group (UFB=81)
- CAT-PMSF phylogeny: monophyly of complex red plastids (BS=72)
- LG+MEOW80+G phylogeny: monophyly of complex red plastids (UFB=68)
- CAT-GTR+G on SR4-recoded dataset: monophyly of complex red plastids (PP=0.98)
- LG+MEOW80+G on stationary trimmed dataset: monophyly of complex red plastids (UFB=72)
- CAT+GTR+G on stationary trimmed dataset: monophyly of complex red plastids (PP=0.99)

Furthermore, Supplementary Table 3 shows that trees constrained to enforce monophyly of complex red plastids monophyletic consistently had higher likelihoods than those with non-monophyly of complex red plastids.

Nevertheless, the lack of strong support for complex red plastids with maximum likelihood analyses means that uncertainty remains and the question of the origin of ochrophytes is still open.

We have now added the following text to the discussion to address the reviewer’s point (lines: 413-416):

“More generally, our advanced phylogenetic analyses and broad taxon sampling continue to recover—albeit with only moderate support—the monophyly of all red algal-derived plastids, unlike in a recent proposal that argues for a separate secondary plastid acquisition in ochrophytes⁴⁶. The origin of the ochrophyte plastid is a pressing issue to address in the future”

- The authors should double-check the ptMAGs they report in their study. As it became clear only today (thanks to a student of mine looking at the data), at least in one case the reconstructed ptMAG includes

sequences that clearly do not belong to a plastid genome. Specifically, TARA_CHLORO_00069 contains a contig (CHL_AON_Bin_218_19_c_000000000001) that based on the annotation provided by the authors themselves corresponds to a part of a mitochondrial genome (not the presence of genes *atp1* and *cob* that have never been seen in plastomes but are typical for mitochondrial genomes). A systematic check of the whole set is needed to identify possible similar cases and the readily detectable contaminating sequences should be removed from the ptMAGs.

-Response: Thank you so much for pointing out this contamination, which we had not detected earlier. We have now systematically checked all non-redundant ptMAGs for mitochondrial contamination which confirmed that TARA_CHLORO_00069 but also four other ptMAGs were contaminated. We have removed the contamination and updated the associated Figshare repository. We also added the following text to the methods section (lines 500-506):

“Check for mitochondrial contamination. To assess potential mitochondrial contamination, all ptMAGs were screened for the presence of 25 canonical mitochondrial genes (Supplementary Table 6), as listed in ⁵⁵. Five non-redundant ptMAGs were found to contain genes annotated as typical mitochondrial genes, which were confirmed by BLAST searches. Further manual inspection indicated that these instances of contamination were not due to chimeric misassemblies. Instead, they resulted from inadvertent co-binning of mitochondrial and plastid contigs. The contaminated contigs were subsequently removed from their respective ptMAGs.”

Various additional issues:

- line 56: should there be a comma before “such as diatoms”?

-Response: Yes, added!

- line 187: instead of “retaining the core components” please write “retaining genes for the core components”, as what is retained in the plastid genomes are genes, not the actual components encoded by them.

-Response: Done.

- lines 188-189: in relation to the previous note, use italics when mentioning the various plastid genes. The same for “*rpl36*” in lines 212 and 310.

-Response: Thank you, fixed.

- line 226: “were less than four points difference” – I think the grammar is perhaps not correct here, please check

-Response: Fixed (the phrase now reads “had less than four points difference”).

- line 232: “show in that order” – the phrase sounds strange to me, I would expect something like “are shown in the following order”, but I am not a native English speaker.

-Response: We have now changed the phrase to “Branch support values are given in the following order:”

- line 235: delete the hyphen from “rpl-36” for the sake of consistency

-Response: Done!

- line 284: “c-type paralog of the ribosomal gene rpl36” – I have two issues here. First, while being aware that the term “paralog” was used already in the original report on the rpl36 gene replacement by Rice and Palmer (2026), I don’t think the term is used properly in this case. By definition, paralogs are genes related via a gene duplication event, so to claim the c-type and p-type rpl36 versions being paralogous one would have to provide evidence they are descendants of two different copies of a single duplicated gene, with the duplication event having happened somewhere at a deep branch of the eubacterial phylogeny. However, no such evidence has been provided (by Rice and Palmer or the authors of the present manuscript) and presently it is completely unknown how the c-type and p-type rpl36 versions evolved. Hence, I would avoid using the term “paralogs” and use something not implying a specific evolutionary event, e.g. “forms” (check also line 326). Second, in my opinion rpl36, rather than being a “ribosomal gene”, is a “ribosomal protein gene”.

-Response: Thank you for the detailed explanation. This point was also raised by Reviewer 3. In response to this comment, we investigated whether the *rpl36* gene has undergone duplication during eubacterial evolution. Previous studies have indeed found evidence for *rpl36* duplication in bacteria (<https://doi.org/10.1186/gb-2001-2-9-research0033>, <https://doi.org/10.1128/jb.01901-06>), with several bacterial genomes containing both paralogs. These paralogs are designated C+ (containing the cysteine zinc ligands, similar to the *rpl36*-p type gene in plastids) and C- (cysteine zinc ligands absent, as in the *rpl36*-c type gene in plastids). Given this information, we think it is appropriate to use the term “paralogs” in the manuscript.

We now refer to rpl36 as a “ribosomal protein gene” in the manuscript.

- line 301: “limited information of plastid genomes” – would it be more appropriate to write “limited information content of plastid genomes”?

-Response: We have now rephrased the phrase to “limited phylogenetic signal of plastid genomes”.

- line 433: “IQTree” – I think this is not how usually the name of the program is spelled; compare also with a different (correct?) spelling, i.e. “IQ-TREE”, at other places in the manuscript.

-Response: Thank you for pointing this out. We now use “IQ-TREE” throughout the manuscript.

- lines 510, 512, 524: there is a problem with correct printing of the Greek letter “chi” here. Note also that the manuscript is inconsistent in that at other places the alternative writing “chi-squared” (e.g., line 239) is used. I would unify this across the whole text.

-Response: We are not sure why the symbol for chi-squared (χ^2) was rendered incorrectly. But we now use “chi-squared” throughout the text to avoid further formatting problems and to be consistent.

- line 545: “16S phylogeny” – I think it is inappropriate to omit “rRNA” or “rDNA” from the name of the locus used as the phylogenetic marker, so please, use “16S rRNA” or “16S rDNA” (the latter may be the best option, as it is the DNA sequence that is de facto used) in the whole text (including Supplementary Fig. 12). Analogously, write “18S rRNA” or “18S rDNA” where appropriate (line 597 and the whole following paragraph).

-Response: Fixed throughout the text!

- line 562: “the rpl36 amino acid sequences” – technically this is not correct. While there is a gene name “rpl36” (in italics), the respective protein product is called L36.

-Response: The reviewer is correct, and we have now corrected this mistake.

- line 583: “We identified HMM psbO sequences” – I don’t understand what is a “HMM psbO sequence”, perhaps delete “HMM” from the sentence

-Response: Done!

- update ref. 30 (Williamson et al.), the preprint has now been published in a journal (Nature)

-Response: Done!

- delete “CB” from the journal title in ref. #54 (Ponce-Toledo et al. 2017)

-Response: Done!

- fix the journal abbreviation in ref. #67 (Yu 2020), the last word correctly is “Bioinform.”

-Response: Done!

- Fig. 1: while I do see using names for various taxa is always to some extent arbitrary, I would still consider to improve the consistency of the nomenclature by changing some of the names. Thus, “Euglenids” is a name referring to a broad euglenozoan lineage that includes a paraphyletic assemblage of aplastid taxa plus a single plastid-bearing (i.e. algal) subclade, usually called Euglenophyceae. Hence, to stay with names of algal taxa when labelling the different branches of the plastid genome tree, I would write “euglenophytes” instead of “Euglenids”. It would also be more consistent with the overall nomenclatural style to write “Rhodophytes” instead of “Red algae” (in fact, “Rhodophyta” is used to label the respective taxa in Supplementary Fig. 7 showing the full version of the tree). Next, the labeling of a particular sector of the tree with “Chrysophytes + Brown algae” is somewhat imprecise, as this part of the tree includes plastid genomes from additional ochrophyte taxa, including raphidophytes or pinguiophytes (note that TARA_CHLORO_00423 in fact comes from a pinguiophyte, which is not obvious from the tree presented by the authors for their omission of pinguiophyte references). One way how to solve this would be to simply write “other ochrophytes” instead of listing

all the constituent taxa here. Finally, the sector of the tree labelled “Diatoms” in fact includes also bolidophytes, and this should be reflected in the tree. I think it would be useful to add a note in the legend to the figure that a full version of the tree is provided in Supplementary Fig. 7 (in analogy with a remark at the end of the legend to Fig. 2).

- taxon names for some of the reference plastomes: I understand it is very difficult to keep track with the taxonomic updates regarding sequence data in deposited in databases under outdated or inconsistently applied names, but still I think it’s important to try to be as accurate as possible. Thus, some of the taxonomic names of the reference plastomes included by the authors should be updated throughout the manuscript (including all tables and figures). Here are some examples I have spotted upon a quick inspection of the list:

REFG_CHLORO_00059 – what is referred as “Chromerida sp. RM11” has been described in 2012 as “*Vitrella brassicaformis*” (<https://www.sciencedirect.com/science/article/pii/S1434461011000939>)

REFG_CHLORO_00030 – what is referred as “Cryptophyta sp. CCMP2293” has been described as “*Bafinella frigidus*” (<https://onlinelibrary.wiley.com/doi/full/10.1111/jpy.12766>)

REFG_CHLORO_00136 – what is referred as “*Dictyocha speculum*” has been reclassified as “*Octactis speculum*” (<https://onlinelibrary.wiley.com/doi/full/10.1111/pre.12181>)

REFG_CHLORO_00009 – what is referred as “*Nannochloropsis gaditana*” has been reclassified as “*Microchloropsis gaditana*” (<https://www.tandfonline.com/doi/full/10.2216/15-60.1>)

REFG_CHLORO_00053 – what is referred as “*Pyconococcus provasolii*” should be named “*Pseudoscourfieldia marina*”, as the former is a junior synonym of the latter (<https://onlinelibrary.wiley.com/doi/full/10.1111/jpy.13482>)

REFG_CHLORO_00012 – is there a reason why the species name “*Prasinococcus capsulatus*” (<https://www.tandfonline.com/doi/full/10.1080/23802359.2019.1698370>) should not be applied to what the authors presently call “*Prasinococcus* sp. CCMP1194”?

REFG_CHLORO_00143 – the organism is referred to as “*Pavlova* sp.” (Supplementary Table 1, Supplementary Fig. 7) or as “*Pavlova* sp. NIVA-4/92” (Fig. 2 and several supplementary figure), but in fact it has been identified as the species *Diacronema lutheri* (<https://academic.oup.com/gbe/article/13/8/evab178/6337978>) as is thus taxonomically redundant with the second plastome from this species included in the dataset by Jamy et al. (NC_020371). Unsurprisingly, these two plastomes branch together in the trees presented in the paper, separate by a zero branch length.

REFG_CHLORO_00021 – while the correct name “*Pavlomulina ranunculiformis*” is used by the authors in most trees, the same organism is labeled as “*Haptophyceae* sp. NIES-3900” in Supplementary Fig. 7 and Supplementary Table 1, so this discrepancy should be fixed, too.

–Response: Thank you very much for pointing these out. We have updated the taxon names of these references in Supplementary Figs. 7, 13, 17-20, Supplementary Table 1, Figs. 2-3, and the trees provided on Figshare.

- Fig. 3: Frankly, I do not know how to understand the meaning of the blue bars in part c of the figure, which are explained in the legend as follows: “Difference in log-likelihood scores between each topology and the best-scoring topology is shown by data bars in blue”. At face value the bar sizes do not really intuitively follow the numbers indicated in the table. Most likely I am missing something, but the authors could try to provide an explanation of the data that is easier to understand.

–Response: We apologise for the confusion, and acknowledge that the data bars may not be immediately intuitive to understand. This panel of Fig. 3 follows the same idea as Fig 2 of Williamson et al 2025 (<https://doi.org/10.1038/s41586-025-08709-5>). Briefly, the data bars are scaled by the difference in log-likelihood scores of the best and worst scoring topologies. The worse-scoring the topology, the longer the data bar in blue, allowing the reader to visually assess which topology is the worst and best scoring (the best-scoring topology is additionally highlighted in bold).

- Fig. 4: I am not sure why the cryptophyte branch in the left part of the figure and the haptophyte branch in the right part of the figure are drawn in colour along their whole length. I would think it makes sense to start colouring the branch from the point of plastid gain to the tip, otherwise the meaning of the branches in colour is elusive to me. The same concern holds to Supplementary Fig. 20.

–Response: This is a good point. We have updated the figure (now Fig. 5) so that the branch colour reflects plastid gain. The original Supplementary Fig. 20 has now been removed, as we no longer consider alternate plastid transfer scenarios probable given the position of the putative leptophyte sequence in the mitochondrial phylogeny.

Reviewer #3 (Remarks to the Author):

New deep-branching environmental plastid genomes on the algal tree of life

Mahwash Jamy¹, Thomas Huber², Thibault Antoine³, Hans-Joachim Ruscheweyh⁴, Lucas Paoli⁵, Eric

Pelletier³, Tom O. Delmont^{3*}, Fabien Burki^{2*}

The authors have mined Tara Oceans data to assemble and curate MAGs corresponding to plastid genomes, with a focus on those derived by higher-order endosymbioses involving red algal donors. It is a remarkable paper. The authors do not over-state its conclusions. To my knowledge it is indeed the most extensive survey of plastid genomes in this way, and they uncover an incredible wealth of information, including discovery of a new algal lineage, which they reasonably assign the following informal name: leptophytes.

The Introduction accurately covers the history of this complex field and is well cited in this regard.

One could argue that, in the complete absence of any knowledge of a host for the leptophyte plastid, it is too soon to be proposing evolutionary scenarios about how this organelle came to have a genome

that looks like it does. My first reading was exactly this. I was wanting more description of the host lineage search in the main text (not passed on to the Methods, which was done on two occasions). But having read the discussion to the end, I can see why the authors are comfortable floating two hypotheses, both with backbone trees that are clearly labeled as hypothetical. Any way one looks at these data, there are still a lot of gaps to fill in our knowledge; all we know for certain now is that this leptophyte plastid lineage clearly puts it in the haptophyte / cryptophyte part of the plastid tree (based on multiple lines of evidence), but it is nevertheless highly distinct. And so we can reasonably say that whatever the host is, it too is rather distinct from the nuclear component of these algae. And if not, well, that would be really interesting too.

To conclude my general comments, this paper has a level of novelty that deserves publication in a high-impact journal – it points to the existence of an entirely new branch of photosynthetic life. That is a rare and exciting find. I have the following specific suggestions for the authors to consider.

–Response: We sincerely thank Prof. Archibald for his encouraging remarks and constructive suggestions. We have addressed each point below, which has strengthened the manuscript.

Specific comments:

-I suggest the authors add some ‘landmark’ taxa on Fig 1 tree. It is not easy to align the gray ‘reference’ OTUs on Ring 2 with the actual branches (which in most cases are quite far away). So why not provide a few well-known taxon names on the actual branches, e.g. *G. huxleyi*, *Phaeocystis*, *Thalassiosira*, *Guillardia*, *Aureococcus*, etc., where there is space. This would help experts pinpoint their favourite groups and see how much new diversity the new paper adds to that particular part of the tree. If done strategically, these additions would not detract from the larger take-home messages.

–Response: Striking the right balance for Fig. 1 between informativeness and visual clarity has been challenging. We went through six iterations of the figure where we experimented with the layout, ring selection, and other elements, before settling on the current version in the manuscript, which we believe best conveys the take-home messages without overwhelming the reader.

We agree that it is difficult to match specific branches with the grey references on Ring 2. However, our intention is not for the reader to trace each reference individually, but to grasp the broader patterns: e.g. ptMAGs have substantially expanded marine haptophyte plastid diversity, but not so much in cryptophytes. We considered your request to add the names of several well-known taxa on the branches, but we feel that while it may aid readers in one way, it will also reduce figure clarity in other ways. For one, it is difficult to add these labels to specific branches (which are rather tiny given the size of the tree) without cluttering the figure. It may also raise questions of why some references are labelled, and others are not.

For these reasons, we have opted not to add taxon names to any references in Figure 1, unless the reviewer strongly prefers otherwise. We refer readers to Supplementary Fig. 7 which shows the full rectangular phylogeny with taxon labels. We have now added a sentence to the Fig. 1 caption: “The full phylogeny with taxon labels is shown in Supplementary Fig. 7”. Additionally, all phylogenies are also provided as newick files in the Figshare repository linked to the manuscript.

-Results – first paragraph. The authors discuss ANI and completeness, which makes sense, but it might be good to also include depth of MAG coverage. This is brought up later when discussing abundance of the leptophytes, but non-specialists (used to sequencing genomes ‘normally’ and not assembling them from metagenomic data) would appreciate getting a sense of the range of depth of sequence coverage in the diversity of ptMAGs.

–Response: This is a good point, and we have now added a sentence on the range of sequencing depth of the ptMAGs (line 114-117):

“The ptMAG abundance, as estimated by sequencing depth, varied by more than four orders of magnitude, ranging from a chlorophyte ptMAG with ~21,000× coverage to a diatom ptMAG with only ~2× coverage (Supplementary Table 1).

-Figure 2. What is the tree topology on the left based on? The legend says a larger version is shown in Supp. Fig 15, but that figure looks to be the exact same as Figure 2 (except for presence of gene names) and doesn’t say what the tree at left is based on either. The legend refers to UPGMA clustering, but which tree, the one on the left or the one on the top? These points need clarification.

–Response: We apologise for the confusion. Both dendrograms of the genes and the taxa are based on UPGMA clustering and we have now updated the caption of Fig. 2 as follows:

“A binary heatmap and dendrograms of genes and taxa generated using the UPGMA algorithm, based on the presence or absence of 237 plastid protein-coding genes. Blue and grey boxes represent gene presence and absence, respectively. For leptophytes, present genes are highlighted in red for improved clarity. A larger version of this figure, including gene names, is presented in Supplementary Fig. 16.”

As noted, Supplementary Fig. 16 is identical to Fig. 2 but includes gene names, which could not be added to the main figure in a readable font due to space constraints.

Lines 135-137 – “In multiple cases, the ptMAGs represented novel genome diversity...” As written, this statement undersells the significance of the data and is not in line with the novelty conveyed in the abstract. Assuming that the gray reference sequences were chosen to represent the full breadth of sequence diversity from cultured representatives, then I would say that novel genome diversity is massively represented in the tree shown in Figure 1. I guess it depends on what is meant by ‘novel diversity’, but looking within haptophytes, diatoms and pelagophytes, the author’s have recovered a tremendous number of novel genomes. I suggest they highlight this even more in the main text, to be more consistent with the (accurate) description in the abstract.

–Response: In the original sentence “In multiple cases, the ptMAGs represented novel genome diversity”, we were referring to deep-branching clades composed only of ptMAGs and no references. However, we agree that the description in the text is not consistent with the one in the abstract and we have now updated the main text to better convey the extent of novel diversity:

Line 141-143: “Conversely, the ptMAGs were affiliated to ochrophytes (n=377, including 305 diatoms displaying a very broad cellular size range), and haptophytes...”

was changed to:

“Conversely, the ptMAGs greatly expanded the known genome diversity of ochrophytes (n=375, including 304 diatoms displaying a very broad cellular size range), and haptophytes ...”

Line 146-148: “In multiple cases, the ptMAGs represented novel genome diversity within established algal groups where cultured references are lacking.”

Was changed to (*italics added here for emphasis*):

“In multiple cases, the ptMAGs represented *deep-branching* novel genome diversity within established algal groups where cultured references are lacking.”

-Lines 165-66: “However, efforts to identify potential hosts for Lepto-01 at stations where its signal was strongest were unsuccessful (see Methods).” As noted above, I think the authors would do well to briefly integrate some of this information into the main text. The MS is very short (like an old Letter to Nature) and thus there should be plenty of room. And furthermore, the authors could touch on why it matters what the host is, as a prelude to the Discussion.

–Response: We agree that the original description in the main text was too brief, and this opinion was also expressed by Reviewer 2. We have now greatly expanded our description of the search for the host in the main text (lines 277-317). This is thanks, in part, to a new exciting finding: a MAG likely corresponding to the mitochondria of Lepto-01. Briefly, this mitochondrial MAG has near-perfect abundance correlation with the Lepto-01 ptMAG across *Tara* Ocean metagenomes, *and* branches separately from established clades in the mitochondrial phylogeny (sister to all haptophytes) (see the new Fig. 4). This result has large implications for possible scenarios for plastid transfer that we deem likely, and we have accordingly updated the discussion and Fig. 5.

-Line 196: present in all other algal groups shown, or all other algal groups (inc. green and glaucophyte algae)?

–Response: This is a good point. Most of these genes are also absent in green algae. We have now updated the sentence to clarify that these genes are “mostly present in all other algal groups with plastids of red-algal origin”

-Line 284: Paralog is not the right word to refer to the c-type rpL36 in the CHL plastid because it doesn't (obviously) involve gene duplication. Probably xenolog is better (which I don't see much and am not a fan of). I think the evidence suggests that rpL36 in CH(and now L) is most likely an example of orthologous replacement (i.e., a bacterial ortholog replacing the canonical plastid copy). Whether it is, strictly speaking, an ortholog (i.e., evolved by sequence divergence post speciation) is unclear.

–Response: A similar comment was made by Reviewer 2 and we copy-paste the response given to them here:

“Thank you for the detailed explanation. This point was also raised by Reviewer 3. In response to this comment, we investigated whether the *rpl36* gene has undergone duplication during eubacterial evolution. Previous studies have indeed found evidence for *rpl36* duplication in bacteria (<https://doi.org/10.1186/gb-2001-2-9-research0033>, <https://doi.org/10.1128/jb.01901-06>), with several bacterial genomes containing both paralogs. These paralogs are designated C+ (containing the cysteine zinc ligands, similar to the *rpl36*-p type gene in plastids) and C- (cysteine zinc ligands absent, as in the *rpl36*-c type gene in plastids). Given this information, we think it is appropriate use the term “paralogs” in the manuscript.”

-Line 337-340. “While we found ample evidence of cryptophyte nucleomorphs elsewhere, we found no evidence of putative leptophyte nucleomorphs”. What does ‘elsewhere’ mean here, exactly? As noted above, I think there would be space to integrate the methods information on how they searched for the leptophyte 18S rRNA gene sequences. This would be good for completeness and would help nonspecialist readers better understand the nuances of the alternative models presented in Fig. k4.

–Response: Our description of the search for putative leptophyte nucleomorphs is brief here, because a more thorough description is part of a sister-study—not yet available as a preprint—focusing on nucleomorph diversity in the sunlit oceans. In this sister-study, we have characterised a major new clade of cryptophyte nucleomorphs (unpublished data). We have modified the sentence slightly to remove the word “elsewhere”, which we agree sounded very vague in this context.

We have now included in the main text a brief description of the (unsuccessful) search for leptophyte host sequences based on the 18S rDNA and psbO genes (lines 279-286). But, as noted earlier, we included a more detailed description of the search for the putative leptophyte mitochondria sequences (which we successfully retrieved), and its implications for alternate models of plastid transfer.

Signed: John Archibald

Reviewer #1 (Remarks to the Author):

The revised manuscript provides significantly improved results and discussion, strengthening its overall impact. Based on the authors' claims in the response to the comments, I no longer doubt its suitability. However, I would still like to add comments on a few points.

I previously noted that a major weakness of this manuscript is the lack of validation for the proposed phylum-level informal grouping of the so-called "leptophyceae", even though the authors' tremendous effort to compile a resource database of ~700 plastid genomes from marine pelagic ocean metagenome. In their response letter, the authors emphasized the considerable difficulty in locating putative leptophyte species with the available genome sequence through FISH etc. and underscored the practical challenges of setting up long-term experiments. I fully understand this limitation. But there still seems to be a reluctance to embrace the scientific proposition of leptophytes, which might stem from the absence of "biological" characterization for this new lineage. I think we have to meet halfway to make this study more meaningful with the current metagenomic dataset.

To that end, I suggest the authors make fuller use of metagenome data to provide environmental and ecological context for leptophyceae by:

- Response: We are thankful to the reviewer for their continuous effort trying to improve the manuscript by complementing our evolutionary-centric study with more visual and ecological insights. As described below, we have completed all the analyses the reviewer suggested regarding (1) correlation with environmental variables, and (2) correlation with other planktonic populations. However, those additional analyses provided limited insights. First, only expected environmental variable correlations were found (e.g., negative correlation between Arctic leptophytes and temperature). Second, the few good correlations with other planktonic populations (e.g., few plastid genomes of haptophytes) only provide very weak interpretations about putative common niches, which we consider would only dilute (no real gain) the main results of our study if integrated extensively into the main text.

We fully understand the reviewer's initial motivation to see (FISH), or at least better understand fundamental ecological aspects of the leptophytes. We certainly share the same interest. We agree that characterizing the biology (e.g. morphology, cell biology, genetics, and behaviour) of the leptophytes are the critical next steps, and to that purpose as stated in our previous response letter, the Burki lab has recently hired a postdoc to isolate and characterize the new lineage over a planned time period of 3 years. We are not reluctant to embrace the high scientific significance of leptophytes, on the contrary, but indeed we have argued before, and continue to do so, why we refrain for now to propose any formal definition until more data are obtained—for example to know whether the leptophytes are a class of haptophytes or deeper than that. We believe that we have carefully drafted the main text so that we only informally refer to the new clade as the leptophytes, without any formal taxonomic ranking of the proposed name.

This is the relevant text in the discussion:

“Both models assume a sister relationship of leptophytes to haptophytes in the eukaryote tree based on our mitochondrial phylogeny. Whether this means that haptophytes should be expanded to include leptophytes as a new deeply diverging class, or that leptophytes is a distinct but related lineage, will require detailed morphological information (such as the presence of a haptonema) not available at the moment.”

Thus, we do not propose that leptophytes correspond to a new phylum, nor do we use the term “leptophyceae”. We consider that we found a good balance in the main text to clearly show that leptophytes correspond to a novel important deep-branching clade positioned in the close vicinity of haptophytes (both with the plastid genome and the mitochondria genome).

More specifically regarding the reviewer’s suggested analyses:

1) Estimating their relative abundance and co-occurrence with other species groups using metagenome reads or assemblies.

Response: We thank the reviewer for suggesting the additional analyses to place leptophytes in an ecological context. We followed the reviewer’s suggestion and inferred pairwise correlations between the leptophyte ptMAGs and all other plastids, along with nuclear eukaryotic genomes previously characterized from the same metagenomic co-assemblies (Delmont et al 2022; <https://www.sciencedirect.com/science/article/pii/S2666979X22000477>). For each pairwise comparison, we calculated Pearson correlation coefficients based on the mean coverage of each genome across the 1,178 TARA Oceans metagenomes. A genome was considered present in a sample only if at least 25% of its length was covered by sequencing reads (a cut-off commonly used, including in the publication on the nuclear eukaryotic genomes). We considered taxa to be co-occurring when their correlation coefficient was ≥ 0.6 .

Results are summarized in the table below:

Pearson correlation coefficients for leptophyte MAGs.

Only pairs with a Pearson correlation of at least 0.6 are presented

Genome_identifiers	Genome type	Taxonomic group	Pearson_Lepto-01_REFM_CHLORO_00001	Pearson_Lepto-02_TARA_CHLORO_00332	Pearson_Lepto-03_TARA_CHLORO_00478	Pearson_Lepto-04_TARA_CHLORO_00158
TARA_CHLORO_00478	Plastid	Leptophyte	-0.008818691	-0.022634309	1	0.820673451
TARA_CHLORO_00158	Plastid	Leptophyte	-0.024632998	-0.022560992	0.820673451	1
TARA_CHLORO_00188	Plastid	Dictyochophyceae	-0.0200355	-0.018350213	0.601506127	0.677533192
TARA_EUK_00374	Euk. Nuclear	Pedinellales	-0.020372867	-0.018659203	0.564771719	0.66350414
TARA_EUK_00122	Euk. Nuclear	Pedinellales	-0.021088811	-0.019314925	0.555936205	0.675454045
TARA_EUK_00252	Euk. Nuclear	Pedinellales	-0.019562886	-0.017917353	0.541997119	0.667497137
TARA_CHLORO_00501	Plastid	Dictyochophyceae	-0.028918052	-0.026513801	0.540944908	0.708290228
TARA_CHLORO_00591	Plastid	Prymnesiales	-0.019463658	-0.020492436	0.508251496	0.650172105
TARA_CHLORO_00189	Plastid	Dictyochophyceae	-0.025176554	-0.023058827	0.477568886	0.632048927
TARA_EUK_00542	Euk. Nuclear	Acanthoecida	-0.018482767	-0.016928088	0.426220257	0.613666286
TARA_CHLORO_00048	Plastid	Prymnesiales	0.0186446	0.602271525	0.157319516	0.21580196
TARA_CHLORO_00240	Plastid	Prymnesiales	0.678734171	0.285249447	0.098526203	0.063482135
TARA_CHLORO_00473	Plastid	Prymnesiales	0.614682542	0.272183295	0.086553269	0.045782555
TARA_CHLORO_00262	Plastid	Prymnesiales	0.86623549	0.341867596	0.037419935	0.02029644
TARA_CHLORO_00255	Plastid	Prymnesiales	0.881715092	0.325456731	0.007887972	0.004919869
TARA_CHLORO_00224	Plastid	Prymnesiales	0.887976377	0.329225782	0.00604388	0.005523949
TARA_CHLORO_00230	Plastid	Prymnesiales	0.198931149	0.604047471	0.002104115	-0.009837511
TARA_CHLORO_00263	Plastid	Prymnesiales	0.687052089	0.313660177	-5.52448E-05	-0.00436595
TARA_EUK_00238	Euk. Nuclear	Prymnesiales	0.647237597	0.251732511	-0.13323275	-0.021740374
REFG_CHLORO_00067	Plastid	Euglenida	0.035398127	0.808880611	-0.016648321	-0.016594394
TARA_EUK_00166	Euk. Nuclear	MAST_unclassified_Clade	0.746621267	0.237757005	-0.017266762	-0.017210832
TARA_CHLORO_00225	Plastid	Prymnesiales	0.051543855	0.721079803	-0.017502245	-0.017445552
TARA_CHLORO_00229	Plastid	Prymnesiales	0.043979923	0.678241824	-0.020665301	-0.020598363
TARA_CHLORO_00256	Plastid	Prymnesiales	0.836717245	0.2846101	-0.020939235	-0.020871409
TARA_EUK_00160	Euk. Nuclear	Oomycota	0.668096733	0.20686192	-0.024904062	-0.024823394
TARA_CHLORO_00272	Plastid	Euglenozoa	0.779347225	0.286363996	-0.025006516	-0.024925515
TARA_CHLORO_00268	Plastid	Chromulinales	0.638149714	0.206417336	-0.026139377	-0.026054707
TARA_CHLORO_00253	Plastid	Prymnesiales	0.627462546	0.190048788	-0.028578289	-0.028485719
TARA_CHLORO_00315	Plastid	Mamiellophyceae	0.773184465	0.275301293	-0.036847996	-0.036728638

In the table, “Genome type” differentiates ptMAGs from nuclear eukaryotic genomes (“Euk. Nuclear”), while “Taxonomic group” provides high-ranking taxonomic information of the environmental genome.

Briefly,

- the non-polar ptMAGs, Lepto-03 and Lepto-04 were found to correlate with each other, and a few dictyochophyceae plastid and nuclear genomes.
- Lepto-02 (polar) co-occurred with a euglenophyte plastid and four haptophyte plastids.
- Lepto-01 was found to co-occur with 14 other genomes, including nine affiliated with haptophytes, one with a chlorophyte, one with a euglenophyte, one with a MAST lineage, and one with an oomycete.

As often with these kinds of analyses, there are many possible interpretations. The co-occurrence of multiple plastids with Lepto-01 could reflect a shared or similar ecological niche. The co-occurrence of the oomycete and MAST lineage with Lepto-01 is intriguing as this could suggest infection of leptophytes by the oomycete, or a predator-prey relationship between Lepto-01 and the MAST lineage, but this is highly speculative. Finally, the lack of *strong* correlation with any nuclear eukaryotic MAG (as expected) confirms that the leptophyte nuclear genome is not in our database of MAGs.

As the insights provided by this co-occurrence analysis are largely speculative and in our opinion not very informative, and thus do not enhance the main story being told, we elected not to include these analyses in the manuscript.

2) Assessing correlations between meaningful environmental variables (e.g., temperature, light intensity, nutrient availability) and the occurrence of leptophyceae.

I recommend making this environmental and ecological metagenomic analysis a main figure in the manuscript. This would offer a broader “biological” perspective on leptophyceae and could stimulate further interest and targeted searches for this lineage in the ocean. For your reference, please see Fig. 4 of Alexander et al. (2023), Eukaryotic genomes from a global metagenomic data set illuminate trophic modes and biogeography of ocean plankton. MBio 14(6): e01676-23.

Response: We thank the reviewer for this helpful suggestion. We have incorporated this analysis into our revised manuscript. The updated methods section now describes this approach (lines 737-747, relevant section below):

“Leptophyte associations with environmental parameters

We assessed correlations between leptophyte abundance and eight physiochemical parameters: sea surface temperature, salinity, dissolved silica, nitrate, phosphate, iron, and seasonality indices of nitrate and sea surface temperature. These parameters were obtained from Delmont et al 2022 ¹⁶, representing data pulled from climatology and biogeochemical modelling data (World Ocean Atlas 2013 and PISCES v2)^{73,74} to account for missing physio-chemical samples in the Tara Oceans in-situ dataset. Seasonality indices were defined as the range of the nitrate and temperature in one grid cell divided by the total range of that variable across all Tara sampling stations.

We calculated Pearson’s correlation coefficients between the abundance of each leptophyte ptMAG and the environmental parameters using the were calculated using the R package, corrgram v1.14.”

The correlations between leptophyte abundance and environmental parameters are depicted in the figure below, with blue boxes indicating positive correlations, red boxes indicating negative correlations, and the shading intensity indicating the strength of the correlations.

Briefly, we found a mild negative association of Lepto-03 and Lepto-04 abundance with nitrate and phosphate concentrations, and a mild positive association with sea surface temperature and salinity. In contrast, the polar Lepto-01 and Lepto-02 ptMAGs were negatively correlated with temperature and salinity, and had a slight positive correlation with nutrient concentrations.

As these associations largely recapitulate the biogeography of the leptophyte ptMAGs (shown in Supplementary Figure 12), we have included this analysis as a Supplementary Figure (Supp. Fig 13). The corresponding text in the manuscript reads as (line 170-174):

*“At the level of individual genomes, Lepto-01 and Lepto-02 were only detected in the Arctic Ocean **characterized by cooler, less saline, and more nutrient-rich waters**, while Lepto-03 and Lepto-04 showed a broader and non-polar distribution **associated with warmer, more saline, and nutrient-poor waters** (Supplementary Figs. 12-13).”*

I especially appreciate the newly added mitochondria section. Considering the coefficient of the regression model for the mt & pt MAG coverage (~6/40), the relative copy number between pt and mt DNA seems to be about 100:15 for leptophytes (though this could vary among species). This led me to think that, given the summed coverage of Lepto 3 and 4 in Supplementary Table 1 (as well as Supplementary Figure 12, where I can see low mean coverage but many dots in panels C and D), which are 116 and 105, respectively, it might be possible to recover some mt contigs from the co-assemblies of all the sites where Lepto 3 and 4 are present—similar to what the authors did for Lepto-01's mt MAG. The presence of the Lepto-01 mt MAG would additionally increase sensitivity in finding contigs, if any exist.

Response: This was a very good suggestion. We agree that finding mtMAGs corresponding to Lepto-03 and Lepto-04 would further strengthen the findings, and we carefully considered the reviewer's request. Ultimately, however, we determined that recovering mtMAGs from Lepto-3 and Lepto-4 is highly unlikely with the existing co-assemblies, due to their consistently low coverage compared to the bloom-type pattern of Lepto-1.

While obtaining the mtMAG of Lepto-1 was feasible from just a few samples, we determined that Lepto-3 and Lepto-4 would require the reconstruction of very large co-assemblies across many samples to accumulate sufficient signal, a task that would take weeks to complete despite our access to very efficient servers. Based on current data, their cumulative coverage suggests only ~8–10x coverage for mitochondria, which most likely is not sufficient and at huge computational cost. Even if these assemblies were attempted, the key limitation remains: coverage per sample would be too low to establish reliable plastid–mitochondria correlations. Indeed, the mtMAGs for Lepto-3 and Lepto-4, if reconstructed, would not be detected in any sample of *Tara* Oceans, if their ratio to the plastid genome is similar to what we observed for Lepto-1. Thus, while additional mtMAGs might be reconstructed, we would then be unable to link them robustly to the plastid leptophyte groups by means of co-occurrence patterns (a most critical metric), limiting the value of such analyses. We are optimistic that with more and more metagenomic data being produced and made publicly available, more mtMAGs for the leptophytes will be characterised in the years to come. Given the blooming ability of leptophytes in the Arctic, we speculate that most mtMAGs will be characterized from high latitudes.

Line 176 " providing evidence that leptophytes can be abundant under certain conditions": This appears to contradict the authors' results, as most *psbO* coverages were ~3 (median), and no putative leptophyte *psbO* was found. As the authors stated that mitochondrial genomes are typically multi-copy (which also holds for plastid genomes), using plastid genome coverage as a proxy for species abundance could be misleading. Any explanation on this?

Response: This is a very good point. On the one hand, the plastid of Lepto_01 is among the most abundant ones in some Arctic *Tara* Oceans filters, indicating that it is quite abundant among photosynthetic eukaryotes at times in high latitudes. On the other hand, we failed to identify any marker gene (including *psbO*) for its nuclear genome in the metagenomic co-assembly covering those filters.

Two hypotheses come to mind. First, leptophytes might have an unusually high copy number of plastid genomes per nuclear genome compared to most of the co-occurring photosynthetic eukaryotes. Or possibly plastid genome copy number increases in leptophytes under favourable conditions, as is known in other algae (<http://doi.org/10.1104/pp.113.216291>, <http://doi.org/10.1105/tpc.106.045427>). In this scenario, we overestimate the relative abundance of leptophytes. Second, the nuclear genome of leptophytes might be quite large, explaining why it is not reconstructed in the co-assembly despite a relatively high abundance of the focal population in the filters. Indeed, very large genomes (e.g., dinoflagellates) simply are not part of the *Tara* Oceans co-assemblies due to insufficient mean coverage.

In summary, the high signal for the plastid genome of Lepto-01 is indicative of a relatively high abundance of this population, but we cannot determine with high accuracy its relative abundance until the nuclear genome is characterized. To address the reviewer's comment, we have modified the text as followed:

“..providing evidence that leptophytes can experience localized increases in abundance under certain conditions.”

Minor points:

- Fig. 1: The taxonomic labels are hidden (Leptophytes etc).

Response: We are not sure why the figures cannot be viewed properly, but we will ensure that there are no discrepancies in the final published version. Thank you for pointing this out. We show Figure 1 below (including taxonomic labels).

- Both "co-assembly" and "coassembly" are used. Was this intentional?

Response: Thank you for pointing this out. We now use "co-assembly" throughout the manuscript.

Reviewer #2 (Remarks to the Author):

This is a substantially revised version of a manuscript I reviewed before. While I previously recognized the significance of the results presented and in principle endorsed publication in Nature Communications, at the same time I raised a number of points to be addressed before the

manuscript is mature enough to go to print. A short summary of my assessment of the new version is that the authors have materially improved the manuscript both formally and with regard to the content. I am especially pleased to see that my suggestion to look for the mitochondrial genome of leptophytes was followed by the authors and that their effort was successful, bringing an important piece of data that help illuminate the phylogenetic position of leptophytes in the eukaryote tree of life. I also appreciate the effort the authors have invested into addressing my question regarding the very large ptMAGs. I agree that the full presentation of the results is better suited for a separate paper. Overall, while the new manuscript version is certainly much more informative and accurate in most details, the newly added material at the same time brought some new problems that are important enough not to be ignored. I comment on them in some detail below. Furthermore, there are also some persisting minor formal issues that also should be addressed before the manuscript may be formally accepted for publication. Hence, one extra round of revision (“minor revision” in the common parlance) is a must in my opinion.

Problems with the new analysis of mitochondrial genome

While I am convinced that the authors have identified a bona fide leptophyte mitogenome sequence, and that the main result of the phylogenetic analysis presented in the new Fig. 4, i.e. that the leptophyte mitochondrion (and most likely the “host cell” as a whole) represents a lineage sister to haptophytes, is robust. Nevertheless, the phylogenetic analysis and its presentation suffer from several problems apparent upon a closer scrutiny. Here you are:

Response: We thank the reviewer once again for their thorough review. We have re-run the mitochondrial phylogenies after addressing all the issues pointed out here, and provide details below.

(1) Regarding the assembly of the dataset for the phylogenetic analysis, the authors provide the following details (lines 787-789): “we used the publicly available dataset of Williamson et al 2025 31 with 93 protein-coding genes and 100 taxa as a starting point. We subset this dataset to retain 54 eukaryotic taxa only”. However, I am somewhat confused by the numbers indicated in the sentence. Having checked the table “Taxa used for mitochondrial phylogenetic analyses”, which I believe corresponds to Supplementary Table S11 (see below for my complains regarding the problem to match the supplementary table files), I think the authors have retained 45, not 54 taxa from the original dataset by Williamson et al. Please double check and fix either the text or the table, something is wrong here. Next, the list of taxa provided by Williamson et al. in their Supplementary Table S7 includes only 87 items, so it is not clear to me how the authors could have started with 100 taxa.

Response: We apologize for the confusion. We did retain 54 eukaryotic taxa from the Williamson et al. dataset, and the corresponding supplementary table was missing 9 of these due to a copy pasting error. Supplementary Table 11 has now been updated accordingly. Thank you for raising this issue.

Next, we indeed started out with 100 taxa (unaligned sequences downloaded from the figshare repository of the Williamson et al dataset). This starting dataset consisted of 87 eukaryotic taxa as noted by the reviewer, as well as 13 bacterial taxa that were used as outgroups in the Williamson et al study.

(2) The authors further state this (line 792-793): “mtMAGs and additional reference genomes were annotated MFannot using the standard genetic code”. However, by sticking to the standard genetic code the authors ignored the fact that some of the newly added “reference genomes” have been previously demonstrated to utilize a non-standard genetic code. Just glancing over the list of mitogenome concerned, this concerns at least the following taxa, all of which employ UGA as a tryptophan codon: *Marophrys* sp. (see <https://www.nature.com/articles/s41598-019-41238-6>), *Cryothecomonas* sp. (MK188936; see <https://www.nature.com/articles/s41564-019-0605-4>; UGA is used to decode Trp in all rhizarians investigated), and *Phaeocystis antarctica* (see the respective [NCBI record: https://www.ncbi.nlm.nih.gov/Taxonomy/Browser/wwwtax.cgi?id=33657](https://www.ncbi.nlm.nih.gov/Taxonomy/Browser/wwwtax.cgi?id=33657); UGA=Trp holds for all members of *Prymnesiophyceae* studied). Indeed, checking the sets of predicted protein sequences employed in the phylogenetic analysis (as provided by the authors on the Figshare repository) shows sequences from these species to have in-frame asterisks (standing for UGA codons). The predictions provided by MFannot in the cases of using a genetic code that ignores stop- to-sense reassignments are not only peppered with asterisks, abut are frequently incomplete. The number of genomes added to the original dataset available thanks to the previous work by Williamson et al. is not that high as to resign of certain degree of accuracy, especially in the cases where correct annotation of the mitogenome is already available (as is the case, e.g. for *Marophrys* sp.). Inspecting the sequence set provided by the authors immediately indicates that code variants with stop-to-sense reassignments are employed also by some of the mtMAGs reconstructed by the authors, and these should also be annotated properly. (Thanks God this does not concern the leptophyte mitogenome, which happens to use the standard genetic code, as quickly assessed by FACIL; <https://pubmed.ncbi.nlm.nih.gov/21653513/>). The mitochondrial phylogenetic analysis is central for the paper – it is presented as one of just four main figures delivering original data. Given also the reputation of the journal *Jamy et al.* want to publish their work in, and the fact that only a properly assembled dataset will make a useful resource for future use, I think it is a must the mitogenomes are reannotated and the analysis is repeated with properly translated sequences.

Response: The reviewer is correct that many mitochondrial genomes were unfortunately annotated using the incorrect genetic code. We have now remedied this by (1) checking the genetic code of all mtMAGs and additional reference genomes using Codetta (Shulgina and Eddy 2023), and (2) rerunning MFannot with the appropriate genetic code. We excluded a reference genome (*Labyrinthulomycetes* sp. S4 with accession number MK188940) at this stage as its genetic code was ambiguous and this lineage is not critical for our work. The phylogenetic dataset was assembled with the reannotated mitochondrial genomes and phylogenies inferred again.

The methods section has been updated to reflect these changes (lines 859-860):

“This was done as follows: (1) the tool Codetta⁸¹ v2.0 was used to determine the genetic code of all mtMAGs and additional reference genomes, (2) mtMAGs and additional reference genomes were annotated MFannot using the standard or mold mitochondrial genetic code as appropriate...”

(4) The authors write that they used for the mitogenome phylogenetic analysis a dataset previously published by Williamson et al. (their ref. #31). However, having checked the documentation of the dataset by Williamson et al. (their Supplementary Table S7), I have noticed some discrepancies that prompted me to dig a bit deeper into the dataset presented by Jamy et al. The conclusion is that there is some chaos regarding the taxonomic source of a subset of sequences employed in the phylogenetic analysis of mitochondrial genomes, and at least some of the sequences come from species different from, although related to, those indicated by name in Fig. 4 or the supplementary table “Taxa used for mitochondrial phylogenetic analyses”. I see it is common in phylogenomic analyses to make composite taxa by combining sequences from different closely related taxa, but this must be explicitly stated, the sources must be documented, and there should be a scientific or methodical reason for such an approach. None of the listed criteria are fulfilled by Jamy et al. Hence, for the sake of consistency and accuracy, also to ensure the assembled dataset is a valuable resource for the future, please carefully check the origin of all sequences included and ensure they come from the taxa specified in the trees and tables. I point to three specific cases I could quickly notice, but there might be others that I have missed.

Micromonas pusilla – Williamson et al. state in their Supplementary Table S7 that they included in their dataset the mitochondrial genome from the species *Micromonas commoda*, not *M. pusilla* (it seems they have combined nuclear genes from the latter with mitochondrial genes from the former). As Jamy et al. explain in Materials and methods, they used only the mitochondrial genes from that dataset, so I assumed that the protein sequences employed in their analysis must all come from *M. commoda*. However, having checked just a few of them it turned out that they represent a mixture, some (such as the Atp4 sequence) coming from *M. commoda* (being identical to YP_002860120.1), while others (such as Atp1) coming from *M. pusilla*. However, regardless their origin from different species (in fact very divergent from each other: Atp4 proteins from the two species share only 31% identity!), they are all labelled “Archaeplastida_Viridiplantae_Micromonas_pusilla” by Jamy et al., and despite the respective branch in the phylogenetic tree in Fig. being labelled “*Micromonas pusilla*”. However, this problem just pointed an even more general serious issue of the dataset employed by Jamy et al. While the authors claim that they used in the analysis protein sequences encoded by mitochondrial genomes [see lines 788-789: “We subset this dataset to retain 54 eukaryotic taxa only, and mitochondrial encoded genes (40 protein-coding genes)”], this in reality does not hold for all the sequences in the analysis. For example, the Atp1 protein from *Micromonas* is encoded by the NUCLEAR genome, which in fact explains why the sequence in the dataset comes from *M. pusilla* and not *M. commoda* (Williamson indeed used the nucleus-encoded proteins from *M. pusilla*). Crucially, this is unlikely to be an isolated issue, as many of the genes included by Jamy et al. in their analysis vary among taxa when it comes to their location: in some they have been retained in the mitogenome, while in others they are nuclear. For example, nad7 to nad11 genes are mitochondrial in some eukaryotes, but nuclear in others (see

<https://bmcbiol.biomedcentral.com/articles/10.1186/s12915-024-01824-1>). Furthermore, there is even no guarantee the nuclear versions must be orthologous to the mitochondrial ones (acquisition of xenologs from various external sources, rather than endosymbiotic gene transfer is common here). Hence, I think Jamy et al. should carefully check their dataset and ensure that only mitogenome-encoded proteins are included, of course properly assigned to the actual source organisms.

Response: The reviewer raised several relevant issues here which we address one by one.

1. Composite taxa (*Micromonas pusilla* and *Micromonas commoda*)

For our previous set of phylogenetic analyses, we had simply used the Williamson et al dataset as made available by the authors. Based on the reviewer's suggestion, we have now:

- Manually checked where the sequences of each *Micromonas* gene in our dataset originated from.
- Retained only those that were present in the *Micromonas commoda* mitochondrial genome (accession number NC_012643). We deleted the genes *atp1*, *nad11*, and *rpl2* as they were only present in the *M. pusilla* transcriptome.
- Renamed the taxon as *Micromonas commoda*.
- Re-run the phylogenies with the updated dataset.

2. Mix of nuclear and mitochondrial genes

Based on this comment, we have now dealt more explicitly with the fact that while some genes are always (or almost always) present in the mitochondrial genome (such as *cox1* and *cox3*), others may be present in the nuclear genome or the mitochondrial genome depending on the taxa.

We can break down which of our taxa might also include nuclear genomic sequences based on the source of the data:

- **derived from Williamson et al** – these taxa comprise a mix of nuclear and mitochondrial data, and thus might contain nuclear genes (including several examples pointed out by the reviewer)
- **mtMAGs** – only contain mitochondrial genes by definition
- **additional mitochondrial genome references** – also only contain mitochondrial genes by definition

We therefore carefully went through the 54 taxa derived from the Williamson et al dataset and determined which genes were of mitochondrial origin. We provide a summary below:

- A mitogenome was publicly available for 43 of these taxa (accession number now provided in Supplementary Table 11). For these 43 taxa, we retained only those genes that we determined to be present on the mitogenomes based on manual checks.
- For the taxon “CRuMs_Collodictyonidae_Collodictyon_sp”, we searched for the mitogenome in the genome scaffolds (BioProject: PRJNA1153407), annotated it with MFannot, and then retained only those genes found on the mitogenome.
- Of the remaining 10 taxa with no publicly available mitogenome, we retained only genes that are (almost) always present on the mitochondrial genome such as *cox1*, *cox3*, *nad4*, and *nad5* (<https://bmcbiol.biomedcentral.com/articles/10.1186/s12915-024-01824-1>). This approach meant that we were unable to retain the following taxa as they were only represented by nuclear genes:
 - Alveolata_Chrompodellids_Vitrella_brassicaformis (which contains only 3 genes on its mitogenome; <https://doi.org/10.1093/molbev/msv021>)
 - Haptista_Haptophyta_Prymnesium_parvum
 - CRuMs_Rigifilida_Rigifila_ramosa
 - Rhizaria_Foraminifera_Reticulomyxa_filosa

As a result of these filtering steps, the taxon “Haptista_Haptophyta_Pavlova_lutheri” was represented by just one gene sequence (*cox2*). Given that the taxon was the sole representative of Pavloales, an important phylogenetic position in the context of resolving the phylogenetic position of the leptophyte mitochondria, we added an additional reference genome for Pavloales (*Diacronema viridis*; accession = MW044629).

Altogether, this resulted in a phylogenetic dataset comprising 102 taxa, and 28 genes (please see the updated Supplementary Table 11). While the dataset was rather patchy, we opted not to filter taxa based on percentage of missing data in order to have a taxon sampling as broad as possible. The new Figure 4 (see below) is based on ML and Bayesian inferences of this dataset, and as before, recovers a strong sister relationship of leptophyte and haptophyte mitochondria.

New Figure 4 in manuscript. Mitochondrial phylogeny based on 28 genes.

Finally, we also generated a subset of the data composed of genes that are present on the mitochondria of nearly all eukaryotic taxa (*cob*, *cox1*, *cox3*, *nad1*, *nad2*, *nad3*, *nad4*, *nad4L*, *nad5*, *nad6*, *tatC*). This dataset comprised 101 taxa, 11 genes, and 3,270 amino sites. Phylogenies inferred with this subset also recovered the same sister relationship of leptophyte and haptophyte plastids (see figure below), but with lower support values (as expected based on the smaller number of genes). We show results based only on the fuller 28-gene dataset in the manuscript.

Mitochondrial phylogeny based on 11 genes. The sister relationship of Lepto-01 and haptophytes is recovered but with slightly lower support (PP=0.96, UFB=93).

The methods section has now been updated to reflect these changes:

“To assemble a mitochondrial phylogenetic dataset, we used the publicly available dataset of Williamson et al 2025³¹ with 93 protein-coding genes and 100 taxa as a starting point.

We subset this dataset to retain **50 eukaryotic taxa and the genes present in their corresponding mitochondrial genomes** (40 protein-coding genes). *To this dataset, we added our 34 mtMAGs and 18 additional reference genomes from various sources to increase the taxon sampling of cryptophytes and haptophytes as well as under-represented lineages such as Picozoa (Supplementary Table 11).*”

And on line 874:

“We concatenated the genes to obtain an alignment with **102 taxa, 28 genes, and 6,302 sites.**”

Mantamonas plastica – here the situation seems to be analogous to the one with *M. pusilla*/*M. commoda*. The dataset by Williamson et al. uses the mitochondrial genome of the different species *Mantamonas sphyrenae*, which is even indicated by Jamy et al. in their own supplementary table entitled “Taxa used for mitochondrial phylogenetic analyses” (583574_2_supp_11198168_t04rss.xlsx). It is then surprising that the species is named “*Mantamonas plastica*” in Fig. 4. However, as with the *M. pusilla*/*M. commoda* case, the solution here is not simply renaming the species: the sequences assigned to the same *Mantamonas* species are again a mixture, some coming from mitogenome of *Mantamonas sphyrenae* but some, surprisingly, indeed derived from *Mantamonas plastica*. My quick analysis indicates the latter concerns *Atp1*, for which a sequence derived from the transcriptome assembly of *M. plastica* is included. This transcript presumably comes from the mitochondrion of the species, as the *atp1* gene is mitochondrial in *M. sphyrenae* (note that the mitogenome sequence from *M. plastica* is most likely unavailable, or at least I am not aware of it). It thus seems that in this case the problem traces back already to the original dataset by Williamson et al., who perhaps used *M. plastica* mitochondrial sequences they could identify in the transcriptome assembly and complemented the dataset by including additional mitochondrial sequences from *M. sphyrenae*. It is obvious Jamy et al. should ensure they use sequences from the single species and single genome, that is *M. sphyrenae* mitogenome.

Response: The reviewer is correct about the mixture of genes from *M. plastica* and *M. sphyrenae*. To address this issue, we manually verified the source of all sequences. We found that only the *atp1* gene originated from *M. plastica*, while all other sequences were derived from the *M. sphyrenae* mitogenome (accession = LC842150). We have corrected this by replacing the *atp1* sequence with the appropriate *M. sphyrenae* sequence. The taxon name has also been updated in all phylogenies and Figure 4.

Telonema subtile – here again the dataset used by Jamy et al. consists of sequences from two different organisms, i.e. two different telonemids, one really being *Telonema subtile*, as holds e.g. to the *Atp1* sequence derived from the transcriptome assembly from this species (as presented in EukProt v3). The other source (check, e.g., *Atp4*) is a very different telonemid by the

mitogenome sequence MN082145, as is also obvious from Supplementary Table S7 by Williamson et al. (not also that those authors indicate the source of data as “Telonemida sp.”, not *Telonema subtile*). The mitogenome sequence comes from the study by Wideman et al. (<https://pubmed.ncbi.nlm.nih.gov/31768028/>). Now the question is what is the identity of the organism behind this sequence. This is not directly apparent, but a recent 18S rRNA-based phylogenetic analysis of *Telonemia* by Zlatogursky et al. (<https://www.sciencedirect.com/science/article/pii/S2589004225014452>; see their Fig. S2) indicates that the mitogenome MN082145 corresponds to a telonemid labelled “T12” and being distantly related to *T. subtile*. Notably, according to the 18S rRNA tree the T12 organism is more closely related to the telonemid T11 than to *T. subtile* or the telonemid T1, and this topology is consistent with the tree inferred by Jamy et al. from the mitochondrial sequences. Hence, I think the set of sequences grouped to represent the single OUT named by Jamy et al. “*Telonema subtile*” is most likely dominated by sequences from the “T12 telonemid”, which have drawn it to that particular position in the tree. I leave on the authors to confirm my insight. At any rate, it is obvious the dataset they used for the analysis is a poorly documented chaotic mixture of sequences, not a proper phylogenomic dataset.

Response: We appreciate the reviewer’s observation about the mixture of sequences from two different telonemid taxa. As described earlier, we manually verified the source of each sequence, retaining only the genes from the mitogenome MN082145 belonging to the telonemid T12. We removed the *atp1* and *nad11* sequences which stemmed from the *Telonema* transcriptome assembly on EukProt.

Related to the issues regarding the sources of the sequences used, while checking the list of taxa provide by Jamy et al. in their table “Taxa used for mitochondrial phylogenetic analyses”, I could not find in it the species *Cyanophora paradoxa*, although it is included in the tree presented as Fig. 4. It is possible that this is not the only such inconsistency, so please, check the table and ensure all taxa include in the analysis are listed.

Response: Thank you for spotting the error. As described in response to a previous comment, Supplementary Table 11 was missing nine taxa due to a copy pasting error. This mistake has now been fixed.

(5) As was the case of the plastid trees presented in the original version of the manuscript, the newly added mitochondrial tree (Fig. 4 and associated/underlying files) also uses outdated nomenclature for certain taxa included in it; please use the updated names when rerunning the tree:

Physcomitrella patens – note that the correct name (revising the historical classification based on molecular phylogenetics) is *Physcomitrium patens*, a name now widely adopted by the community (<https://academic.oup.com/plcell/article/32/5/1361/6115584>).

Goniomonas avonlea – note that there is a preprint (now in the revised third version) available at bioRxiv reporting on a thorough taxonomic revision of the traditional extremely broad genus Goniomonas, splitting it into multiple genera; see <https://www.biorxiv.org/content/10.1101/2024.07.17.603845v3>. The species previously known as G. avonlea has been assigned into a new genus as Neptunogoniomonas avonlea. I believe the preprint will soon be published and the updated name should be used by Jamy et al.

Spironema multiciliatum – note that the data in fact come from an organism originally referred to as “Spironema cf. multiciliatum”, which means it resembled the species Spironema multiciliatum but the authors thought the identification is not certain (<https://pubmed.ncbi.nlm.nih.gov/30429611/>). I believe this designation should be kept.

Response: Thank you for catching these details. The taxa names have now been updated in the figures and provided data on the Figshare repository.

Other (generally minor) issues:

- I think it would be fair to explicitly mention the preprint by Shrestha et al. 2025 in the text, as it reports on an independent discovery of leptophytes (not named such) based on metagenomic data.

Response: This is a fair point. The Shrestha et al 2025 preprint was released 3 months after ours, and notably, they identified a ptMAG 99.9% similar to Lepto-01 from metagenomes generated by a different sampling expedition. We have now added this sentence to our discussion:

*“One very interesting new group of plastid genomes are the leptophytes, a globally distributed and generally rare deep-branching lineage formed by four ptMAGs in our data. This is remarkable because novel plastid diversity at this taxonomic depth is very rarely reported. Currently, only the environmental plastid lineages DPL1 and especially DPL2 are still considered possible deep-branching phytoplankton lineages outside of eukaryotic supergroups¹³. Based on 16S rDNA data recovered from the ptMAGs and biogeographical comparison, we show not only that one of the leptophytes (Lepto-04) corresponds to DPL2, thus representing the first genomic data available for this enigmatic group, but also that leptophytes form a more diverse, widespread, and abundant group than previously known. **Part of our findings have now been independently recapitulated, with the notable recovery of the Lepto-01ptMAG from a different Arctic metagenomic data set (REF).**”*

- please insert into each supplementary table a legend including the title of the table and its number (Table SX). Note that as a reviewer I see the tables only as files with strange names like “583574_2_supp_11198166_t04rss.xlsx” , and I am only guessing which file corresponds to

which table mentioned in the text. I believe adding these details directly into the tables is also a favour to any prospective reader.

Response: Thank you for pointing this out! We labelled our file names as “Supplementary_Table_1.xlsx” etc, but the file names presumably got changed during the submission process. We have now added the table numbers and captions directly into the tables.

- line 174: “coverage of 2.3x”, line 175: “124x mean coverage” – replace the letter “x” with the symbol for “times (i.e., “x”) and check other possible analogous instances

Response: Done!

- line 197-198: “leptophyte plastids genomes” – correct to “leptophyte plastid genomes”

Response: Done!

- line 277: “places leptophytes as sister to haptophytes” – is this linguistically OK? Please double check.

Response: We believe the wording “places leptophytes as sister to haptophytes” is appropriate as is, as it follows standard terminology used in phylogenetic literature to describe sister-group relationships (e.g., “placed X as sister to Y”). Therefore, we have retained the original wording.

- line 314: “encodes 37 protein coding genes as well as 22 tRNA genes” – I am not sure if it is OK to speak about genes to be “encoded” by a genome, I would say that genes are “harboured”, “contained” of something like this by a genome.

Response: Thank you for your comment. We have replaced “encodes” with “contains” here.

- line 397: delete the space after “nucleomorph”

Response: Done!

- lines 401-402: “this model is the first to propose haptophytes as the host of the secondary endosymbiosis” – this statement is in fact inconsistent with the narrative that precedes it, as the author explicitly assume the origin of the secondary plastid BEFORE the split of haptophytes and leptophytes. I do see the possibility that leptophytes may best be eventually classified as a haptophyte lineage (this will always be an arbitrary decision), but I think it is at any rate more accurate to write something like this: “this model is the first to propose the haptophyte ancestor as the host of the secondary endosymbiosis”.

Response: This is a very good point. We agree completely with the reviewer (and the proposed scenario in Figure 5 is consistent with that). We have now updated the sentence as follows:

“To our knowledge, this model is the first to propose the haptophyte **ancestor** as the host of the secondary endosymbiosis with red alga...”

- line 412: the placement of the reference to Supplementary Fig. 23 at the end of the sentence is misleading, as it does not show the phylogenies of SELMA proteins and its content is relevant only to the first half of the sentence, so move it to the place just after “gene content comparison (5 vs 33)”.

Response: Done!

- line 469: “plastid genomes from culture” – sounds a bit weird to me, I would write “plastid genomes from cultured organisms”

Response: Changed as suggested.

- lines 500-506: would it make better sense to integrate the newly added section (“Check for mitochondrial contamination”) into the section “Creation of a non-redundant plastid genomic database”? I would think that the removal of mitochondrial contaminants should precede estimation of ptMAG completeness (as some of the genes are shared by plastid and mitochondrial genes).

Response: Yes, that does make more sense, and indeed statistics for plastid MAGs were re-estimated after removing mitochondrial contamination. We have now integrated that section into “Creation of a non-redundant plastid genomic database”

- line 502: remove the space before the period ending the sentence. Note that this is just one example a general problem, when in the newly added text the inserted citations are followed by an extra space that breaks the text and should be removed.

Response: We have carefully checked the manuscript and confirmed that there are no extra spaces before periods or after citations in the text. It is possible that the spacing issue appeared due to formatting differences during the conversion process. But we will ensure that there are no discrepancies in the final published version.

- lines 707-722: note that italics for the gene name “psbO” are used inconsistently in this paragraph, so please, check and unify.

Response: Thank you! This is now fixed.

- line 753: “cryptista” – I think the adjective form “cryptist” is needed here

Response: Updated.

- line 757-758: “This step revealed 13 functions that were present in these mitochondrial genomes” – what are “functions” here and how they can be “present” in genomes? What you list in the sentence are proteins or protein domains encoded by the mitochondrial genomes, so please, rephrase accordingly.

Response: The reviewer is correct. We have now updated the text as follows:

“This step revealed 13 core genes present in these mitochondrial genomes that were functionally annotated as:...”

- lines 758-759: you should exert more strive towards consistency, it’s strange to see “Cytochrome c oxidase subunit III” and “Cytochrome C oxidase subunit II periplasmic domain” next to each other. I believe that “cytochrome c”, with the letter “c” not capitalized and instead italicized, is the convention adopted by biochemists. The same on line 767.

Response: Apologies for the inconsistencies. We have now updated the text as follows:

“This step revealed 13 core genes present in these mitochondrial genomes that were functionally annotated as: Cytochrome c oxidase subunit III, Cytochrome c and quinol oxidase polypeptide I, Cytochrome c oxidase subunit II (periplasmic domain), Cytochrome b/b₆, Proton-conducting membrane transporter, NADH-ubiquinone/plastoquinone oxidoreductase chain 4L, NADH-ubiquinone/plastoquinone oxidoreductase chain 6, NADH-ubiquinone/plastoquinone oxidoreductase chain 3, ATP synthase subunit α , ATP synthase subunit c, Ribosomal protein L16p/L10e, Ribosomal protein S12/S23, ATP synthase F₀ subunit.”

- line 797: “sequences for each” – for the sake of consistency, capitalize “s” at the beginning of “sequences”

Response: Done.

- line 924: replace the page numbers “1 – 21” by the article number “210”, the paper has been published in a journal that does not have continuous pagination (see <https://bmcecolevol.biomedcentral.com/articles/10.1186/1471-2148-10-210>). The same correction needs to be implemented in the same reference in the Supplementary Information file.

Response: Good catch! This has now been fixed!

- line 926: add the missing year of publication of the paper (2023)

Response: Fixed!

- I did not have enough mental power to carefully check the legends to supplementary figures, but just glancing over the file I have noticed minor issues, such as inconsistent use of italics for gene names (e.g., “*psaD*” in the legend to Supplementary Fig. 15)

Response: Thank you. We have gone through the supplementary material and fixed any errors that we spotted. The biggest change is updating the number of ptMAGs presented in Supplementary Figure 1.

Reviewer #3 (Remarks to the Author):

I appreciate the extremely thorough revision (and response to reviewer comments) undertaken by the authors. I am satisfied with the changes made, and am excited to see the discovery of a putative 'leptophyte' mitochondrial MAG -- a very important addition to an already important manuscript.

Response: We thank Prof. Archibald for his positive assessment!

Dear Editor,

Please find below our responses to all comments in this third round of revisions. We have reproduced the decision letter verbatim with our detailed responses to all comments interleaved in blue for clarity. We have addressed all comments the reviewers, and modified our manuscript accordingly, which we hope will now satisfy the requirements for publication in *Nature Communications*.

Reviewer #1 (Remarks to the Author):

The authors have addressed adequately the comments that I raised in the revised version of manuscript. Therefore, I have no further comments. It seems it is ready for publication.

- Response: We thank the reviewer for their positive remarks!

Reviewer #2 (Remarks to the Author):

I have provided a very detail account on the previous two versions of the manuscript, so this time I took the liberty of immersing myself more shallowly into the updated version, also because of being extremely occupied by other duties for an extended period of time from now on. Hence, I have restricted myself to checking the responses by the authors to the critical points I raised in my previous review and glanced over the version of the manuscript with tracked changes, and it seems that all the issues have been dealt with satisfactorily. Crucially, it seems Jamy at al. have completely rebuilt the mitochondrial phylogenomic dataset, which was my major concern in the previous review. Importantly, as I expected the new phylogenetic tree does not change the key conclusion of the previous analysis that the putative leptophyte mitogenome constitutes as sister lineage of haptophyte mitogenomes. However, having focused on the new version of Fig. 4 I do see a few minor issues that need to be fixed:

- Response: We thank the reviewer for their efforts to improve our work. We have made all the changes requested and provide details below. We also show the new Figure 4 at the end of the comments with all the requested changes.

1. Note that compared to the previous version you have changed the name of the haptophyte correctly called “Diacronema lutheri” to the outdated name “Pavlova lutheri”. I think this is most likely not intentional, as you have kept the name “Diacronema lutheri” in the figures displaying results of plastid phylogenomic analyses. Please fix the name for the sake of accuracy and internal consistency.

- Response: The species name has now been fixed!

2. I would object the delimitation of “Cryptophytes” in the mitochondrial tree, which is expanded to embrace Neptuniomonas avonlea. The point is that “cryptophytes” are generally presented in

the paper as a group stemming from an ancestor that has acquired a higher-order plastid. Indeed, the very etymology of cryptophytes/Cryptophyta/Cryptophyceae is indicative is an algal group comprised of members that are predominantly, or at least ancestrally, photosynthetic. Hence, despite being aware of the fact that the concept of “cryptophytes” varies in the literature, I strongly endorse its use as an equivalent of the cryptist clade “seeded” by an acquisition of a plastid, i.e. as the algal group in Cryptista (analogously to Euglenophyceae being the algal subgroup of Euglenida). As *Neptuniomonas avonlea* and most likely presumably goniomonads in general do not have a plastid, and there is also considerable genomic evidence that this is not a secondary state, I find including goniomonads among cryptophytes illogical. Imagine a naïve reader of the paper comparing Fig. 4 and Fig. 5. A conclusion such a reader must reach is that the plastid acquisition in the stem cryptophyte lineage depicted in Fig. 5 must have occurred before the divergence of *Neptuniomonas avonlea* and the (other) cryptophytes, which is however nearly certainly not true. Hence, consider narrowing the delimitation of cryptophytes in Fig. 4 to those taxa that have a plastid to avoid conveying a potentially misleading message. A note, the clade now annotated as “Cryptophytes” by Jamy et al. may alternatively (more aptly) be called Cryptomonada.

- Response: This is a fair point. We have retained the term “cryptophytes” to be consistent throughout the manuscript, but no longer include *Neptuniomonas avonlea* in the delimitation in Figure 4.

3. Related to the latter point is an issue with the paraphyletic grade delimited in Fig 4 as “Other Cryptista”. If the previous point (narrowing cryptophytes to the plastid-bearing lineage) is accepted, then in principle “Other Cryptista” would have to be extended to embrace also *N. avonlea*. However, the real issue here is that as delimited at the moment the grouping comprises *Microheliella maris*, whose status as a “cryptist” is contentious. More specifically, the recent literature discriminates between a narrower clade Cryptista, which does not include *Microheliella*, and a more inclusive clade that has *Microheliella* sister to Cryptista and is called Pancryptista; see <https://royalsocietypublishing.org/doi/10.1098/rsob.210376>. I do admit that an alternative concept of Cryptista exists, advocated by the late Tom Cavalier-Smith, including in his posthumously published 2022 paper (<https://link.springer.com/article/10.1007/s00709-021-01665-7>), in which *Microheliella* is classified as a member of a newly established subphylum Endohelia in the phylum Cryptista. However, I personally find the distinction of Cryptista and Pancryptista well founded and functional, and I encourage the authors to embrace it in their paper.

- Response: Also a fair point. We tried several alternatives to fix the labelling that would convey the classification without cluttering the figure. In the final version that we settled on, we have changed the label “Other Cryptista” to “Other Pancryptista”. This wider taxonomic grouping includes all non-photosynthetic cryptomonads (such as goniomonads), kathablepharids, and also *Microheliella*. Please see the new Figure 4 below.

And this is it, please accept the paper for publication after these few remaining points have been addressed by Jamy et al. I am not interested in seeing the manuscript once again before I find this important study published in Nature Communications.

Referee's report on the manuscript "**A new deep-branching environmental lineage of algae**" submitted to *Nature Communications* by Jamy et al.

This is a substantially revised version of a manuscript I reviewed before. While I previously recognized the significance of the results presented and in principle endorsed publication in *Nature Communications*, at the same time I raised a number of points to be addressed before the manuscript is mature enough to go to print. A short summary of my assessment of the new version is that the authors have materially improved the manuscript both formally and with regard to the content. I am especially pleased to see that my suggestion to look for the mitochondrial genome of leptophytes was followed by the authors and that their effort was successful, bringing an important piece of data that help illuminate the phylogenetic position of leptophytes in the eukaryote tree of life. I also appreciate the effort the authors have invested into addressing my question regarding the very large ptMAGs. I agree that the full presentation of the results is better suited for a separate paper. Overall, while the new manuscript version is certainly much more informative and accurate in most details, the newly added material at the same time brought some new problems that are important enough not to be ignored. I comment on them in some detail below. Furthermore, there are also some persisting minor formal issues that also should be addressed before the manuscript may be formally accepted for publication. Hence, one extra round of revision ("minor revision" in the common parlance) is a must in my opinion.

Problems with the new analysis of mitochondrial genome

While I am convinced that the authors have identified a bona fide leptophyte mitogenome sequence, and that the main result of the phylogenetic analysis presented in the new Fig. 4, i.e. that the leptophyte mitochondrion (and most likely the "host cell" as a whole) represents a lineage sister to haptophytes, is robust. Nevertheless, the phylogenetic analysis and its presentation suffer from several problems apparent upon a closer scrutiny. Here you are:

(1) Regarding the assembly of the dataset for the phylogenetic analysis, the authors provide the following details (lines 787-789): "we used the publicly available dataset of Williamson et al 2025³¹ with 93 protein-coding genes and 100 taxa as a starting point. We subset this dataset to retain 54 eukaryotic taxa only". However, I am somewhat confused by the numbers indicated in the sentence. Having checked the table "Taxa used for mitochondrial phylogenetic analyses", which I believe corresponds to Supplementary Table S11 (see below for my complains regarding the problem to match the supplementary table files), I think the authors have retained 45, not 54 taxa from the original dataset by Williamson et al. Please double check and fix either the text or the table, something is wrong here. Next, the list of taxa provided by Williamson et al. in their Supplementary Table S7 includes only 87 items, so it is not clear to me how the authors could have started with 100 taxa.

(2) The authors further state this (line 792-793): "mtMAGs and additional reference genomes were annotated MFannot using the standard genetic code". However, by sticking to the standard genetic code the authors ignored the fact that some of the newly added "reference genomes" have been previously demonstrated to utilize a non-standard genetic code. Just glancing over the list of mitogenome concerned, this concerns at least the following taxa, all of which employ UGA as a tryptophan codon: *Marophrys* sp. (see <https://www.nature.com/articles/s41598-019-41238-6>), *Cryothecomonas* sp. (MK188936; see <https://www.nature.com/articles/s41564-019-0605-4>; UGA is used to decode Trp in all rhizarians investigated), and *Phaeocystis antarctica* (see the respective NCBI record: <https://www.ncbi.nlm.nih.gov/Taxonomy/Browser/wwwtax.cgi?id=33657>; UGA=Trp holds

for all members of Prymnesiophyceae studied). Indeed, checking the sets of predicted protein sequences employed in the phylogenetic analysis (as provided by the authors on the Figshare repository) shows sequences from these species to have in-frame asterisks (standing for UGA codons). The predictions provided by MFannot in the cases of using a genetic code that ignores stop-to-sense reassignments are not only peppered with asterisks, but are frequently incomplete. The number of genomes added to the original dataset available thanks to the previous work by Williamson et al. is not that high as to resign of certain degree of accuracy, especially in the cases where correct annotation of the mitogenome is already available (as is the case, e.g. for *Marophrys* sp.). Inspecting the sequence set provided by the authors immediately indicates that code variants with stop-to-sense reassignments are employed also by some of the mtMAGs reconstructed by the authors, and these should also be annotated properly. (Thanks God this does not concern the leptophyte mitogenome, which happens to use the standard genetic code, as quickly assessed by FACIL; <https://pubmed.ncbi.nlm.nih.gov/21653513/>). The mitochondrial phylogenetic analysis is central for the paper – it is presented as one of just four main figures delivering original data. Given also the reputation of the journal Jamy et al. want to publish their work in, and the fact that only a properly assembled dataset will make a useful resource for future use, I think it is a must the mitogenomes are reannotated and the analysis is repeated with properly translated sequences.

(4) The authors write that they used for the mitogenome phylogenetic analysis a dataset previously published by Williamson et al. (their ref. #31). However, having checked the documentation of the dataset by Williamson et al. (their Supplementary Table S7), I have noticed some discrepancies that prompted me to dig a bit deeper into the dataset presented by Jamy et al. The conclusion is that there is some chaos regarding the taxonomic source of a subset of sequences employed in the phylogenetic analysis of mitochondrial genomes, and at least some of the sequences come from species different from, although related to, those indicated by name in Fig. 4 or the supplementary table “Taxa used for mitochondrial phylogenetic analyses”. I see it is common in phylogenomic analyses to make composite taxa by combining sequences from different closely related taxa, but this must be explicitly stated, the sources must be documented, and there should be a scientific or methodical reason for such an approach. None of the listed criteria are fulfilled by Jamy et al. Hence, for the sake of consistency and accuracy, also to ensure the assembled dataset is a valuable resource for the future, please carefully check the origin of all sequences included and ensure they come from the taxa specified in the trees and tables. I point to three specific cases I could quickly notice, but there might be others that I have missed.

Micromonas pusilla – Williamson et al. state in their Supplementary Table S7 that they included in their dataset the mitochondrial genome from the species *Micromonas commoda*, not *M. pusilla* (it seems they have combined nuclear genes from the latter with mitochondrial genes from the former). As Jamy et al. explain in Materials and methods, they used only the mitochondrial genes from that dataset, so I assumed that the protein sequences employed in their analysis must all come from *M. commoda*. However, having checked just a few of them it turned out that they represent a mixture, some (such as the Atp4 sequence) coming from *M. commoda* (being identical to YP_002860120.1), while others (such as Atp1) coming from *M. pusilla*. However, regardless their origin from different species (in fact very divergent from each other: Atp4 proteins from the two species share only 31% identity!), they are all labelled “Archaeplastida_Viridiplantae_Micromonas_pusilla” by Jamy et al., and despite the respective branch in the phylogenetic tree in Fig. being labelled “*Micromonas pusilla*”. However, this problem just pointed an even more general serious issue of the dataset employed by Jamy et al. While the authors claim that they used in the analysis protein sequences encoded by mitochondrial genomes [see lines 788-789: “We subset this dataset to retain 54 eukaryotic taxa only, and mitochondrial encoded genes (40 protein-coding genes)”], this in reality

does not hold for all the sequences in the analysis. For example, the Atp1 protein from *Micromonas* is encoded by the NUCLEAR genome, which in fact explains why the sequence in the dataset comes from *M. pusilla* and not *M. commoda* (Williamson indeed used the nucleus-encoded proteins from *M. pusilla*). Crucially, this is unlikely to be an isolated issue, as many of the genes included by Jamy et al. in their analysis vary among taxa when it comes to their location: in some they have been retained in the mitogenome, while in others they are nuclear. For example, nad7 to nad11 genes are mitochondrial in some eukaryotes, but nuclear in others (see <https://bmcbiol.biomedcentral.com/articles/10.1186/s12915-024-01824-1>). Furthermore, there is even no guarantee the nuclear versions must be orthologous to the mitochondrial ones (acquisition of xenologs from various external sources, rather than endosymbiotic gene transfer is common here). Hence, I think Jamy et al. should carefully check their dataset and ensure that only mitogenome-encoded proteins are included, of course properly assigned to the actual source organisms.

Mantamonas plastica – here the situation seems to be analogous to the one with *M. pusilla*/*M. commoda*. The dataset by Williamson et al. uses the mitochondrial genome of the different species *Mantamonas sphyrenae*, which is even indicated by Jamy et al. in their own supplementary table entitled “Taxa used for mitochondrial phylogenetic analyses” (583574_2_supp_11198168_t04rss.xlsx). It is then surprising that the species is named “*Mantamonas plastica*” in Fig. 4. However, as with the *M. pusilla*/*M. commoda* case, the solution here is not simply renaming the species: the sequences assigned to the same *Mantamonas* species are again a mixture, some coming from mitogenome of *Mantamonas sphyrenae* but some, surprisingly, indeed derived from *Mantamonas plastica*. My quick analysis indicates the latter concerns Atp1, for which a sequence derived from the transcriptome assembly of *M. plastica* is included. This transcript presumably comes from the mitochondrion of the species, as the *atp1* gene is mitochondrial in *M. sphyrenae* (note that the mitogenome sequence from *M. plastica* is most likely unavailable, or at least I am not aware of it). It thus seems that in this case the problem traces back already to the original dataset by Williamson et al., who perhaps used *M. plastica* mitochondrial sequences they could identify in the transcriptome assembly and complemented the dataset by including additional mitochondrial sequences from *M. sphyrenae*. It is obvious Jamy et al. should ensure they use sequences from the single species and single genome, that is *M. sphyrenae* mitogenome.

Telonema subtile – here again the dataset used by Jamy et al. consists of sequences from two different organisms, i.e. two different telonemids, one really being *Telonema subtile*, as holds e.g. to the Atp1 sequence derived from the transcriptome assembly from this species (as presented in EukProt v3). The other source (check, e.g., Atp4) is a very different telonemid by the mitogenome sequence MN082145, as is also obvious from Supplementary Table S7 by Williamson et al. (not also that those authors indicate the source of data as “Telonemida sp.”, not *Telonema subtile*). The mitogenome sequence comes from the study by Wideman et al. (<https://pubmed.ncbi.nlm.nih.gov/31768028/>). Now the question is what is the identity of the organism behind this sequence. This is not directly apparent, but a recent 18S rRNA-based phylogenetic analysis of Telonemia by Zlatogursky et al. (<https://www.sciencedirect.com/science/article/pii/S2589004225014452>; see their Fig. S2) indicates that the mitogenome MN082145 corresponds to a telonemid labelled “T12” and being distantly related to *T. subtile*. Notably, according to the 18S rRNA tree the T12 organism is more closely related to the telonemid T11 than to *T. subtile* or the telonemid T1, and this topology is consistent with the tree inferred by Jamy et al. from the mitochondrial sequences. Hence, I think the set of sequences grouped to represent the single OUT named by Jamy et al. “*Telonema subtile*” is most

likely dominated by sequences from the “T12 telonemid”, which have drawn it to that particular position in the tree. I leave on the authors to confirm my insight. At any rate, it is obvious the dataset they used for the analysis is a poorly documented chaotic mixture of sequences, not a proper phylogenomic dataset.

Related to the issues regarding the sources of the sequences used, while checking the list of taxa provide by Jamy et al. in their table “Taxa used for mitochondrial phylogenetic analyses”, I could not find in it the species *Cyanophora paradoxa*, although it is included in the tree presented as Fig. 4. It is possible that this is not the only such inconsistency, so please, check the table and ensure all taxa include in the analysis are listed.

(5) As was the case of the plastid trees presented in the original version of the manuscript, the newly added mitochondrial tree (Fig. 4 and associated/underlying files) also uses outdated nomenclature for certain taxa included in it; please use the updated names when rerunning the tree:

Physcomitrella patens – note that the correct name (revising the historical classification based on molecular phylogenetics) is *Physcomitrium patens*, a name now widely adopted by the community (<https://academic.oup.com/plcell/article/32/5/1361/6115584>).

Goniomonas avonlea – note that there is a preprint (now in the revised third version) available at bioRxiv reporting on a thorough taxonomic revision of the traditional extremely broad genus *Goniomonas*, splitting it into multiple genera; see <https://www.biorxiv.org/content/10.1101/2024.07.17.603845v3>. The species previously known as *G. avonlea* has been assigned into a new genus as *Neptunogoniomonas avonlea*. I believe the preprint will soon be published and the updated name should be used by Jamy et al.

Spironema multiciliatum – note that the data in fact come from an organism originally referred to as “*Spironema cf. multiciliatum*”, which means it resembled the species *Spironema multiciliatum* but the authors thought the identification is not certain (<https://pubmed.ncbi.nlm.nih.gov/30429611/>). I believe this designation should be kept.

Other (generally minor) issues:

- I think it would be fair to explicitly mention the preprint by Shrestha et al. 2025 in the text, as it reports on an independent discovery of leptophytes (not named such) based on metagenomic data.
- please insert into each supplementary table a legend including the title of the table and its number (Table SX). Note that as a reviewer I see the tables only as files with strange names like “583574_2_supp_11198166_t04rss.xlsx”, and I am only guessing which file corresponds to which table mentioned in the text. I believe adding these details directly into the tables is also a favour to any prospective reader.
- line 174: “coverage of 2.3x”, line 175: “124x mean coverage” – replace the letter “x” with the symbol for “times (i.e., “x”) and check other possible analogous instances
- line 197-198: “leptophyte plastids genomes” – correct to “leptophyte plastid genomes”
- line 277: “places leptophytes as sister to haptophytes” – is this linguistically OK? Please double check.

- line 314: “encodes 37 protein coding genes as well as 22 tRNA genes” – I am not sure if it is OK to speak about genes to be “encoded” by a genome, I would say that genes are “harboured”, “contained” of something like this by a genome.
- line 397: delete the space after “nucleomorph”
- lines 401-402: “this model is the first to propose haptophytes as the host of the secondary endosymbiosis” – this statement is in fact inconsistent with the narrative that precedes it, as the author explicitly assume the origin of the secondary plastid BEFORE the split of haptophytes and leptophytes. I do see the possibility that leptophytes may best be eventually classified as a haptophyte lineage (this will always be an arbitrary decision), but I think it is at any rate more accurate to write something like this: “this model is the first to propose the haptophyte ancestor as the host of the secondary endosymbiosis”.
- line 412: the placement of the reference to Supplementary Fig. 23 at the end of the sentence is misleading, as it does not show the phylogenies of SELMA proteins and its content is relevant only to the first half of the sentence, so move it to the place just after “gene content comparison (5 vs 33)”.
- line 469: “plastid genomes from culture” – sounds a bit weird to me, I would write “plastid genomes from cultured organisms”
- lines 500-506: would it make better sense to integrate the newly added section (“Check for mitochondrial contamination”) into the section “Creation of a non-redundant plastid genomic database”? I would think that the removal of mitochondrial contaminants should precede estimation of ptMAG completeness (as some of the genes are shared by plastid and mitochondrial genes).
- line 502: remove the space before the period ending the sentence. Note that this is just one example a general problem, when in the newly added text the inserted citations are followed by an extra space that breaks the text and should be removed.
- lines 707-722: note that italics for the gene name “psbO” are used inconsistently in this paragraph, so please, check and unify.
- line 753: “cryptista” – I think the adjective form “cryptist” is needed here
- line 757-758: “This step revealed 13 functions that were present in these mitochondrial genomes” – what are “functions” here and how they can be “present” in genomes? What you list in the sentence are proteins or protein domains encoded by the mitochondrial genomes, so please, rephrase accordingly.
- lines 758-759: you should exert more strive towards consistency, it’s strange to see “Cytochrome c oxidase subunit III” and “Cytochrome C oxidase subunit II periplasmic domain” next to each other. I believe that “cytochrome *c*”, with the letter “c” not capitalized and instead italicized, is the convention adopted by biochemists. The same on line 767.
- line 797: “sequences for each” – for the sake of consistency, capitalize “s” at the beginning of “sequences”
- line 924: replace the page numbers “1 – 21” by the article number “210”, the paper has been published in a journal that does not have continuous pagination (see <https://bmcecolvol.biomedcentral.com/articles/10.1186/1471-2148-10-210>). The same correction needs to be implemented in the same reference in the Supplementary Information file.
- line 926: add the missing year of publication of the paper (2023)

- I did not have enough mental power to carefully check the legends to supplementary figures, but just glancing over the file I have noticed minor issues, such as inconsistent use of italics for gene names (e.g., "psaD" in the legend to Supplementary Fig. 15)